# Last Interglacial climate and sea-level evolution from a coupled ice sheet-climate model

Heiko Goelzer[1,a], Philippe Huybrechts[1], Marie-France Loutre[2], Thierry Fichefet[2]

[1]Earth System Sciences & Departement Geografie, Vrije Universiteit Brussel, Brussels, Belgium

[2]Université catholique de Louvain, Earth and Life Institute, Georges Lemaître Centre for Earth and Climate Research (TECLIM), Louvain-la-Neuve, Belgium.

[a]now at: Institute for Marine and Atmospheric Research, Utrecht University, Utrecht, the Netherlands

Correspondence to: H. Goelzer (h.goelzer@uu.nl, heiko.goelzer@vub.ac.be)

## 1    Abstract

As the most recent warm period in Earth's history with a sea-level stand higher than present, the Last Interglacial (~130 to 115 kyr BP) is often considered a prime example to study the impact of a warmer climate on the two polar ice sheets remaining today. Here we simulate the Last Interglacial climate, ice sheet and sea-level evolution with the Earth system model of intermediate complexity LOVECLIM v.1.3, which includes dynamic and fully coupled components representing the atmosphere, the ocean and sea ice, the terrestrial biosphere and the Greenland and Antarctic ice sheets. In this set-up, sea-level evolution and climate-ice sheet interactions are modelled in a consistent framework.

Surface mass balance change governed by changes in surface meltwater runoff is the dominant forcing for the Greenland ice sheet, which shows a peak sea-level contribution of 1.4 m at 123 kyr BP in the reference experiment. Our results indicate that ice sheet-climate feedbacks play an important role to amplify climate and sea-level changes in the Northern Hemisphere. The sensitivity of the Greenland ice sheet to surface temperature changes considerably increases when interactive albedo changes are considered. Southern Hemisphere

polar and sub-polar ocean warming is limited throughout the Last Interglacial and surface and
sub-shelf melting exerts only a minor control on the Antarctic sea-level contribution with a
peak of 4.4 m at 125 kyr BP. Retreat of the Antarctic ice sheet at the onset of the LIG is
mainly forced by rising sea-level and to a lesser extent by reduced ice shelf viscosity as the
surface temperature increases. Global sea level shows a peak of 5.3 m at 124.5 kyr BP, which
includes a minor contribution of 0.35 m from oceanic thermal expansion. Neither the
individual contributions nor the total modelled sea-level stand show fast multi-millennial time
scale variations as indicated by some reconstructions.
**2    Introduction**
The climate and sea-level evolution of past warm periods in the history of the Earth can give
important insights into expected changes in the future. The Last Interglacial (LIG) in
particular is often considered as a prime candidate for a potential, albeit limited, analogue for
a warmer future world, due to a wealth of available reconstructions of climate and sea level
for this period ~130-115 thousand years (kyr) ago (e.g. Dutton et al., 2015). Problems for the
direct comparison between LIG and future climates arise mainly from the different forcing
responsible for the warming, which can be ascribed to orbital variations during the LIG and to
elevated levels of greenhouse gases in the future. During the LIG, global mean annual surface
temperature is thought to have been 1°C to 2°C higher and peak global annual sea surface
temperatures 0.7°C ± 0.6°C higher than pre-industrial (e.g. Turney and Jones, 2010; McKay
et al., 2011), with the caveat that warmest phases were assumed globally synchronous in these
data syntheses (Masson-Delmotte et al., 2013). These numbers are largely confirmed by a
recent compilation, which resolves the temporal temperature evolution (Capron et al., 2014).
Due to polar amplification, high latitude surface temperatures, when averaged over several
thousand years, were at least 2°C higher than present (Masson-Delmotte et al., 2013) and
were up to 5°C higher over the ice sheets (EPICA community members, 2004; Masson-
Delmotte et al., 2015). These high temperatures had severe consequences for the evolution of
the ice sheets at the onset and during the LIG as evidenced in large variations of sea level
(Rohling et al., 2014; Grant et al., 2012). Coming out of the penultimate glaciation with a sea-
level depression of up to 130 m, the global sea level has peaked during the LIG, estimated at
5.5 to 9 m higher than today (Dutton and Lambeck, 2012; Kopp et al., 2009; 2013), with a
current best estimate of 6 m above the present level (Masson-Delmotte et al., 2013).

A higher-than-present sea-level stand almost certainly implies a complete melting of the Laurentide and Fennoscandian ice sheets and a contribution from the Greenland ice sheet (GrIS), from the Antarctic ice sheet (AIS), or from both. However, ice sheet retreat should not be assumed synchronous in the Northern and Southern hemispheres and between individual ice sheets. Fluctuations in global sea-level during the LIG period (Thompson et al., 2011; Kopp et al., 2013) could be a consequence of differences in the timing of retreat and regrowth between the GrIS and AIS.

Because thus far direct evidence for an AIS contribution to the LIG sea-level high-stand is elusive, support for a contribution from the AIS is usually given as a residual of total sea-level stand minus contributions from the GrIS, thermal expansion (THXP) and glaciers and small ice caps. This illustrates that the attribution problem is so far largely underdetermined. It appears that the lower bound of 5.5 m for the LIG sea-level high-stand (Dutton and Lambeck, 2012; Kopp et al., 2013) could be fully explained by maximum values given in the IPCC AR5 (Masson-Delmotte et al., 2013) for the contributions of the GrIS (1.4 - 4.3 m), glaciers and small ice caps (0.42 ± 0.11 m) and THXP (0.4 ± 0.3 m) combined. However, assuming central estimates for all individual components and the total would indicate an Antarctic contribution of ~ 3 m, which would be in line with the contribution estimated for a collapse of the West Antarctic ice sheet (WAIS) alone (Bamber et al., 2009). An Antarctic component is generally assumed to have foremost come from the WAIS, which is thought to be vulnerable due to its marine-based character. It is often speculated to be sensitive to ocean warming and increased sub-shelf melting (e.g. Duplessy et al., 2007; Holden et al., 2010), possibly caused by the interhemispheric see-saw effect (Stocker, 1998). However, a combination of partial WAIS collapse and some East Antarctic ice sheet (EAIS) retreat is also a possibility due to the large size of the latter. High-end estimates of sea-level change can only be reconciled with an additional EAIS contribution, supposedly from marine-based sectors in the Wilkes and Aurora basins (Pollard et al., 2015; DeConto and Pollard, 2016). One issue complicating the residual argument is the aforementioned possibility of different timing of the GrIS and AIS contributions. Indirect evidence of a WAIS reduction or collapse may come from climate modelling studies that attempt to explain stable-isotope ratios from ice (core) records (Holden et al., 2010; Steig et al., 2015).

The GrIS evolution is somewhat better constrained than the AIS evolution by ice core records both in the central part (GRIP, NGRIP, NEEM) and at the periphery (Dye-3, Camp Century),

even if interpretation of the lower parts of the records remains ambiguous. To this date, none
of the Greenland ice cores shows continuous and undisturbed information back in time
through the LIG and into the penultimate glacial maximum. The relatively high temperatures
during the LIG as reconstructed from the folded lower parts of the NEEM ice core (NEEM
community members, 2013; Landais et al., 2016) seem to be incompatible with the general
view that the ice sheet has lost rather little volume during the LIG (e.g. Robinson et al., 2011;
Colville et al., 2011). Several studies have therefore attempted to identify possible biases in
the NEEM reconstructions (e.g. van de Berg et al., 2013; Merz et al., 2014; Sjolte et al., 2014;
Steen-Larsen et al., 2014; Masson-Delmotte et al., 2015; Merz et al., 2016; Pedersen et al.,
2016). Furthermore, the minimum extent and margin position of the northeastern part of the
ice sheet is not well constrained, leaving room for alternative retreat scenarios (e.g. Born and
Nisancioglu, 2012).
Modelling studies of the GrIS for the entire LIG period so far often use parameterised
representations of the climate forcing (e.g. Huybrechts, 2002), forcing based on time slice
climate experiments (e.g. Born and Nisancioglu, 2012; Stone et al., 2013) or asynchronous
coupling (Helsen et al., 2013), while full coupling between ice and climate models is still a
challenge and limited to models of intermediate complexity (e.g. Robinson et al., 2011). Ice
sheet modelling studies with specific focus on the AIS during the LIG are rare due to the
aforementioned lack of climate and geomorphological constraints for that period. However,
some results on the AIS during the LIG have been presented in studies with main focus on
other time periods (e.g. Huybrechts, 2002) or with interest on longer time scales (e.g. Pollard
and DeConto, 2009; de Boer et al., 2013, 2014). A recent study by DeConto and Pollard
(2016) utilizes simulations of the AIS during the LIG to constrain future sea-level projections.
Despite recent advances (e.g. Capron et al., 2014), the fundamental shortcoming at present for
improving modelled constraints on the LIG ice sheet contribution to sea level with physical
models is the sparse information on LIG polar climate and oceanic conditions. Consequently,
our effort is directed towards studying key mechanisms and feedback processes in the coupled
climate-ice sheet system during the LIG. Here, we present modelling results from the first
fully coupled climate-ice sheet simulation of the LIG period (135 kyr BP to 115 kyr BP) using
ice sheet models of the GrIS and AIS and a climate model of intermediate complexity. In this
set-up LIG sea-level evolution and climate-ice sheet interactions can be modelled in a
consistent framework. With focus on climate and ice sheet changes in Greenland and
Antarctica and corresponding sea-level changes, we compare results from the fully coupled
model to former climate simulations with prescribed ice sheet changes and uncoupled ice
sheet experiments. In the following, we describe the model (section 3) and the experimental
setup (section 4) and present results (section 5) and conclusions (section 6).

## 3   Model description

We use the Earth system model of intermediate complexity LOVECLIM version 1.3, which
includes components representing the atmosphere, the ocean and sea ice, the terrestrial
biosphere and the Greenland and Antarctic ice sheets (Fig. 1). The model has been utilised in
a large number of coupled climate-ice sheet studies (e.g. Driesschaert et al., 2007;
Swingedouw et al., 2008; Goelzer et al., 2011; 2012). Version 1.2 is described in detail in
Goosse et al. (2010). The present set-up of the climate model component is identical to the
model used in Loutre et al. (2014) and Goelzer et al. (2016). Where in the latter study the ice
sheet components were prescribed and used as forcing for the climate model, in the present
work, they are fully two-way coupled with information exchanged every full year. The model
components for the GrIS and AIS are three-dimensional thermomechanical ice-dynamic
models (Huybrechts and de Wolde, 1999), which have been utilised for long-term stand-alone
ice sheet simulations in the past (Huybrechts, 2002). Their behaviour in the coupled system
and detailed analysis of the ice sheet mass balance components are described in Huybrechts et
al. (2011). The surface mass balance model is based on the positive degree-day (PDD)
method (Janssens and Huybrechts, 2000) and distinguishes between snow accumulation,
rainfall and meltwater runoff, all parameterized as a function of temperature. Surface melt is
estimated based on two distinct PDD factors for ice and snow and may be retained and
refreeze in the snow pack. Melt model parameters are unmodified compared to earlier studies
(Goosse et al., 2010; Huybrechts et al., 2011) and have been extensively validated for the
present day (e.g. Vernon et al., 2013).
Because of the relatively coarse resolution of the atmosphere in LOVECLIM (T21), the
higher resolution ice sheet models (10x10 km for Greenland and 20x20 km for Antarctica) are
forced with temperature anomalies and precipitation ratios relative to the pre-industrial
reference climate. Climate anomalies are interpolated to the ice sheet grids using Lagrange
polynomials and the SMB-elevation feedback is accounted for natively in the PDD model on
the ice sheet grid.
The ice sheet models in turn provide the climate model with changing topography, ice sheet
extent (albedo) and spatially and temporally variable freshwater fluxes. The coupling
procedure for these variables is unmodified to earlier versions of the model (Goosse et al.,
2010), while recent model improvements for the ice-climate coupling interface are described
in Appendix A.

## 3.1   Pre-industrial reference model state

A pre-industrial climate state required as a reference for the anomaly forcing mode is
generated by running the climate model with fixed present-day modelled ice sheet
configuration to a steady state. Standard settings for orbital parameters and greenhouse gas
forcing for this experiment are applied following the PMIP3 protocol
(https://pmip3.lsce.ipsl.fr/). The present-day ice sheet configurations for the GrIS and AIS are
the result of prolonging the same stand-alone ice sheet experiments used to initialise the LIG
ice sheet configuration described below towards the present day (Huybrechts and de Wolde,
1999; Huybrechts, 2002; Goelzer et al., 2016).

## 3.2   Northern Hemisphere ice sheet forcing

At the onset of the LIG, large Northern Hemisphere (NH) ice sheets other than on Greenland
were still present and melted away over the course of several millennia. To account for these
ice sheet changes and their impact on climate and ocean evolution, a reconstruction of the
penultimate deglaciation of the NH is necessary for our experiments starting in 135 kyr BP.
Because there is very little geomorphological evidence for NH ice sheet constraints during
Termination II, a reconstruction of NH ice sheet evolution is made by remapping the retreat
after the Last Glacial Maximum according to the global ice volume reconstruction (Lisiecki
and Raymo, 2005) during the onset of the LIG. The same procedure was already used in
earlier work to produce NH ice sheet boundary conditions for climate model simulations
(Loutre et al., 2014; Goelzer et al., 2016).

## 3.3   Modelled sea-level change

The modelled sea-level evolution takes into account contributions from the prescribed NH ice
sheets, the GrIS and AIS and the steric contribution due to density changes of the ocean
water. The only component not explicitly modelled is the contribution of glaciers and small
ice caps, which have been estimated to give a maximum contribution of $0.42 \pm 0.11$ m during
the LIG (Masson-Delmotte et al., 2013) and may contain as much as 5-6 m sea-level
equivalent during glacial times (CLIMAP, 1981; Clark et al., 2001).
Changes in the sea-level contribution of the GrIS can be directly related to its net mass
balance ($MB$), composed of snow accumulation ($ACC$), surface meltwater runoff ($RUN$), basal
melting ($BAS$) and iceberg calving flux ($CAL$):
$MB = ACC - RUN - BAS - CAL$
Since the GrIS model ignores the small bodies of floating ice in the north, these values are
taken over the ice sheet proper only.
For the AIS, $CAL$ is replaced by the flux across the grounding line ($GRF$) in the definition of
the net mass balance of the grounded ice sheet $MB_{gr}$, which needs further corrections to
estimate changes in sea level (see below):
$MB_{gr} = ACC - RUN - BAS - GRF$
The net mass balance of Antarctic floating ice shelves $MB_{fl}$ given here for completeness
includes $GRF$ as an additional source term, but does not contribute to sea-level changes in our
model:
$MB_{fl} = GRF + ACC - RUN - BAS - CAL$
The Antarctic contribution to global sea-level change is calculated taking into account
corrections for grounded ice replacing seawater, grounded ice being replaced by seawater *and*
seawater being replaced by isostatic bedrock movement. These effects are mainly of
importance for the marine sectors of the WAIS. Note that these effects are not considered in
the climate model, which operates with a fixed present-day land-sea mask. The additional
correction for bedrock changes is responsible for a ~3 m lower sea-level contribution at 135
kyr BP compared to taking only changes in volume above floatation into account. This
additional sea-level depression arises from depressed bedrock under the load of the ice in the
marine sectors of the ice sheet.
For the GrIS, the same corrections are applied, where the marine extent of ice grounded
below sea level is parameterised. However, the corrections imply only a ~30 cm lower
contrast to present-day sea level due to GrIS expansion at 135 kyr BP and ~15 cm higher at
130 kyr BP compared to calculations based on the entire grounded ice volume. The change in
sign arises from bedrock changes in delayed response to ice loading changes coming out of
the penultimate glacial period.
The steric component of global sea level considers density changes due to local changes of
temperature and salinity, but global salinity is restored as often done in ocean models to
guarantee stability.

**4    Experimental setup**
**4.1    Model forcing**
All simulations are forced by time-dependent changes in greenhouse gas (GHG)
concentrations and insolation running from 135 kyr BP until 115 kyr BP (Fig. 2). The
radiative forcing associated with the reconstructed GHG levels is below pre-industrial values
for most of this period and hardly exceeds it at ~128 kyr BP (Fig. 2b). The changes in the
distribution of insolation received by the Earth are computed from the changes in the orbital
configuration (Berger, 1978) and represent the governing forcing during peak LIG conditions
(Fig. 2a).
In order to account for coastline changes and induced grounding line changes, both ice sheet
models are forced by changes in global sea-level stand (Fig. 2c) using a recent sea-level
reconstruction based on Red Sea data (Grant et al., 2012). The chronology of this data is
thought to be superior compared to sea-level proxies based on scaled benthic $\delta^{18}O$ records
(Grant et al., 2012; Shakun et al., 2015). In this sea-level forcing approach, local changes due
to geoidal eustasy are not taken into account, which would result in lower amplitude sea-level
changes close to the ice sheets, but that would not be consistent with the stand-alone spin-up
of the ice sheet models.
As mentioned earlier, the ice sheet models are forced with temperature anomalies relative to
the pre-industrial reference climate. To ensure a realistic simulation of the GrIS evolution, the
temperature anomaly forcing from the climate model over the GrIS needs to be rescaled. In
absence of such scaling, the ice sheet almost completely melts away over the course of the
LIG in disagreement with the ice core data, which suggests a large remaining ice sheet during
the LIG (Dansgaard et al., 1982; NEEM community members, 2013). In the absence of firm
constraints on the climate evolution over the ice sheet, the temperature scaling in the present
study represents a pragmatic solution to produce a  GrIS evolution reasonably in line with ice
core constraints on minimum ice sheet extent during the LIG. The scaling is only applied for
the GrIS, since we have not identified a physical process that would justify a similar
procedure for to the AIS.

## 4.2   Reference simulation and sensitivity experiments

Our reference simulation is a fully coupled experiment with a uniform scaling of the
atmospheric temperature anomaly over Greenland with a factor of R=0.4, which was chosen
to give a good match to constraints on minimum extent of the GrIS during the LIG.
Additional sensitivity experiments are listed in Table 1 and are described in the following.
Two sensitivity experiments with modified scaling (R=0.5, 0.3) are added to evaluate the
impact on the results. The range of parameter R is chosen to retain an acceptable agreement of
the minimum GrIS extent during the LIG with reconstructions. In practice, the high scaling
factor (R=0.5) is chosen to produce the smallest minimum ice sheet extent, which still has ice
at the NEEM site. The low scaling factor (R=0.3) was adopted to produce the smallest
minimum ice sheet extent, which is still covering Camp Century.
The three fully coupled experiments are complemented by additional sensitivity experiments,
in which the ice sheet models are forced with (modified) climate forcing produced by the
fully coupled reference run. These experiments serve to study ice sheet sensitivity in response
to changes in the climate forcing and are also used to evaluate ice sheet-climate feedbacks by
comparing the coupled and uncoupled system. The ice sheet evolution in the forced reference
experiment (ice sheet model run offline with the recorded climate forcing of the coupled
reference run) should by construction be identical to the response in the fully coupled run, and
only serves as a control experiment. Two additional forced experiments have been run with
modified temperature scaling for the GrIS (R=0.5, 0.3), which can be directly compared to the
respective fully coupled experiment.
For the AIS, an experiment with suppressed sub-shelf melting has been performed to isolate
the effect of ocean temperature changes on the ice volume evolution and sea-level
contribution.

## 4.3 Initialisation of the reference simulation

The goal of our initialisation technique is to prepare a coupled ice sheet-climate model state for the transient simulations starting at 135 kyr BP exhibiting a minimal coupling drift. Both ice sheet models are first integrated over the preceding glacial cycles in order to carry the long-term thermal and geometric history with them (Huybrechts and de Wolde, 1999; Huybrechts, 2002; Goelzer et al., 2016). The climate model is then initialized to a steady state with ice sheet boundary conditions, greenhouse gas forcing and orbital parameters for the time of coupling (135 kyr BP). When LOVECLIM is integrated forward in time in fully coupled mode, the climate component is already relaxed to the ice sheet boundary conditions. The mismatch between stand-alone ice sheet forcing and climate model forcing is incrementally adjusted in the period 135-130 kyr BP with a linear blend between the two to minimize the effect of changing boundary conditions for the ice sheet model. A small, unavoidable coupling drift of the ice sheet component arises from a switch of spatially constant to spatially variable temperature and precipitation anomalies at the time of coupling, but is uncritical to the results.

## 5 Results

The modelled LIG climate evolution and comparison with proxy reconstructions were presented in detail in two earlier publications (Loutre et al., 2014; Goelzer et al., 2016) for the same climate model setup. In the following, we focus on differences to those two works that arise from a different ice sheet evolution and from the incorporation of feedbacks between climate and ice sheets that are taken into account in our present, fully coupled approach. In addition, we present results pertaining to the ice sheet evolution and simulated sea-level changes.

## 5.1 Climate evolution

Global annual mean near-surface air temperature in the reference experiment (Fig. 3) shows a distinct increase until 129 kyr BP in response to orbital and greenhouse gas forcing (Fig. 2) and to an even larger extent in response to changes in ice sheet boundary conditions. The peak warming reaches 0.3 °C above the pre-industrial at 125.5 kyr BP. Thereafter, cooling sets in and continues at a much lower rate compared to the rate of warming before 129 kyr BP. The importance of ice sheet changes is illustrated by comparing the reference experiment with a

climate simulation (Loutre et al., 2014) forced by insolation and GHG changes only (noIS) and with a one-way coupled climate model run (Goelzer et al., 2016) forced with prescribed NH, Antarctic and Greenland ice sheet changes (One-way). The fully coupled experiment exhibits a global mean temperature evolution during the LIG, which is very similar to One-way (Fig. 3). A much larger temperature contrast at the onset of the LIG in the reference experiment compared to noIS arises mainly from changes in surface albedo and melt water fluxes of the NH ice sheets, which freshen the North Atlantic and lead to a strong reduction of the Atlantic meridional overturning circulation (Loutre et al., 2014). All three simulations show only small differences in the global mean temperature evolution after 127 kyr BP. The episode of relative cooling in the reference experiment with a local temperature minimum at 128 kyr BP is due to cooling of the Southern Ocean (SO) and sea-ice expansion in response to large Antarctic freshwater fluxes caused mainly by the retreat of the WAIS. This mechanism was already described by Goelzer et al. (2016), but now occurs 2 kyr later in the fully coupled experiment, due to a modified timing of the AIS retreat. The effect of including ice-climate feedbacks by means of a two-way coupling is otherwise largely limited to the close proximity of the ice sheets as discussed in the following.

## 5.2 Greenland ice sheet

The Greenland ice sheet evolution over the LIG period is largely controlled by changes in the surface mass balance dominated by surface meltwater runoff (Fig. 4c). Specifically, summer surface melt water runoff from the margins is the dominant mass loss of the GrIS after 130 kyr BP, when the ice sheet has retreated largely on land. Due to increased air temperatures over Greenland, the mean accumulation rate (averaged over the ice covered area) is consistently above the present-day reference level after 128 kyr BP, but increases to at most 18% higher (not shown). In contrast, net accumulation over grounded ice (Fig. 4b) is strongly modulated by the retreat of the ice sheet and exhibits a marked increase towards the end of the simulation as ice sheet grows again and into regions with higher precipitation. Conversely, surface meltwater runoff over the Greenland ice sheet shows an up to threefold increase compared to the present day at the beginning with consistently higher-than present values between 130.5 kyr to 120.5 kyr BP (Fig. 4c). Temperature anomalies responsible for the increased runoff are on average above zero between 129.5 kyr to 120.5 kyr BP and peak at 1.3 °C (after scaling) around 125 kyr BP (Fig. 4a). The calving flux (Fig. 4d) decreases as surface melting and runoff (Fig. 4c) increase, removing some of the ice before it can reach the coast

and also as the ice sheet retreats from the coast (cf. Fig. 5), in line with decreasing area and
volume (Fig. 4f). In the second half of the experiment, runoff decreases with decreasing
temperature anomalies and the calving flux increases again with increasing ice area and
volume. The net mass balance of the ice sheet (Fig. 4e) reflects the compounded effect of all
components with negative values before and positive values after the time of minimum
volume.
Entering the warm period, the furthest retreat of the ice sheet occurs in the southwest and
northwest (Fig. 5), accompanied by an overall retreat from the coast. At the same time, the ice
sheet gains in surface elevation over the central dome due to increased accumulation. By 115
kyr BP, the ice sheet has regrown beyond its present-day area almost everywhere and contact
with the ocean is increasing. The GrIS volume change implies a sea-level contribution peak of
1.4 m at 123 kyr BP (Fig. 11a). For the two sensitivity experiments (High, Low) with
modified scaling (R=0.5, 0.3), the contribution changes to 2.7 m and 0.65 m, respectively,
crucially controlled by the scaling factor (Table 2).
NEEM ice core data (NEEM community members, 2013) and radiostratigraphy of the entire
ice sheet (MacGregor et al., 2015) indicate that the NEEM ice core site was ice covered
through the entire Eemian as is the case for our reference experiment. Elevation changes from
that ice core are however not very well constrained and even if they were, would leave room
for a wide range of possible retreat patterns of the northern GrIS (e.g. Born and Nisancioglu,
2012). The Camp Century ice core record contains some ice in the lowest part with a colder
signature then ice dated as belonging to the Eemian period (Dansgaard et al., 1982). It is
likely that this ice is from before the Eemian even in view of possible disturbance of the lower
levels, which was shown to exist for the NEEM core site (NEEM community members,
2013). In view of this evidence, the northwestern retreat of the ice sheet in our reference
simulation may be too far inland, as a direct result of the largely unconstrained climatic
forcing in this area. It was shown that a different climate forcing could produce a larger
northern retreat still in line with the (limited) paleo evidence (Born and Nisancioglu, 2012).
Some more thinning and retreat in the south is also possible without violating constraints on
minimal ice sheet extent from Dye-3 (Dansgaard et al., 1982). LIG ice cover of the Dye-3 site
is not a necessity when taking into consideration that older ice found at the base of the core
could have flowed in from a higher elevation.
A comparison of modelled temperatures in North-East Greenland (Fig. 6) shows differences
of up to 5 degrees between annual mean and summer temperatures in the reference
experiment. Comparison with temperature reconstructions based on the NEEM ice core
record indicates that the steep temperature increase marking the onset of the LIG occurs 2-3
kyr earlier in the model compared to the reconstructions. The amplitude of modelled summer
temperatures attains levels of the central estimate, while annual mean temperatures fall in the
lower uncertainty range of the reconstructions. Temperatures exceeding the central estimate
are only reached in the One-way experiment, which exhibits a somewhat different retreat
pattern of the GrIS due to the different climate forcing (Goelzer et al., 2016).
The strength of the ice-climate feedback on Greenland was examined by comparing additional
experiments in which the coupling between ice sheet and climate is modified. Results from
the fully coupled model are compared to those from forced ice sheet runs that are driven with
the climate forcing from the coupled reference model run (Table 2 and Fig. 7a). The scaling
of Greenland forcing temperature is set to a magnitude of 0.3 (Forced low), 0.4 (Forced
reference) and 0.5 (Forced high), respectively. When the feedback between ice sheet changes
and climate is included in the coupled experiments, the warming over the margins is
considerably increased (reduced) for experiment High (Low) compared to the respective
forced experiments. Consequently, ice volume changes show a non-linear dependence on the
temperature scaling for the fully coupled run, while they are near linear for the forced runs
(Table 2 and Fig. 7a). The dominant (positive) feedback mechanism arises from how
changing albedo characteristics are taken into account for a melting ice sheet surface (Fig.
7b). The underlying surface type with different characteristic albedo values for tundra and ice
sheet is determined by the relative amount of ice cover, which is modified when the area of
the ice sheet is changing. On much shorter time scales, the albedo can change due to changes
in snow depth and also due to changes of the snow cover fraction, which indicates how much
surface area of a grid cell is covered with snow (Fig. 7b). Both snow processes lead to lower
albedo and increased temperatures in places where the ice sheet starts melting at the surface.
The difference in warming between forced and fully coupled experiments is however located
over the ice sheet margins and this does not have a considerable influence on the NH or
global temperature response. The snow albedo effects are near-instantaneous and their
importance for the ice sheet response underline earlier findings that a basic albedo treatment
is an essential aspect of a coupled ice–climate modelling system (e.g. Robinson and Goelzer,
2014). A comparatively smaller effect and operating on much longer time scales arises from
the retreating ice sheet margin being replaced by tundra with a lower albedo (Fig. 7b).

## 5.3   Antarctic ice sheet

The annual mean air temperature anomaly over Antarctica (averaged over grounded ice)
increases at the beginning of the experiment to reach a peak of up to 2°C at 125 kyr BP (Fig.
8a), before cooling sets in and continues until 115 kyr BP. The warming before the peak is
around a factor two faster than the cooling afterwards, with both transitions being near linear
on the millennial time scale. The surface climate over the AIS appears to be largely isolated
from millennial time scale perturbations occurring in the SO in response to changing
freshwater fluxes in both hemispheres (Goelzer et al., 2016). While freshwater fluxes from the
retreating AIS itself lead to sea-ice expansion and surface cooling in the SO, freshwater fluxes
from the decay of the NH ice sheets are communicated to the Southern Hemisphere (SH) by
the interhemispheric see-saw effect (Goelzer et al., 2016). Pre-industrial surface temperature
levels are first reached at 128 kyr BP and then again at 118 kyr BP after cooling throughout
the second half of the experiment. The accumulation (over grounded ice) shows an initial
increase in line with the higher temperatures until 130 kyr BP (Fig. 8b) but records a changing
grounded ice sheet area further on, which mostly follows the marked retreat and later slow
regrowth of the ice sheet. Relative to the pre-industrial, the mean accumulation rate (averaged
over grounded ice) increases at most 20 % in annual values and up to 12 % for the long-term
mean (not shown). As a consequence of the surface forcing, the AIS shows a small volume
gain until 130.5 kyr BP (Fig. 8f) due to increase in precipitation before a large-scale retreat of
the grounding line sets in. The surface meltwater runoff over grounded ice equally increases
with increasing temperature (Fig. 8c) but remains of negligible importance (note difference of
vertical scales between panel b and c in Fig. 8) for the net mass balance (Fig. 8e) of the ice
sheet. This is also the case for basal melting under the grounded ice sheet (not shown).
Changes in the sub-shelf melt rate play an important role for the present mass balance of the
AIS and are often discussed as a potential forcing for a WAIS retreat during the LIG (e.g.
Duplessy et al., 2007; Holden et al., 2010) and during the last deglaciation (Golledge et al.,
2014). The average sub-shelf melt rate diagnosed for the area of the present-day observed ice
shelves in our reference simulation (Fig. 8d) increases to at most 20 % above the pre-
industrial with a peak in line with the air temperature maximum (Fig. 8a, d). However, ocean
warming to above pre-industrial temperatures occurs already before 130 kyr BP (not shown),
more than 2 kyr earlier compared to the air temperature signal. This is a consequence of the
interhemispheric see-saw effect (Stocker, 1998), which explains SO warming and cooling in
the North Atlantic as a consequence of reduced oceanic northward heat transport due to a
weakening of the Atlantic meridional overturning circulation (Goelzer et al., 2016).
Ice sheet area and volume (Fig. 8f) decrease rapidly between 129 and 127 kyr BP, and
indicate a gradual regrowth after 125 kyr BP, also visible in the net mass balance (Fig. 8e).
Those changes arise mainly from a retreat and re-advance of the WAIS (Fig. 9). In our model,
the ice sheet retreat exhibits characteristics of an overshoot behaviour due to the interplay
between ice sheet retreat and bedrock adjustment. The rebound of the bedrock, which is
initially depressed under the glacial ice load, is delayed compared to the relatively rapid ice
sheet retreat, giving rise to a grounding-line retreat well beyond the pre-industrial steady-state
situation. These results are in line with earlier work with a stand-alone ice sheet model
(Huybrechts, 2002), but also rely on a relatively large glacial-interglacial loading contrast in
these particular models. The sea-level contribution above the present-day level from the AIS
peaks at 125 kyr BP at 4.4 m (Fig. 11b).
Sensitivity experiments, in which specific forcing processes are suppressed, show that surface
melting (not shown) and sub-shelf melting play a limited role for the AIS retreat in our
experiments. The sea-level contribution peak in an experiment with suppressed sub-shelf
melting (Fig. 11b) is about 40 cm lower compared to the reference experiment and remains
around one meter lower between 123 kyr BP until the end of the experiment. The difference
between the experiments at a given point in time arises from a lower overall sea-level
contribution when sub-shelf melting is suppressed, but also from a difference in timing
between both cases. The dominant forcing for the AIS retreat in our model is a combination
of rising global sea level and increasing surface temperature, which leads to increasing
buoyancy and reduced ice shelf viscosity, respectively. The relative timing between sea-level
forcing (Fig. 2c) and temperature forcing (Fig. 8a) is therefore of critical importance for the
evolution of the ice sheet at the onset of the LIG.
The limited effect of surface melting and sub-shelf melting on the sea-level contribution is
ultimately due to a limited magnitude of surface temperature and ocean temperature changes.
The limited Antarctic and SO temperature response has already been highlighted in earlier
studies with the same climate component (Loutre et al., 2014; Goelzer et al., 2016) and is

confirmed here with a fully coupled model. The feedback mechanism suggested by Golledge et al. (2014) for Termination I, which draws additional heat for sub-shelf melting from freshwater-induced SO stratification and sea-ice expansion is also active in our experiment, but too short-lived and of too little amplitude to lead to substantially increased melt rates. Our limited AIS response to climatic forcing is also in line with other modelling results for the LIG period (Pollard et al., 2015), albeit with a different forcing strategy, where substantial retreat of marine based sectors of the EAIS can only be achieved by including special treatment of calving fronts and shelf melting, which was not included here.

As mentioned earlier, direct constraints of the AIS configuration during the LIG are still lacking. Goelzer et al. (2016) suggested that the timing of the main glacial-interglacial retreat of the AIS could be constrained by a freshwater induced oceanic cold event recorded in ocean sediment cores (Bianchi and Gersonde et al., 2002). The main retreat in their one-way coupled climate model run happened ~129.5 kyr BP, a timing predating the time of retreat in the fully coupled model by ~2 kyr due to the difference in atmospheric and oceanic forcing. This lag is also visible in modelled temperature changes over the East Antarctic ice sheet (EAIS) that have been compared to temperature reconstructions for four ice core locations (Fig. 10). One-way and Reference show a larger temperature contrast, better in line with the ice core data, compared to the experiment with a fixed ice sheet (noIS). However, the timing of warming was better matched in One-way with an earlier ice sheet retreat.

It is noteworthy in this context that the prescribed sea-level forcing imposes an important control on the timing of the Antarctic retreat and is a source of large uncertainty. We have only used the central estimate of the Grant et al. (2012) sea-level reconstruction, but propagated dating uncertainties could accommodate a shift of the forcing by up to 1 kyr either way. Former experiments (not shown) have indicated that the main retreat appears another 2 kyr later when a sea-level forcing based on a benthic $\delta^{18}O$ record (Lisiecki and Raymo, 2005) is used instead of the sea-level reconstruction of Grant et al. (2012).

## 5.4  Thermal expansion of the ocean

The steric sea-level component due to ocean thermal expansion (Fig. 11c) is largely following the global temperature evolution (Fig. 3), but is also strongly modified by changes in ice sheet freshwater input. Ocean expansion is rapid during peak input of freshwater and stagnant

during episodes of decreasing freshwater input. This is because the net ocean heat uptake is
large when freshwater input peaks, which happens in three main episodes in our experiment.
Two episodes of freshwater input from the NH centred at 133.6 and 131.4 kyr BP are
followed by an episode of combined input from the NH and the AIS centred at 128.2 kyr BP
(not shown). The anomalous freshwater input leads to stratification of the surface ocean, sea-
ice expansion and reduction of the air-sea heat exchange, effectively limiting the ocean heat
loss to the atmosphere. This implies that global sea-level rise due to ice sheet melting is
(weakly and temporarily) amplified by the freshwater impact on ocean thermal expansion. We
simulate a peak sea-level contribution from thermal expansion of 0.35 m at 125.4 kyr BP,
which forms part of a plateau of high contribution between 127.3 and 124.9 kyr BP (Fig. 11c).
The amplitude is within the range of current estimates of $0.4 \pm 0.3$ m (McKay et al., 2011;
Masson-Delmotte et al., 2013).

### 5.5   Global sea-level change

Combining contributions from GrIS, AIS and thermal expansion, global sea level peaks at
~5.3 m at 124.5 kyr BP (Fig. 12c) with a slow decrease thereafter as first the AIS and 2 kyr
later the GrIS start to regrow. For the AIS the model indicates a clear asymmetry between
relatively fast retreat and much slower regrowth (Fig. 12b).
Modelled GrIS and AIS sea-level contributions together with prescribed NH sea level are
within the 67% confidence interval of probabilistic sea-level reconstructions (Kopp et al.,
2009) for the period ~125-115 kyr BP (Fig. 12). The last 20 m rise in sea-level contributions
from the NH (including Greenland) is steeper and occurs 1~2 kyr earlier in our model
compared to what the reconstructions suggest, which is consequently also the case for the rise
in global sea level at the onset of the LIG. The Antarctic retreat in our model is more rapid
compared to the reconstruction and does not show the regrowth ~131-129 kyr BP suggested
by the data from Kopp et al. (2009). The modelled ice sheet evolution in our reference run
reproduces well the global average sea-level contribution 125-115 kyr BP based on the best
estimate of Kopp et al. (2009) when taking into account the modelled steric contribution (0.35
m) and assuming a maximum possible contribution (0.42+-0.11 m) of glaciers and small ice
caps (Masson-Delmotte et al., 2013). The multi-peak structure of global sea-level
contributions during the LIG suggested by the median reconstructions (Kopp et al., 2009;
2013) is not reproduced with our model (Fig. 12c), mainly owing to the lack of such variation
in the climate forcing and to the long response times of the ice sheets during regrowth to
changing climatic boundary conditions.

## 6  Discussion
### 6.1  Global sea-level change
While the median projections in Kopp et al., (2009) visually suggest a double-peak structure
in the global sea-level evolution during the LIG, our results show that the uncertainty range is
wide enough to accommodate a global sea-level trajectory based on physical models without
intermediate low stand. The simulated climate forcing in our case does not favour the
presence of such variability, which admittedly could be due to missing processes or feedbacks
in our modelling. Nevertheless, based on our own modelling results and the Kopp et al.,
(2009) reconstruction we are not convinced reproducing a double peak structure is a given
necessity.
### 6.2  Greenland ice sheet evolution
The temperature anomaly over central Greenland in the coupled model shows a flat maximum
around 127 kyr BP (Fig. 4a), similar to the global temperature evolution, but 2 kyr earlier
compared to the NEEM reconstruction (NEEM community members, 2013). If assuming
present-day configuration and spatially constant warming, ice mass loss from the GrIS could
be expected to occur approximately as long as the temperature anomaly remains above zero,
which is the case until ~ 122 kyr BP in our reference model and until ~ 119 kyr BP in the
NEEM reconstruction. With a lower surface elevation, the time the ice sheet starts to gain
mass again would be further delayed. Even with considerable uncertainty due to uncertain
spatial pattern of the warming, which modifies this simple reasoning, it is clear that the peak
sea-level contribution from the GrIS has to occur late during the LIG. This argument is
confirmed by our model results and in line with conclusions recently drawn by Yau et al.
(2016) based on data from another Greenland ice core and modelling. Based on the same
argument, there is no evidence in the reconstructed NEEM temperature evolution suggesting a
regrowth or substantial pause of melting of the GrIS any time during the LIG.
The need for scaling the temperature forcing to produce a realistic GrIS evolution would
equally apply when our ice sheet model were forced directly with the temperature
reconstructed from the NEEM ice core record (NEEM community members, 2013). It appears
that practically any ice sheet model with (melt parameters tuned for the present day) would
project a near-complete GrIS meltdown, if the amplitude and duration of warming suggested
by the NEEM reconstructions would apply for the entire ice sheet. This problem would be
further amplified if insolation changes were explicitly taken into account in the melt model
(van de Berg et al., 2011; Robinson and Goelzer, 2014). We refer to this mismatch between
reconstructed temperatures and assumed minimum ice sheet extent as the "NEEM paradox"
(see also Landais et al., 2016). Several attempts to solve this paradox have been made by
suggesting possible biases in the interpretation of the relationship between isotope ratio and
temperature, which may not be assumed temporally and spatially constant (e.g. Merz et al.,
2014; Sjolte et al., 2014; Steen-Larsen et al., 2014; Masson-Delmotte et al., 2015) or may be
affected by changes in the precipitation regime (van de Berg et al., 2013) and sea ice
conditions (Merz et al., 2016; Pedersen et al., 2016). From a modelling point of view, the
decisive question is over what spatial extent and when during the year the temperature
reconstruction (and possible future reinterpretations) for the NEEM site should be assumed. A
central Greenland warming of large magnitude could only be reconciled with the given
geometric constraints if a (much) lower warming was present over the margins and during the
summer, which is where and when the majority of the mass loss due to surface melting is
taking place.

## 6.3  Antarctic ice sheet evolution

The main forcing for WAIS retreat during Termination II and the LIG was found to be global
sea-level rise from melting of the NH ice sheets, and to a lesser extent surface warming
causing a gradual thinning of the ice shelves as the ice softened, contributing to an additional
grounding-line retreat as there is less buttressing and increased thinning at the grounding line.
These processes also played during Termination I and into the Holocene in simulations with
the same ice sheet model (Huybrechts, 2002), but did not produce an overshoot in the sense
that the WAIS retreated further inland from its present-day extent. The difference in
behaviour between the LIG and the Holocene is mainly the speed of sea-level rise, which was
slower during Termination I, and the fact that the global sea-level stand itself did not
overshoot the present-day level during the Holocene, giving a less strong forcing. Of
particular importance to generate overshoot behaviour is the speed of sea-level rise relative to
the speed of bedrock rebound as both control the water depth at the grounding line and hence,
grounding-line migration because of the criterion for floatation (hydrostatic equilibrium). If
the sea-level rise is fast compared to the bedrock uplift, grounding line retreat will be
enhanced, as was the case during Termination II in our model experiments. In that case, the
grounding line is able to retreat to a more inland position until the lagged bedrock rebound
halts and reverses the process. If on the contrary, the bedrock rebound after ice unloading is
fast compared to the sea-level rise, this will tend to dampen grounding-line retreat, as shown
in the sensitivity experiments discussed in Huybrechts (2002).
Ice shelf viscosity changes also played a role during Termination II and the LIG, but were not
found to be the dominant forcing. The response time of viscosity changes in the ice shelves is
governed by vertical heat transport, having a typical characteristic time scale of 500 years
with respect to surface temperature (Huybrechts and de Wolde, 1999). The mechanism can
only be effective over longer time scales and for a limited warming such as occurred during
the LIG as otherwise the ice shelves would largely disintegrate from both surface and basal
melting. In future warming scenarios, the effect of shelf viscosity changes is therefore usually
too slow compared to the anticipated direct effect of increased surface and basal melting rates.
For instance, in the future warming scenarios performed with LOVECLIM under $4xCO_2$
conditions (Huybrechts et al., 2011), shelf melt rates increased 5-fold, and the ice shelves
were largely gone before they had a chance to warm substantially. The implication is that
analogies between these different time periods should be reserved on account of different
processes playing at different time scales.

## 6.4   Comparison with other work

An earlier attempt to model the coupled climate-ice sheet evolution for the Greenland ice
sheet over the LIG period (Helsen et al., 2013) applied an asynchronous coupling strategy to
cope with the computational challenge of such long simulations. While it can be assumed that
their high-resolution regional climate model provides a more accurate climate forcing
compared to our approach, we still lack substantial climate and ice sheet reconstructions for
the LIG period to effectively validate model simulations. This applies to the simulated climate
as well as to the resulting ice sheet geometries, limiting attempts to constrain the GrIS sea-
level contribution to arrive at relatively large and overlapping uncertainty ranges (e.g.
Robinson et al., 2011; Stone et al., 2013; Helsen et al., 2013). Incidentally, our range of
modelled GrIS sea-level contribution is in very close agreement with recent results from a
large ensemble study of the LIG sea-level contribution constrained against present-day
simulations and elevation changes at the NEEM ice core site (Calov et al., 2015). Despite a
possible degree of coincidence in this particular case, the overlap between results reached by
largely different methods is indicative of the lack of better constraining data needed to arrive
at much narrower uncertainty ranges.
**6.5  Model limitations**
Simulating the fully coupled ice sheet-climate system for the entire duration of the LIG as
presented here is an important step forward for a better understanding of the Earth system
during this period. However, our attempt deserves a critical discussion of the limitations of
the model setup.
A so far unavoidable side effect to running a fully coupled model for several thousands of
years is the limited horizontal resolution of the atmospheric model. The katabatic wind effect
discussed by Merz et al. (2014) and other small-scale circulation patterns are therefore likely
underrepresented. A quantification of how much the strength of ice sheet-climate feedbacks
depends on spatial resolution of the climate model would be an interesting study, but is not
something we could add to with our model set-up.
The applied PDD scheme has been extensively validated with results of more complex
Regional Climate Models for simulations of the recent past (e.g. Vernon et al., 2013), but
several studies point to limitations of this type of melt model when applied for periods in the
past with a different orbital configuration (e.g. van de Berg et. al., 2011; Robinson and
Goelzer, 2014). Their results indicate that the stronger northern summer insolation during the
LIG should result in additional surface melt on the Greenland ice sheet compared to
simulations based on temperature changes alone. We note that this suggests an
underestimation of LIG melt with the PDD model and increased melt if it was corrected for.
Thus, including an additional melt contribution due to insolation would further increase the
contrast of the NEEM paradox in our simulation. Our modelling therefore provides no
arguments to support the contention that the limited LIG warming implied over Greenland
would be indicative of an overly sensitive ice sheet and mass balance model.
Instead, the applied scaling of the temperature anomaly forcing for the GrIS is a necessity to
keep the ice sheet from losing too much mass during the warm period and to maintain ice
sheet retreat to within limits of reconstructions. Clearly, this implies a limited predictive
capability of our model, which is now forced to comply with the given constraints on
minimum ice extent during the LIG. However, the Antarctic simulation would not be strongly
affected by changes in the melt model due to the limited role of surface melting for the
evolution of the AIS during the LIG.
The sea-saw effect evoked by NH freshwater forcing leads to millennial time scale
temperature variations in the SO, but the surface climate over the AIS is hardly affected in our
simulations. Despite some improvement when ice sheet changes are included, the limited
Antarctic temperature response appears to be a general feature of the LOVECLIM model (e.g.
Menviel et al., 2015), which fails to reproduce a several degree warming during the LIG
reconstructed at deep ice core locations. We suspect that the limited resolution of the
atmospheric model contributes to this shortcoming but we have not been able to quantify that.
**6.6  Possible improvements**
Uncertainty in the age model of the Grant et al. (2012) sea-level reconstruction could in
principle be used to force the AIS to an earlier retreat, better in line with the Kopp et al.
(2009) reconstructions. We have not attempted that, since other uncertainties, in particular in
the climate forcing are large and do not warrant to attempt a precise chronology. Earlier
experiments (not shown) indicate however that using a benthic $\delta^{18}$O-stack (Lisiecki and
Raymo, 2005) would lead to an even later retreat of the AIS and thus increase the mismatch
with the Kopp et al. (2009) reconstruction. Ultimately, it would be desirable to apply a
consistent sea-level forcing, based on physical models (e.g. de Boer et al., 2014). However,
this would require a prognostic model of NH ice sheet evolution (e.g. Zweck and Huybrechts,
2005) and a general solution of the sea-level equation, which would considerably increase
complexity and required resources.
Targeting model limitations described in the previous sub-section hinges to a large extent on
improving the atmospheric component of the climate model, which equally goes hand in hand
with an increase in needed computational resources. Given the large remaining uncertainties
in the climate forcing during the LIG and a limited impact of an improved physical
approximation for ice flow applied to future projections (Fürst et al., 2013), we consider
improving the representation of ice sheet dynamics as of secondary importance. However,
fully physical treatment of the surface mass balance solution in a coupled climate-ice sheet
model framework, as currently targeted by several groups (e.g. Nowicki et al., 2016) appears
like a promising development that may eventually be applied for paleo applications such as
the transient LIG simulations of interest in the present paper.

## 7    Conclusion

We have presented the first coupled transient simulation of the entire LIG period with
interactive Greenland and Antarctic ice sheet components. In our results, both ice sheets
contribute to the sea-level high stand during the Last Interglacial, but are subject to different
forcing and response mechanisms. While the GrIS is mainly controlled by changes in surface
melt water runoff, the AIS is only weakly affected by surface and sub-shelf melting. Instead,
grounding line retreat of the AIS is forced by changes in sea level stand and to a lesser extent
surface warming, which lowers the ice shelf viscosity. The peak GrIS contribution in our
reference experiment is 1.4 m. However, this result is strongly controlled by the need to scale
the climate forcing to match existing ice core constraints on minimal ice sheet extent. This
shortcoming in our modelling reflects the NEEM paradox, that strong warming over the ice
sheet coincides with limited mass loss from the GrIS, indicative of a fundamental missing link
in our understanding of the LIG ice sheet and climate evolution. The Antarctic contribution is
4.4 m predominantly sourced from WAIS retreat. The modelled steric contribution is 0.35 m,
in line with other modelling studies. Taken together, the modelled global sea-level evolution
is consistent with reconstructions of the sea-level high stand during the LIG, but no evidence
is found for sea-level variations on a millennial to multi-millennial time scale that could
explain a multi-peak time evolution. The treatment of albedo changes at the atmosphere-ice
sheet interface play an important role for the GrIS and constitute a critical element when
accounting for ice sheet-climate feedbacks in our fully coupled approach. Large uncertainties
in the projected sea-level changes remain due to a lack of comprehensive knowledge about
the climate forcing at the time and a lack of constraints on LIG ice sheet extent, which are
limited for Greenland and virtually absent for Antarctica.

## 8    Data availability

The    LOVECLIM    version    1.3    model    code    can    be    downloaded    from
http://www.elic.ucl.ac.be/modx/elic/index.php?id=289.


**Appendix A: Ice-climate coupling improvements**


Compared to earlier versions of the model (Goosse et al., 2010), recent model improvements
for the coupling interface between climate and ice sheets have been included for the present
study. Ocean temperatures surrounding the AIS are now used directly to parameterise
spatially explicit sub-ice-shelf melt rates, defining the flux boundary condition at the lower
surface of the AIS in contact with the ocean. The sub-shelf basal melt rate $M_{shelf}$ is
parameterised as a function of local mid-depth (485-700 m) ocean-water temperature $T_{oc}$
above the freezing point $T_f$ (Beckmann and Goosse, 2003):
$$M_{shelf} = \rho_w c_p \gamma_T F_{melt} (T_{oc} - T_f) / L \rho_i,$$
where $\rho_i$=910 kg m$^{-3}$ and $\rho_w$=1028 kg m$^{-3}$ are ice and seawater densities, $c_p$=3974 J kg$^{-1}$ °C$^{-}$
$^1$ is the specific heat capacity of ocean water, $\gamma_T = 10^{-4}$ is the thermal exchange velocity and
L=3.35 x $10^5$ J kg$^{-1}$ is the latent heat of fusion. The local freezing point is given (Beckmann
and Goosse, 2003) as
$$T_f = 0.0939 - 0.057 \cdot S_0 + 7.64 \times 10^{-4} z_b,$$
with a mean value of ocean salinity $S_0$=35 psu and the bottom of the ice shelf below sea level
$z_b$. A distinction is made between protected ice shelves (Ross and Ronne-Filchner) with a
melt factor of $F_{melt}$ = 1.6x10$^{-3}$m s$^{-1}$ and all other ice shelves with a melt factor of $F_{melt}$ =
7.4x10$^{-3}$m s$^{-1}$. The parameters are chosen to reproduce observed average melt rates (Depoorter
et al., 2013) under the Ross, Ronne-Filchner and Amery ice shelves for the pre-industrial
LOVECLIM ocean temperature and Bedmap2 (Fretwell et al., 2013) shelf geometry. For ice
shelves located inland from the fixed land-sea mask of the ocean model, mid-depth ocean
temperature from the nearest deep-ocean grid point in the same embayment is used for the
parameterisation.
In addition, surface melting of the Antarctic ice shelves has been taken into account,
compared to earlier model versions where all surface meltwater was assumed to refreeze at
the end of summer. The surface mass balance of ice sheet and ice shelf are now treated

consistently with the same positive-degree-day model including capillary water and refreezing terms. The same melting schemes for basal and surface melt have been used for the AIS model version that participated in the PlioMIP intercomparison exercise of de Boer et al. (2015).

The atmospheric interface for the GrIS was redesigned to enable ice sheet regrowth from a (semi-) deglaciated state given favourable conditions. This is accomplished by calculating surface temperatures independently for different surface types (ocean, ice sheet, tundra), which most importantly prevents tundra warming to affect proximal ice sheet margins. At the same time, the full range of atmospheric forcing is taken into account by allowing the ice sheet forcing temperature to exceed the melting point at the surface. This provides an in principle unbounded temperature anomaly forcing for increasing atmospheric heat content for the positive-degree-day melt scheme.

## 9  Acknowledgements

We acknowledge support through the Belgian Federal Science Policy Office within its Research Programme on Science for a Sustainable Development under contract SD/CS/06A (iCLIPS). Computational resources have been provided by the supercomputing facilities of the Université catholique de Louvain (CISM/UCL) and the Consortium des Equipements de Calcul Intensif en Fédération Wallonie Bruxelles (CECI) funded by the Fond de la Recherche Scientifique de Belgique (FRS-FNRS). We thank all four reviewers and the editor for constructive comments and their follow-up of the manuscript.

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

## 11  Tables

**Table 1. Overview of all discussed model experiments. The second column gives the scale factor R for temperature anomalies over the Greenland ice sheet.**

| Name | R | Description |
|---|---|---|
| Reference | 0.4 | Fully coupled reference simulation |
| High | 0.5 | Fully coupled simulation |
| Low | 0.3 | Fully coupled simulation |
| Forced reference | 0.4 | Forced with climate output from Reference |
| Forced high | 0.5 | Forced with climate output from Reference |
| Forced low | 0.3 | Forced with climate output from Reference |
| No sub-shelf melting | 0.4 | Suppressed Antarctic sub-shelf melting |

**Table 2. Peak sea-level contribution in sea-level equivalent (SLE) and timing from the Greenland ice sheet above present-day levels for three different parameter choices.**

| | Fully coupled experiments | | Forced repeat experiments | |
|---|---|---|---|---|
| Name | SLE (m) | time of peak (kyr BP) | SLE (m) | time of peak (kyr BP) |
| High | +2.72 | 122.8 | +2.01 | 123.6 |
| Reference | +1.42 | 123.3 | +1.42 | 123.3 |
| Low | +0.65 | 124.0 | +0.81 | 123.7 |

**12 Figures**

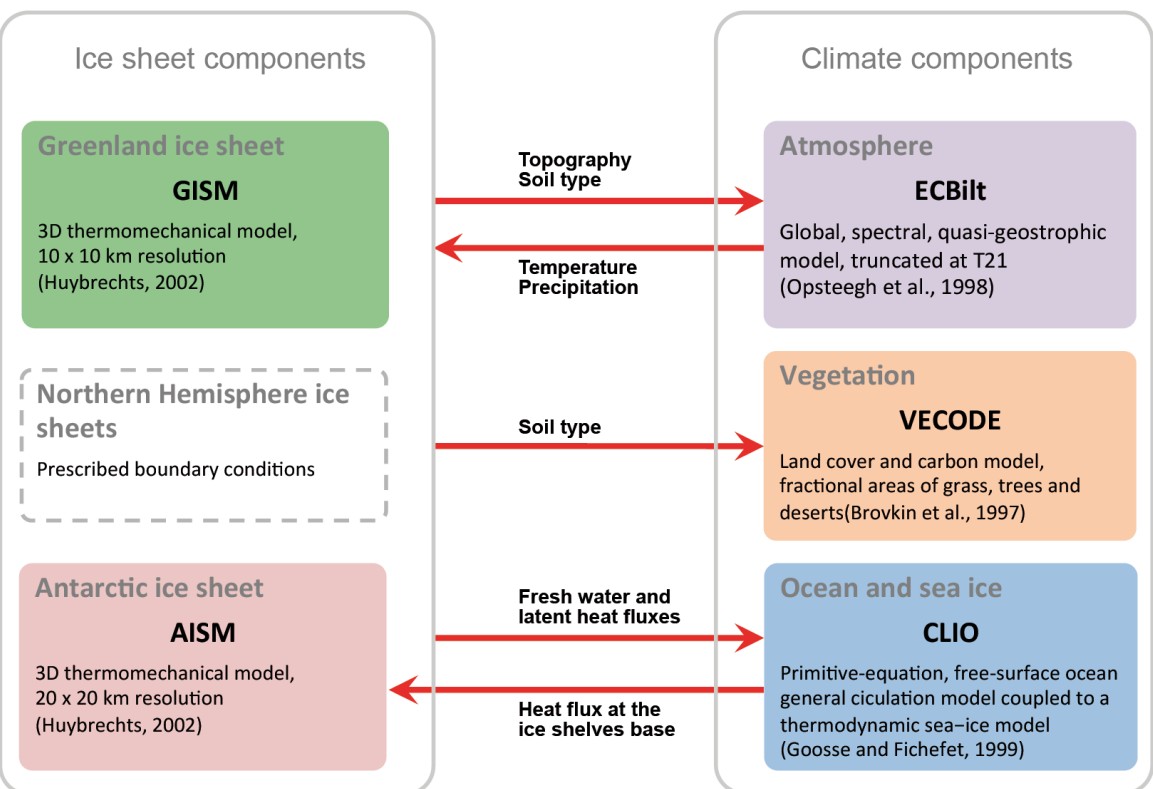


**Fig. 1. LOVECLIM model setup for the present study including dynamic components for the Greenland**
**and Antarctic ice sheets and prescribed Northern Hemisphere ice sheet boundary conditions.**


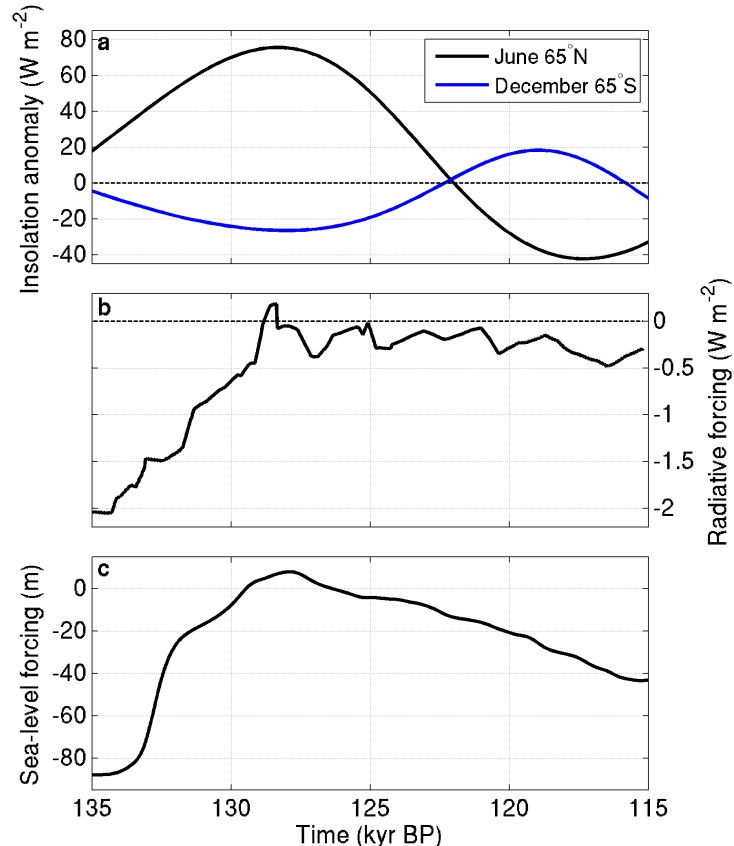


Fig. 2. Prescribed model forcing. Average monthly insolation anomaly (a) at 65° North in June (black) and
65° South in December (blue) to illustrate the spatially and temporally resolved forcing (Berger, 1978),
combined radiative forcing anomaly of prescribed greenhouse gas concentrations relative to the present
day (b) and sea-level forcing for the ice sheet components (c) derived from a Red Sea sea-level record
(Grant et al. 2012).




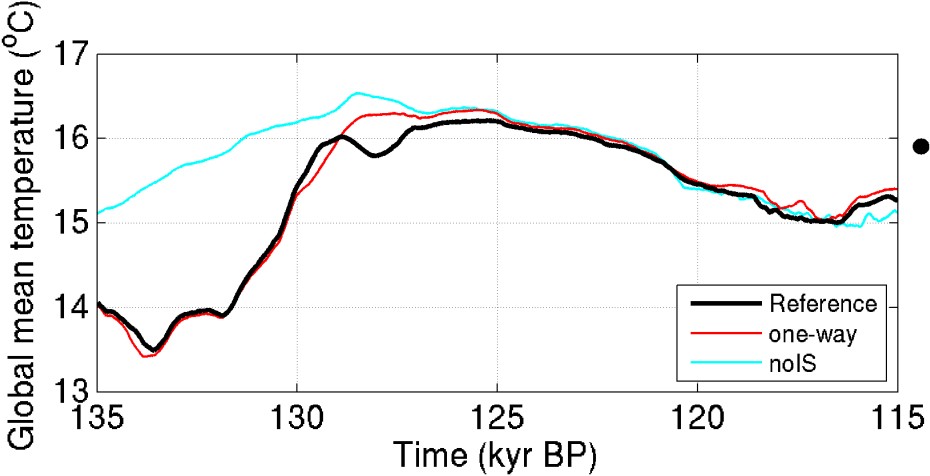


**Fig. 3. Global annual mean near-surface air temperature evolution of the reference run (black) compared**
**to experiments with prescribed Greenland and Antarctic ice sheet evolution from stand-alone experiments**
**(One-way, red) and no ice sheet changes at all (noIS, light blue). The filled circle on the right axis indicates**
**the temperature for a pre-industrial control experiment of the reference model with present-day ice sheet**
**configuration.**


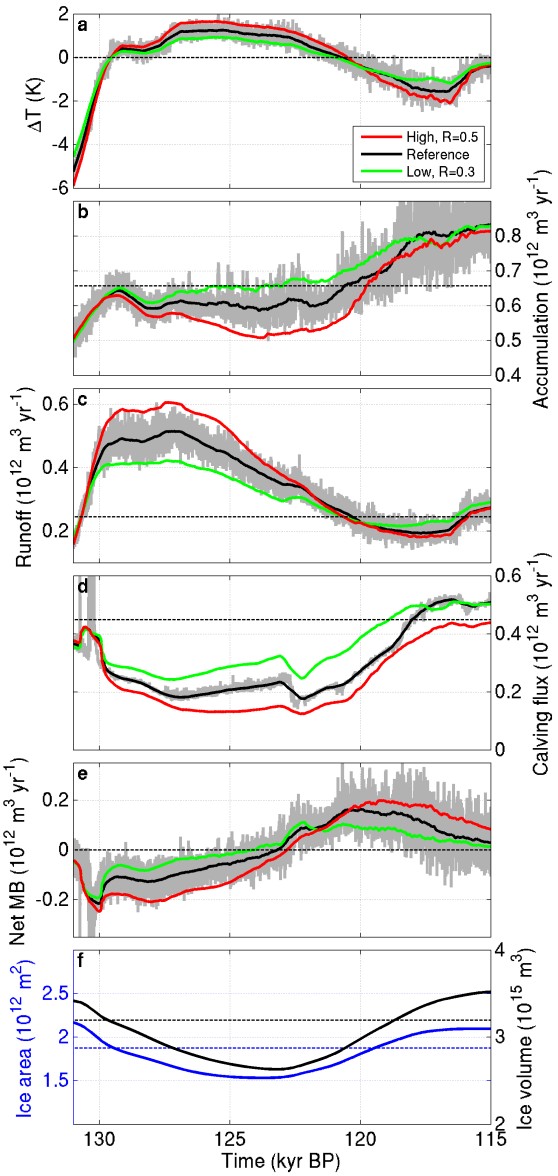

**Fig. 4. Greenland ice sheet forcing characteristics for the reference run (black) and with higher (red) and**
**lower (green) temperature scaling. Climatic temperature anomaly relative to pre-industrial (a).**
**Accumulation (b) and surface meltwater runoff (c) over grounded ice. Calving flux (d), net mass balance**
**(e) and other mass balance terms (b, c) given in water equivalent. Ice area (blue) and ice volume (black)**
**for the reference run (f). All lines are smoothed with a 400 years running mean except for the grey lines**
**giving the full annual time resolution for the reference run. Horizontal dashed lines give the pre-industrial**
**reference values, except for panel e, where it is the zero line.**

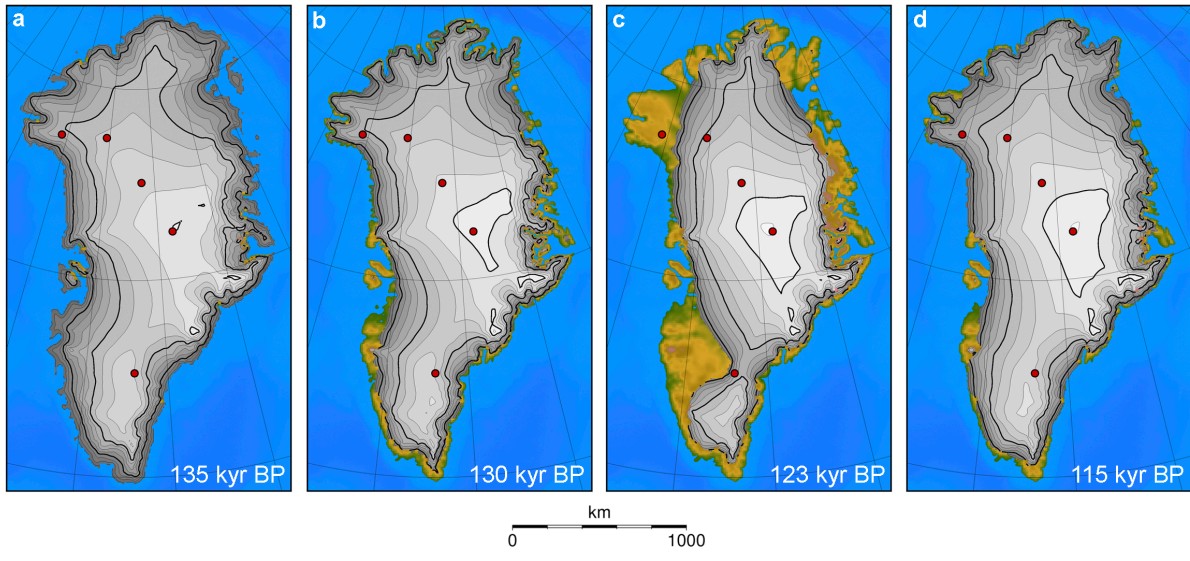

Fig. 5. Greenland ice sheet geometry at 135 kyr BP (a), 130 kyr BP (b), for the minimum ice sheet volume at 123 kyr BP with a sea-level contribution of 1.4 m (c) and at the end of the reference experiment at 115 kyr BP (d). The red dots indicate the deep ice core locations (from south to northwest: Dye-3, GRIP, NGRIP, NEEM, Camp Century).

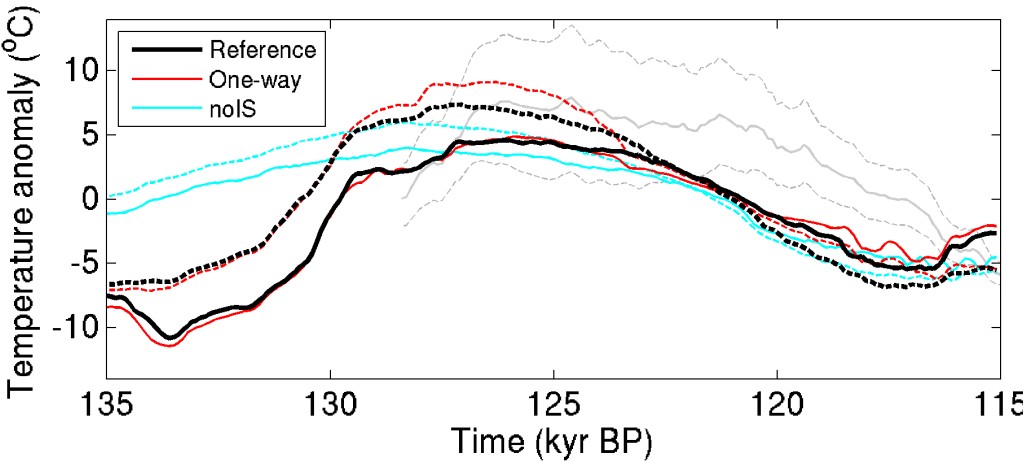


**Fig. 6. Comparison of modelled North-East Greenland annual mean (solid) and summer (June-July-August, dashed) surface temperature evolution (72° - 83° N and 306°33' - 317° 48' E) with reconstructed temperature changes (grey) at deep ice core site NEEM (77°27' N, 308°56' E). The solid grey line is the central estimate and grey dashed lines give the estimated error range for NEEM (NEEM community members, 2013).**


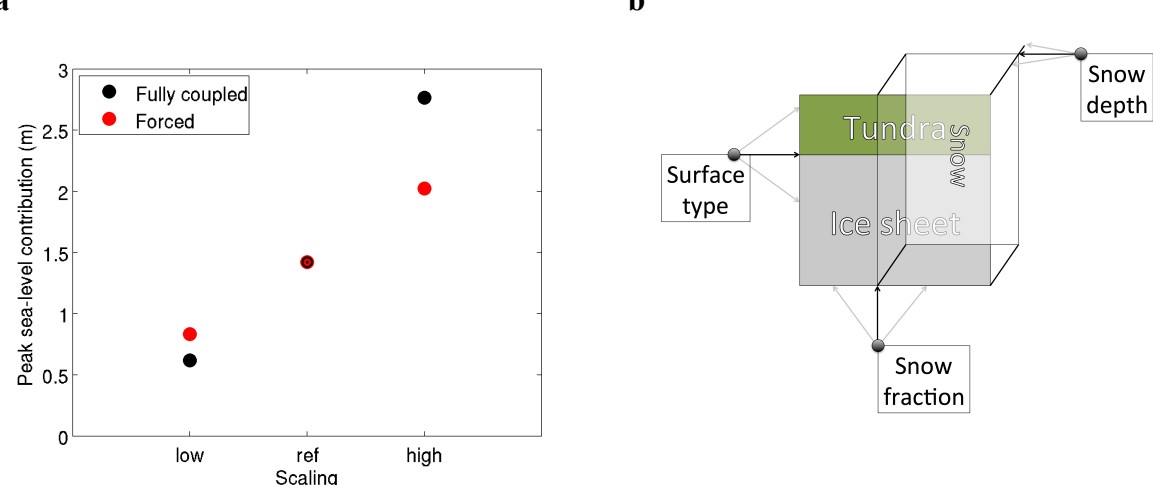

**Fig. 7. Scaling of sea-level contribution from the Greenland ice sheet as a function of temperature changes**
**for the full model (black) and forced model (red) in comparison (a). Schematic of the albedo**
**parameterisation in the land model for (partially) ice-covered areas (b), which is a function of the**
**underlying surface type, snow fraction and snow depth. See main text for details**

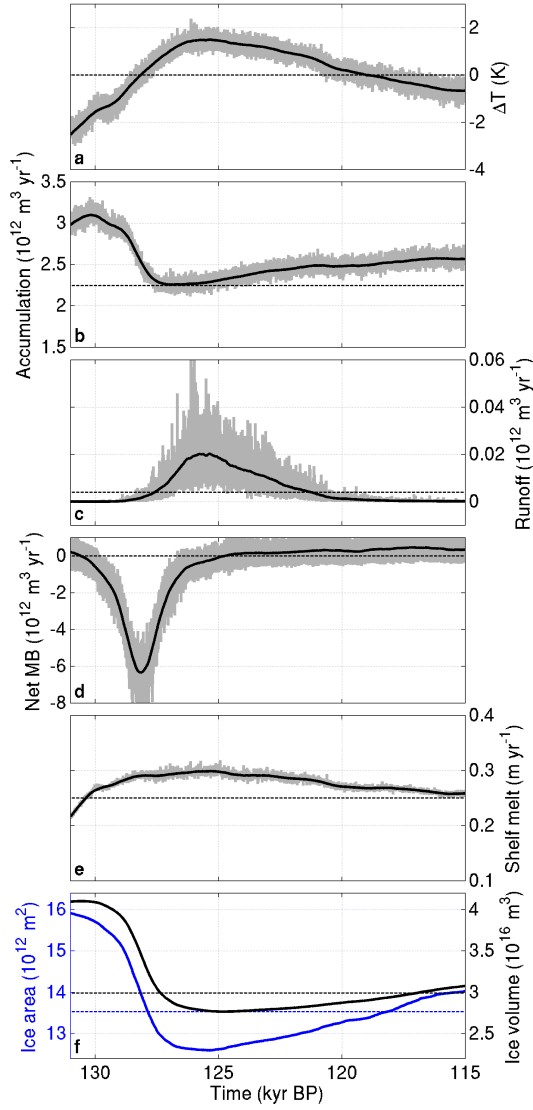

**Fig. 8. Antarctic ice sheet forcing and characteristics. Temperature anomaly relative to pre-industrial (a),**
**accumulation (b), surface meltwater runoff (c) and net mass balance of the grounded ice sheet (d), and**
**average sub-shelf melt rate diagnosed for the area of the present-day observed ice shelves (e). Mass**
**balance terms (b-e) are given in water equivalent. (f) Grounded ice sheet area (blue) and volume (black).**
**Grey lines give full annual time resolution, while black lines (and blue in f) are smoothed with a 400 years**
**running mean. Horizontal dashed lines give the pre-industrial reference values, except for panel d, where**
**it is the zero line.**

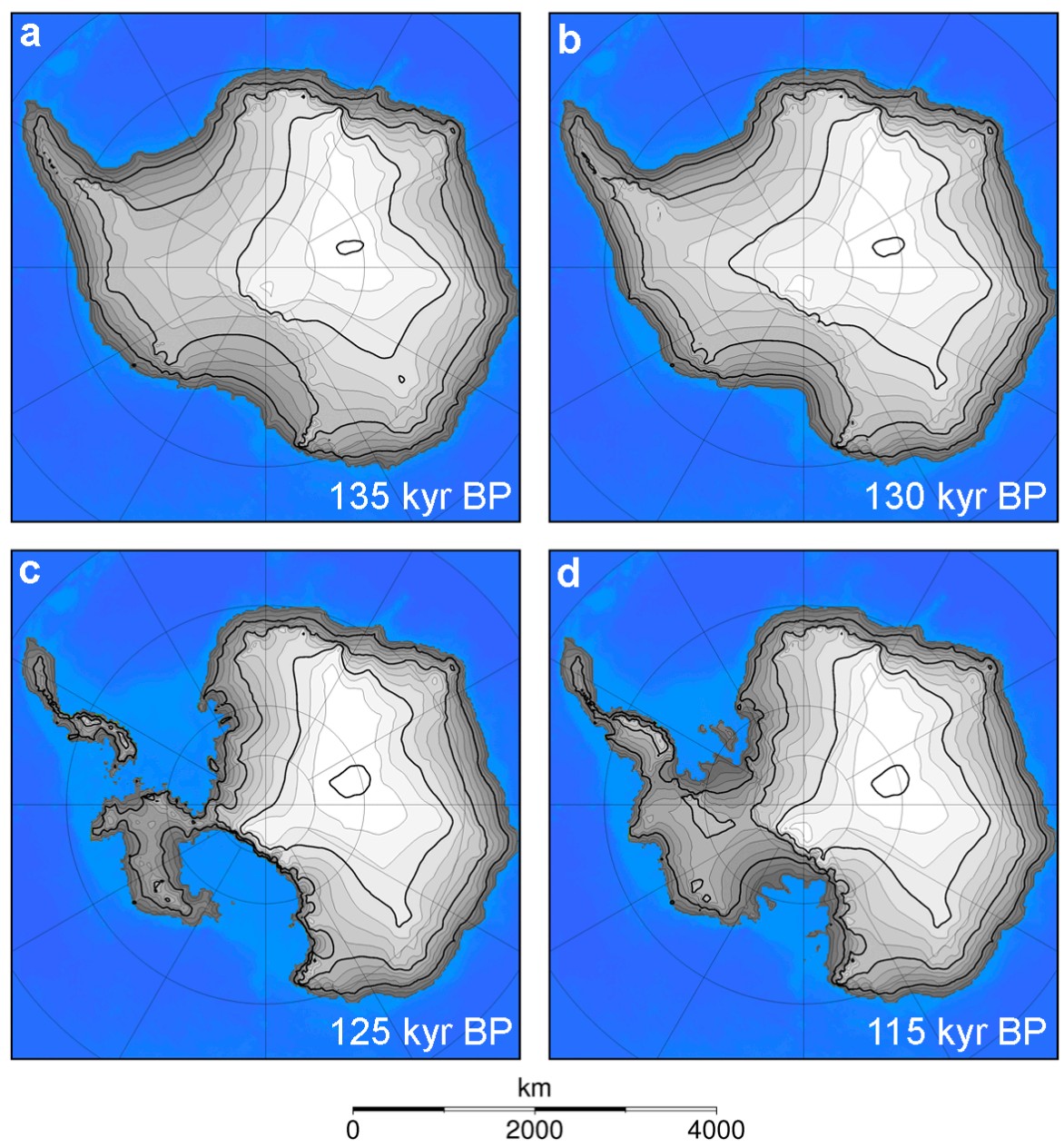

**Fig. 9. Antarctic grounded ice sheet geometry at 135 kyr BP (a), 130 kyr BP (b), for the minimum ice sheet**
**volume at 125 kyr BP with a sea-level contribution of 4.4 m (c) and at the end of the reference experiment**
**at 115 kyr BP (d).**

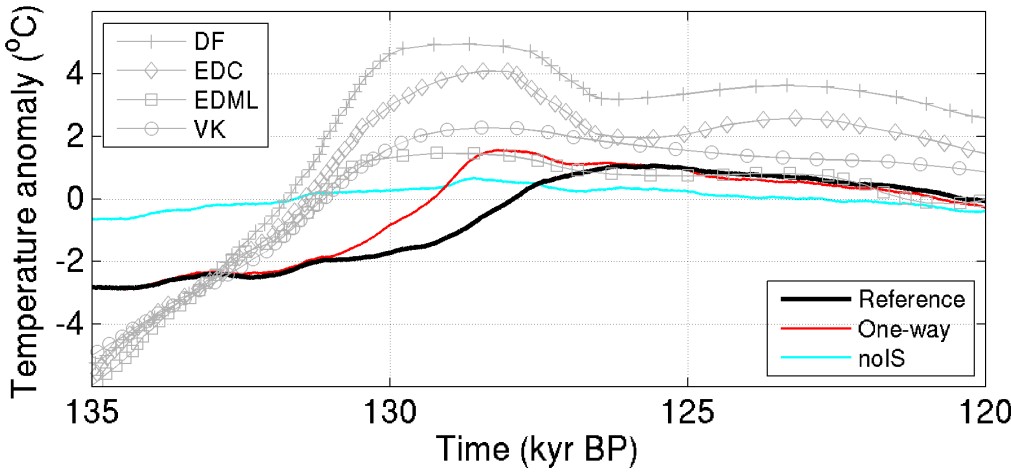


**Fig. 10. Comparison of modelled East Antarctic temperature evolution with reconstructed temperature**
**changes at deep ice core sites. Modelled temperature anomalies are averaged over a region 72° - 90° S and**
**0° - 150° E. Ice core temperature reconstructions for the sites EPICA Dronning Maud Land (EDML,**
**75°00′ S, 00°04′ E), Dome Fuji (DF, 77°19′ S, 39°40′ E), Vostok (VK, 78°28′ S, 106°48′ E) and EPICA**
**Dome C (EDC, 75°06′ S, 123°21′ E) are from Masson-Delmotte et al. (2011).**


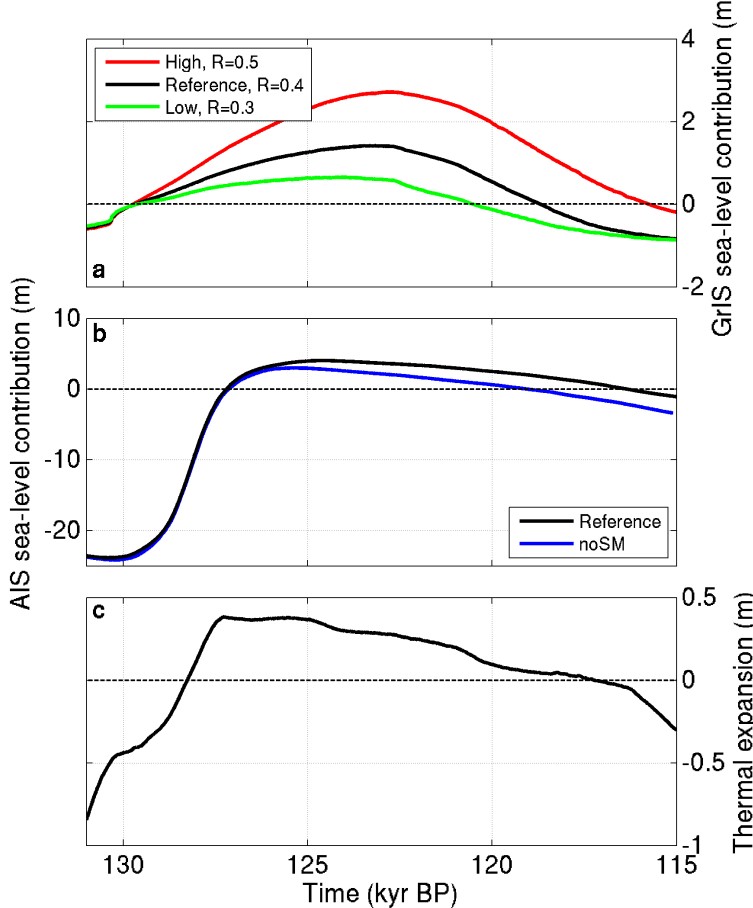


**Fig. 11. Sea-level contribution from the Greenland ice sheet for the reference run (black) and two**
**sensitivity experiments with higher (red) and lower (green) temperature scaling (a). Sea-level contribution**
**from the Antarctic ice sheet (b) from the reference run (black) and from a sensitivity experiment without**
**sub-shelf melting (blue). Sea-level contribution from oceanic thermal expansion from the reference run**
**(c).**


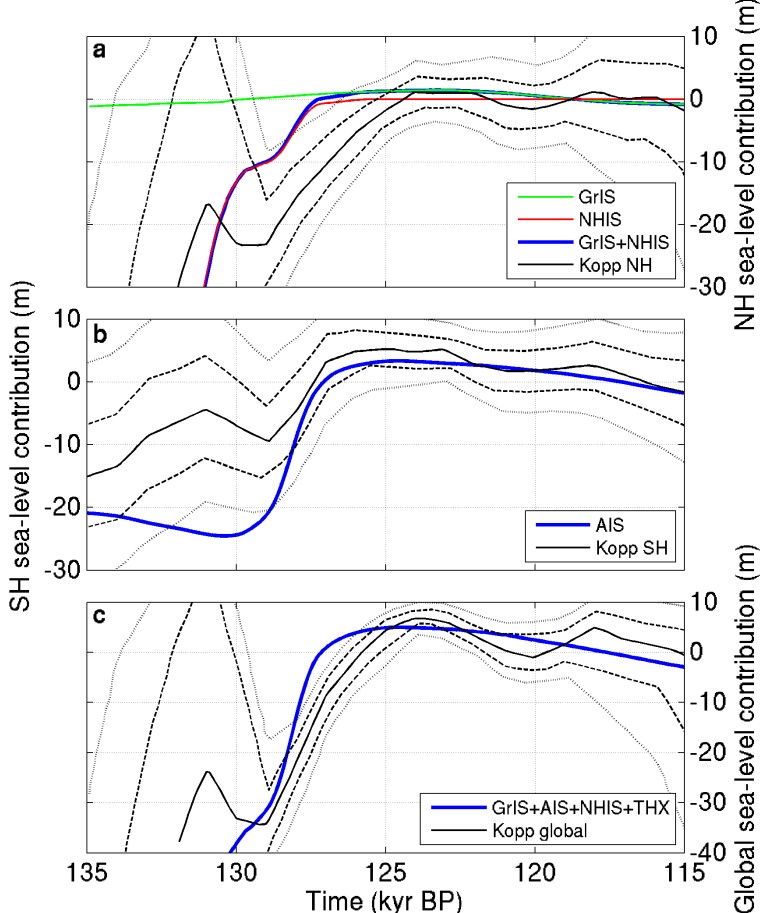


**Fig. 12. Modelled sea-level contributions from this study (colour lines) compared to probabilistic sea-level**
**reconstructions (black lines) from Kopp et al. (2009) for the NH (a) the SH (b) and global (c). For the**
**reconstructions, solid lines correspond to the median projection, dashed lines to the 16th and 84th**
**percentiles, and dotted lines to the 2.5th and 97.5th percentiles.**