# Peer review of "Heiko Goelzer1,a, Philippe Huybrechts1, Marie-France Loutre2, Thierry Fichefet2 4"

_Climate of the Past, 2015_

## Referee Comment (RC1) · Anonymous Referee #1 · 25 Jan 2016

**General comments:**

The submitted manuscript by Goelzer et al. investigates a new transient simulation of the last interglacial (LIG) period with a bi-directionally coupled climate-ice sheet model. More precisely, the authors use LOVECLIM1.3, an earth system model of intermediate complexity including interactive components for Greenland and Antarctica, i.e., the solely remaining ice sheets during the LIG. Consequently, the focus of the paper lies on climate and ice sheet changes in Greenland and Antarctica and the resulting sea level evolution throughout the LIG. The simulation is compared to previous experiments which exclude ice sheet changes or use a one-way coupling approach. Furthermore, they analyze different sensitivity experiments where specific climate processes are modified or omitted in the experimental setup. The main result of the paper is that the evolution of the Greenland ice sheet (GrIS) is dominated by changes in the surface mass balance whereas the Antarctic ice sheet (AIS) is mainly governed by melting of the shelf area driven by sea-level rise and reduced ice shelf viscosity in a warming climate. A valuable outcome of the model effort is also the temporal and spatial evolution of both the GrIS and the AIS generated within the same climate-ice sheet simulation and thus a consistent experimental setting.

The baseline of the paper is the remarkable technical effort to produce a fully-coupled climate-ice sheet model simulation for the LIG, i.e., a simulation which allows feedbacks between all components of the climate system and hence somewhat represents the "best possible estimate" of the LIG climate with a modeling approach. To my knowledge, the simulation is the first of its kind for the LIG and certainly a valuable contribution for the paleoclimate science community. However, in the present manuscript I am missing a comparison of the simulated climate with proxy records, at least for the two key regions Greenland and Antarctica, as this comparison would have the role of an evaluation of the novel model setup. Moreover I expect a more critical discussion of the chosen model approach regarding remaining improvements and challenges.

Concerning the formal aspects, I think the manuscript needs to be improved in several aspects. Whereas Sections 1-4 are mostly well-written, the results (Section 5) are sometimes hard to follow and need a revision to become more complete and comprehensive. Some figures are only partly described and very poorly referenced in the text (Table 1 is not mentioned a single time in the text). As you will see, I have many minor comments where I feel the language could be more precise to make the manuscript more reader-friendly.

Please find below the full list of major and minor issues.

**Major issues:**

**1. Critical discussion of experimental setup**
As stated above I expect a section which critically reflects on the quality of your model setup. I think as much as your reference simulation deserves credit for having a pioneering role as a fully-coupled simulation of the LIG it asks for a discussion of its strengths and weaknesses as well as of remaining challenges and possible improvements. This additional section could be in form of a "discussion" or an "outlook" section which both are non-existent at the moment. The discussion should also include a comparison to Helsen et al. 2013 CP, who previously assessed the GrIS retreat during the LIG with a bi-directionally coupled model approach.

Regarding modeling the climate in Greenland I further wonder if your setup includes the relevant feedbacks on temperature and precipitation found in response to a retreating Greenland ice sheet (Merz et al. 2014 CP, Merz et al, 2014 JGR, Hakuba et al. 2012 JGR). I suspect that the limited spatial resolution of the EMIC in the atmosphere (T21) might be a problem here. Furthermore, the authors should address the use of the positive-degree-day

method (PDD) for the ice sheets as this is a serious issue for the LIG as shown by van de Berg et. al. 2011 NatGeo.

**2. Scaling factor**

You use the scaling factor (described on lines 192-203) as a necessary tuning factor to avoid a complete loss of the Greenland ice sheet during the LIG. I wonder if the scaling factor is necessary due to the simplified representation of the climate in LOVECLIM over Greenland as I guess that the climate-ice sheet feedbacks previously mentioned in major issue 1 (described in Merz et al. 2014 CP, Merz et al, 2014 JGR, Hakuba et al. 2012 JGR) are probably not included.

How do you feel that this artificial control affects your result concerning Greenland ice sheet evolution and consequently its contribution to the LIG sea level?

Do I understand it correctly that no scaling factor is applied for the Antarctic ice sheet?

Line 274: I think you should state clearly here that the choice of the scaling factor crucially affects the contribution of the GrIS to the sea-level high stand of the LIG.

**3. Additional part describing all (sensitivity) experiments**

Currently, the manuscript presents results from various sensitivity experiments at different occasions, which makes it hard for the reader to keep the overview. Therefore, it would be much more reader-friendly to add a subsection to Section 4 describing all (sensitivity) experiments and their purpose. I think this subsection could be complemented with a respective list in a table.

I further advise to clearly state in the text that you define the two-way coupled simulation as "reference". Similar definitions might be worth for the stand-alone experiments etc. Make sure that you use these terms consistently in all text and figures.

**4. Extended analysis/description of results**

I think the manuscript would greatly profit from an extended analysis and some additional figures in order to present a complete picture of your two-way coupled simulation rather than just showing selected aspects.

Specifically I request:

As I like Figure 3 showing the gained value of the two-way coupling, I think a similar figure for temperature in Greenland and Antarctica would be highly appreciated as these two regions are the main areas of interest in your paper.

Moreover, I am missing a clear statement in the text regarding the results shown in Fig. 3: (i) the simulation with two-way coupling only marginally differentiates from the simulation with one-way coupling with respect to global mean temperature throughout the LIG. (ii) Excluding ice sheet changes and freshwater forcing as done in the noIS simulation leads to a decreased glacial-interglacial temperature contrast and an earlier warming going into the LIG. However, there is only a small difference to the one-way and reference simulation after ca. 128ka! I wonder whether the latter result also applies for temperatures in Greenland and Antarctica?

In order to evaluate your two-way simulation against data I strongly suggest a comparison of the simulated Greenland and Antarctic temperature evolution with respective ice cores (e.g., NEEM, EPICA). As the NEEM delta18O-based temperature reconstruction likely assumes an

overestimated delta18O-temperature relationship you could also include the NEEM temperature curve based on the recent delta18O-temperature relationship presented in Masson-Delmotte et. al. 2015 Cryosphere.

It might also be worth to show the evolution of the freshwater fluxes throughout the LIG to complement your findings presented at Line 353pp and Line 429pp.

**5. Mass balance of Greenland and Antarctic ice sheet**
I think a proper definition of the (surface) mass balances for the Greenland and the Antarctic ice sheet is required. Please clearly state what you refer to as accumulation, ablation, runoff, (surface) melting, calving flux and how they combine to the mass balance. Please use the same terms in the text as in the axis labels of Figs. 4 & 7.

I think it would be a valuable addition to show the net mass balance as a further panel in Figs. 4 & 7 so the reader can reconstruct the evolution of the ice volume shown in Figs. 4e and 7e. Whenever possible use the same scales for the different terms of the mass balance in Figs. 4 and 7.

**6. References to figures in text**
Throughout the paper I miss many references to the corresponding figures, which would substantially help the reader to understand the descriptions in text-form. Please be more precise when discussing panel plots, e.g. put reference to Fig. 4a rather than just to Fig. 4.

Some examples of missing/imprecise figure references:

Line 365: Fig. 7b
Line 375: Fig. 7a,d
Line 382: Fig. 7d
Line 393: Fig. 9b after "experiments"
Line 402: Fig. 7a and 2c
Line 427: Fig. 3 after "evolution"
Line 445: Fig. 10c
Line 446: Fig. 10b

**Minor issues:**

Lines 23: Please be more specific than "surface mass balance changes"

Lines 24-25: "Our results indicate" could be replaced with "The comparison of fully-coupled with stand-alone Greenland ice sheet simulations emphasizes"

Line 68: change "lower bound of 5.5m" to "lower bound of Eemian sea level rise of 5.5 m above present-day levels"

Lines 75-78: This sentence is misleading as it implies that any Southern Ocean warming is induced by the interhemispheric seesaw effect.

Line 104: Wrong reference: Robinson et al. 2011's ice sheet model uses output of a transient EMIC simulation as input but does not give feedback to the climate model. Helsen et al. 2013 CP would be a more appropriate reference here.

Line 111: "climate and oceanic conditions over the ice sheets and in their proximity" seems not to be a correct/precise statement.

Lines 109-118: Whereas I like the rest of the introduction, this last paragraph should be improved to better stress the focus and strategy of the paper. I think you should highlight here that you generate the first transient simulation of the LIG with a bi-directional coupling of climate and GrIS/AIS model components. Furthermore, please clarify that you study key mechanisms and feedback processes with the aid of sensitivity experiments and with the comparison to one-way coupled and stand-alone ice sheet simulations. I would also state here that you focus on climate and ice sheet changes in Greenland and Antarctica and the resulting sea-level evolution throughout the LIG.

Lines 170-171: specify "ice loading changes" e.g., with "ice loading changes coming out of the penultimate glacial period".

Line 184-191: You use the sea-level reconstruction by Grant et al. 2012 as boundary condition for your simulations. Wouldn't it be more consistent to use the "internal" sea level corresponding to the simulated global ice sheet changes? What would be the consequences for the melting of the AIS which apparently most strongly responds to sea level changes? What are the reasons for driving the model with a respective sea-level reconstruction instead?

Line 205: introduce the abbreviation "(SA)" here.

Line 215: The title of this subsection could be more specific, e.g., "Initialisation of the reference simulation"

Line 241: might be more precise to replace "importance of ice sheet changes" with "importance of two-way coupling between the climate model and the ice sheet models for the GrIS and the AIS"

Line 247-251: This finding is somewhat hard to understand. May be it would help if you show the freshwater fluxes in a figure (as also requested in major issue 4) and put a respective reference.

Line 264: Rather put the reference to Fig. 4e here.

Lines 334-336: I think you could add here that the ice-albedo feedback is a positive feedback.

Line 351pp: "The warming necessary…." This sentence is not easy to comprehend. Please revise.

Line 365: show the freshwater fluxes in a figure or add (not shown) after "hemispheres".

Line 367: You speak of "ablation" but in Figure 7 you name it runoff – is this the same? Please be consistent with all terms describing the mass balance of the GrIS and the AIS (see also major comment 5)

Line 376: add (not shown) after "130 kyr BP".

Line 380: add (not shown) at end of sentence or put a reference to Goelzer et. al. 2015.

Line 418-419: Is "their model" equal to the simulation you termed "one-way" at other occasions in the script?

Line 470-472: I think this sentence should be rephrased to state more clearly that you artificially limit the melting of the GrIS to conform to existing ice core constraints.

Line 477-478: Please be more specific. I think "ice-climate feedback" is a too general term for a take-home message in the conclusions.

Line 482: I think it should also be stated here that an unconstrained fully-coupled climate-ice sheet simulation does not fully agree with data, e.g., the GrIS would melt away completely during the LIG. This implies deficiencies in the model physics or unknown/excluded processes. It also emphasizes the NEEM paradox of strong warming coinciding with limited GrIS melting that can hardly be understood in a model perspective.

Table 1: Needs to be discussed in the text or should be removed.

Figure 1: The references (Opsteegh et al. 1998, Brovkin et al. 1997 and Goosse and Fichefet, 1999) mentioned in Fig. 1 should also be added to the reference list.

Fig. 4b,c,d: Does the horizontal stippled line represents the pre-industrial level? Please clarify in figure caption.

Fig. 6b: This schematic is somewhat difficult to comprehend and it is only mentioned once in the text. Should be revised or removed.

Figure 7d and text: Is there a difference between shelf melting and sub-shelf melting? Please be consistent in text and figures

Figure 9b: Does the blue curve represent the experiment with excluded surface AND sub-shelf melting or just the latter? In line 392 you mention both. Please revise to be consistent in text and figures.

Figure 10: please number the panels with a,b,c. Furthermore, the figure caption should include additional information, e.g., the meaning of the stippled lines.

**Technical corrections:**

Line 89: remain**s**

Line 96: van **d**e Berg

Line 256: "is retreating" rather than "has retreated"

Line 379: **a** weakening

Caption of Fig. 8: move listing of (a), (b), (c) in front of description as done in all other figure captions.

---

## Editor Comment (EC1) · D.-D. Rousseau (Editor) · 1 Feb 2016

Dear Authors,

Reviewer 1 posted a very detailed report about your manuscript. As Climate of the Past offers the opportunity for its authors to exchange directly with both their paper reviewers but also the community, I strongly encourage you to interact with reviewer 1 by taking the offered opportunity to reply or comment this review. All the very best and thanking you for considering Climate of the Past for publishing your results.

All the very best

denis-didier Rousseau Clim. Past co-editor in chief
* * *

---

## Referee Comment (RC2) · Anonymous Referee #2 · 16 Feb 2016

This study assesses the Last Interglacial climate and ice sheet evolution in a two-way coupled approach. The novelty is in the fully coupled method. Especially promising is the simulated evolution of both the Greenland and the Antarctic ice sheet in one overarching climate-ice sheet framework, which allows for assessing their relative contributions to the global mean sea-level highstand during the Last Interglacial. As such the study is interesting as should be published. However, some parts are unclear and lack information and/or discussion.
Please discuss the comments below before publication in CP.

**GENERAL COMMENTS**

1) Sea-level forcing from a Red Sea record is prescribed. Are the simulated sea-level changes from the Greenland and Antarctic ice sheet somehow added to this during the simulation? How certain is the Red Sea record? And how much does it affect the sea-level contributions of the two ice sheets and the total sea-level changes simulated? The discussion on this (lines 420-424) is too short.

2) Related to this: Would it be possible to fit your model results better to the Kopp et al. (2009) reconstructions if uncertainties in the Red Sea sea-level record are included, or if you use the benthic d18O-stack? In other words can you suggest improvements to the NH ice sheet retreat records, based on the comparison between your simulations and the Kopp reconstructions?

3) Why is the temperature forcing over Greenland so high that it melts away the Greenland ice sheet entirely? What are the summer and annual mean temperature anomalies for the Last Interglacial? Please compare and discuss this with respect to proxy data, and previous climate model simulations (see e.g. Bakker et al. (2013) and Lunt et al. (2013) for global intercomparisons). The method of uniform scaling is a bit eccentric, and needs better argumentation.

4) Related to this: the experimental set-up misses a section that describes how the simulated temperatures (and accumulation) are converted to (surface) mass balance. Which scheme do you use? With which parameter settings? The latest studies simulating the Last Interglacial Greenland ice sheet evolution show that differences in parameter settings have a huge effect on how much the ice sheet melts (e.g. Robinson et al., 2011; Stone et al., 2013; Langebroek and Nisancioglu, 2016).

5) These studies validate their ice sheet model results to the present-day observed ice sheets. I think this is what you need to do as well. Compare your present-day or pre-industrial climate and ice sheet configuration to observations and discuss the differences. This will validate the model set-up, and increase confidence in your model results.
How do you deal with the differences between the atmospheric and ice sheet model grids?

6) Also for Antarctica some discussion is lacking:

a. Lines 375-380: Can you show model "evidence" for the see-saw effect taking place in your model results? E.g. assess Atlantic meridional ocean circulation or heat transport. Do they really decrease?

b. Lines 381-390: What do you mean with "overshoot behaviour"? Is the Antarctic ice loss not related to the positive temperature anomaly? Which part is overshoot?

c. Also, how does the present-day/pre-industrial simulated Antarctic ice sheet look like? Is this not too sensitive to the temperature forcing, as is the case for Greenland? So in other words, no correction is needed for the temperature forcing over Antarctica?

d. Lines 391-402: these sensitivity experiments need more explanation, and a reference to Figure 9b.

7) The Section about freshwater input and thermal expansion of the ocean is very interesting, but also lacking information. How large is the freshwater input (Sv) and how long do the episodes take? Another figure or table would be useful.

8) Concerning the "double" peak in the Kopp reconstruction: Do you have suggestions why your model results do not reproduce this? Is it because of too constant the climate forcing, too slow regrowth of the ice sheets, or other missing feedbacks? Please discuss.

**SPECIFIC AND TECHNICAL COMMENTS**

1) Greenland and Antarctic ice sheets are abbreviated in line 59, please use these abbreviations in the remainder of the text

2) A bit more information on the coupling procedure is necessary (Section 4.2). How often do they interact or are the components updated, every day/year/1000 years?

3) Lines 275-293: You can also use the reconstructed limits for the Last Interglacial surface elevation change at the ice core locations compared to PI (e.g. NGRIP-members, 2004, Johnsen and Vinther, 2007, NEEM community members, 2013) to evaluate your model results.

4) Lines 294-305: I don't understand the need of such a speculative section. What is the surface mass balance evolution over the Greenland ice sheet? The resulting ice volume changes are shown in Fig. 4.

5) Lines 325-346: This section is difficult to read. It would be better to better explain the sensitivity experiments. Is "forced" the same a "stand-alone" as you call it earlier in the text? Better also to discuss the simulated maximum sea-level contribution in two steps: 1) effect of temperature scaling factor on resulting ice volume changes, 2) effect of coupling ("forced/stand-one" vs "coupled") on ice volume change.

6) Lines 360-365: comparing the Last Interglacial accumulation to pre-industrial is a bit difficult if you base the calculation on differently sized areas. Maybe the accumulation actually didn't increase in many locations? What happens over NEEM? Maps for certain time slices would be much more helpful.

Line 24: "reference experiment", either describe the reference experiment, or omit the mentioning of this and change the values to express the full range of your results (0.62-2.77m)

Lines 32-33: would be nice to add which part of the ~5m is due melting of the Greenland and which due to the Antarctic ice sheet

Line 63: skip "e.g."

Line 71: "mean" instead of "central"

Line 77: add "possibly" caused by

Lines 84-86: make new section, and add "evidence" for possible reduction of the LIG AIS

Line 87: better constrained than ...? (I assume AIS evolution)

Line 102: also mention latest work (Langebroek and Nisancioglu, 2016)

Line 99: correct reference is Born and Nisancioglu, 2012; please also update in rest of text

Line 104: incorrect reference, maybe you meant regional climate model, or a different reference

Line 106: reformulate "results" – what results?

Line 108: check correct reference in reference list for Pollard and DeConto, 2009 or 2015?

Lines 113-114: skip "high-resolution", grid boxes of 10 or 20 km is normal, not high for ice sheet models

Line 121: EMIC description with capital letters or not – make consistent with abstract

Lines 123-124: "The model has been utilised ..." – but without dynamic ice sheets, and two-way coupling, right? Rewrite to make clear.

Lines 133-134: what is the resolution of T21 in degrees or km, approximately? "high-resolution ice sheet models", see earlier comment

Lines 137-138: are the freshwater fluxes etc the same as in the earlier version of the model, or is the set-up the same? Please rewrite.

Section 3.1: Would make more sense to make Section 3.1 a part of 3.2

Line 157: change to "sea-level equivalents (SLE)"

Lines 158-160: sentence very unclear, please rewrite

Lines 181-183: Is insolation calculated for each latitude and for each month? Not entirely clear, especially because figure only shows 2 months and 2 latitudes.

Line 186: change "the latter" to "this data"
Would be nice to explain what this reconstruction is based on.

Line 193: Skip "As a measure"

Line 208: skip "comparison between"

Line 209: skip "recorded"

Lines 208-210: The ice sheet response to what?

Line 211: Are these "Additional experiments" stand-alone experiments or coupled?

Lines 217-219: What is the climate forcing for this initialisation? And how large are the 'initial' Greenland and Antarctic ice sheets, so at 135ka?

Line 231: The first section of the Result should be named "5.1 Climate evolution" or something similar

Lines 231-235: and what are the differences to Loutre et al., 2014?

Line 249: Southern Ocean (SO**)**

Lines 250-251: I don't see this cooling event in the one-way experiment, please rewrite.

Line 254: change to "mass balance dominated by ablation"

Section 5.1: What do you call "ablation"? runoff + calving or only runoff? Need for some definitions here.

Lines 254-255: "Marginal" could mean "just a bit" or "on the rim", please clarify.

Section 5.1: "Temperatures", are these summer mean or annual mean? Surface or air temperatures? Please be more precise.

Figure 4: are the dashed lines the pre-industrial values? Would be great to have these numbers also for the ice area and volume.

Line 268: change "furthest" to "maximum"

Line 269: change "Conversely" to "At the same time"

Line 317: Not sure if Merz et al., 2014 is the correct reference here, as they focus on the effect of topography on precipitation during the Last Interglacial.

Line 334: "Figure 6, left" should be "Figure 6a", check also rest of section.

Line 340: skip "therefore"

Line 365, "Figure 7**b**"

Line 367: so ablation is runoff?

Figure 7: what is the present-day ice area and volume in your model set-up?

Line 375: include reference to Figure 7d

Line 414: "included" instead of "attempted"

Line 428: "Ocean expansion is **rapid** during …"

Line 439: skip "well"

Lines 439-440: the estimated LIG ocean thermal expansion is 0.4±0.3m according to the IPCC report, they use McKay et al., 2011 as a reference. Please rewrite.

Line 443: "AIS **and** thermal expansion"

Lines 443-445: add reference to Figure 10

Figure 10: add information on confidence levels to figure caption

Line 453: change "hiatus" to "regrowth" or similar

**REFERENCES**

Bakker, P., Stone, E. J., Charbit, S., Gröger, M., Krebs-Kanzow, U., Ritz, S. P., Varma, V., Khon, V., Lunt, D. J., Mikolajewicz, U., Prange, M., Renssen, H., Schneider, B., and Schulz, M.: Last interglacial temperature evolution – a model inter-comparison, Clim. Past, 9, 605–619, doi:10.5194/cp-9-605-2013, 2013.

Born, A., and Nisancioglu, K. H.: Melting of Northern Greenland during the last interglaciation, Cryosphere, 6, 1239-1250, doi:10.5194/tc-6-1239-2012, 2012.

Johnsen, S. J. and Vinther, B. M.: Greenland stable isotopes, in: Encyclopedia of Quaternary Science, edited by Elias, S., vol. 2, pp. 1250–1258, Elsevier, 2007.

Kopp, R. E., Simons, F. J., Mitrovica, J. X., Maloof, A. C., and Oppenheimer, M.: Probabilistic assessment of sea level during the last interglacial stage, Nature, 462, 863-867, 628 doi:10.1038/nature08686, 2009.

Langebroek, P. M. and Nisancioglu, K. H.: Moderate Greenland ice sheet melt during the last interglacial constrained by present-day observations and paleo ice core reconstructions, The Cryosphere Discuss., doi:10.5194/tc-2016-15, in review, 2016.

Loutre, M. F., Fichefet, T., Goosse, H., Huybrechts, P., Goelzer, H., and Capron, E.: Factors controlling the last interglacial climate as simulated by LOVECLIM1.3, Clim. Past., 10, 1541-1565, doi:10.5194/cp-10-1541-2014, 2014.

Lunt, D. J., Abe-Ouchi, A., Bakker, P., Berger, A., Braconnot, P., Charbit, S., Fischer, N., Herold, N., Jungclaus, J. H., Khon, V. C., Krebs-Kanzow, U., Langebroek, P. M., Lohmann, G., Nisancioglu, K. H., Otto-Bliesner, B. L., Park, W., Pfeiffer, M., Phipps, S. J., Prange, M., Rachmayani, R., Renssen, H., Rosen- bloom, N., Schneider, B., Stone, E. J., Takahashi, K., Wei, W., Yin, Q., and Zhang, Z. S.: A multi-model assessment of last in- terglacial temperatures, Clim. Past, 9, 699–717, doi:10.5194/cp- 9-699-2013, 2013.

McKay, N. P., J. T. Overpeck, and B. L. Otto-Bliesner, 2011: The role of ocean thermal expansion in Last Interglacial sea level rise. *Geophys. Res. Lett.*, **38**, L14605.

NEEM community members: Eemian interglacial reconstructed from a Greenland folded ice core, Nature, 493, 489–494, 2013.

NGRIP-members: High-resolution record of Northern Hemisphere climate extending into the last interglacial period, Nature, 431, 147–151, 2004.

Robinson, A., Calov, R., and Ganopolski, A.: Greenland ice sheet model parameters constrained using simulations of the Eemian Interglacial, Clim. Past., 7, 381-396, doi:10.5194/cp-7-381-2011, 2011.

Stone, E. J., Lunt, D. J., Annan, J. D., and Hargreaves, J. C.: Quantification of the Greenland ice sheet contribution to Last Interglacial sea level rise, Clim. Past., 9, 621-639, doi:10.5194/cp-9-621-2013, 2013.

---

## Referee Comment (RC3) · A. Ganopolski (Referee) · 16 Feb 2016

A. Ganopolski (Referee)

andrey@pik-potsdam.de

The manuscript by Goelzer et al. presents results of the first fully interactive simulation of climate and ice sheet evolution during the penultimate glacial termination and the last interglacial (LIG) using an Earth system model of intermediate complexity. The authors show that reconstructed temporal dynamics of sea level during the LIG can be successfully reproduced by their model. The authors for the first time demonstrated that disintegration of the last fraction of the West Antarctic ice sheet (WAIS) at the beginning of LIG can be solely explained by the dynamical response of the ice sheet to sea level rise. The manuscript presents in depth analysis of the processes and feedbacks operating in the system supported by a set of sensitivity experiments. The manuscript is well-written and properly illustrated. I believe this is an important scientific contribution and I would recommend it for publication in CP after minor revision.
[Figure]

General comments

1. Although the manuscript by Goelzer et al. is not the first paper produced in the framework of the same project and many technical details have been already described in Loutre et al (2014) and Goelzer et al (2015), for the readers' convenience a more detailed description of experimental design would be helpful. In particular I would suggest (i) provide information of how surface mass balance of ice sheets was simulated and give in the table the values of semi-empirical parameters; (ii) explain how temperature and precipitation anomalies from low-resolution climate component were applied to high resolution ice sheet models and how changes in ice sheet elevation and extent were accounted for; (iii) how simulated ocean temperature anomalies were used to compute submarine melt of ice shelves; (iv) how one-way coupling experiments have been performed; (v) how "present" GrIS and AIS have been simulated.

2. I have a question concerning scaling technique to reconstruct Northern Hemisphere (NH) continental ice sheets during penultimate termination. According to the manuscript, evolution of NH ice sheets were prescribed using Lisiecki and Raymo (2005) benthic stack L&R04 and the Fig. 4 from Goelzer et al. (2015) shows that according to L&R04 the termination was only half-way at 130 ka with the global sea level still ca. 50 m below present. This would imply existence of large continental ice sheets in the NH which is consistent with the Fig. 2 from Goelzer et al. (2015). However, according to the Figure 10 (top) from the new manuscript, the volume of NH ice sheets at 130 ka was only 10 meters in sea level equivalent which is only 10% of their LGM value. If I misunderstood your approach, please clarify.

3. To prevent GrIS from complete melt, the authors scaled down simulated temperature anomalies used for calculation GrIS surface mass balance. This is somewhat surprising in a view that simulated glacial-interglacial global temperature change in the model is only about 2C which is much less than results of PMIP2 and 3 models which simulated global LGM cooling of 4-5C. Moreover, uncorrected simulated GrIS temperature anomalies during LIG are only about 3C which is still well below "NEEM temperature

reconstructions". It would be useful to show simulated summer temperature anomalies over the GrIS because summer temperatures are the most important for ice sheet mass balance.

4. While I have no problem with the pragmatic decision to scale GrIS temperature anomalies down, I am missing an explanation why the authors decided to use the factor 0.4 as the reference value and considered 0.3 and 0.5 as the upper and lower limits. I wonder whether simulation for scaling factor 0.4 is better than for other two, can the value 0.5 can be accepted or rejected by empirical constraints and whether any larger scaling factors can (or cannot) be ruled out? I believe that at present the only thing we can say with some confidence about GrIS during LIG is that melting of more than half of modern GrIS would be difficult to reconcile with the existing empirical constraints. Any number below 3 meters is equally probable and therefore implied accuracy of reported "1.4 m" significantly underestimates uncertainties of this estimate. I also found it noteworthy that three numbers for the range of GrIS contribution during LIG ( 0.6, 1.4, 2.8 m) given by the authors are almost identical to the values given in the recent paper by Calov et al. (2015, CP): 0.6, 1.4, 2.5 m.

5. While the estimates of GrIS contribution fall well within the range reported in a number of previous studies, dynamical collapse of the WAIS during LIG is new and very important finding presented in the manuscript. Thereby it would be interesting to learn more about the mechanisms. The authors show that Antarctic ice volume overshoot is not related to enhanced surface or subsurface melting, as was proposed in some previous studies, but mostly of dynamical WAIS response to prescribed global sea level rise. In this relation I have a question. What is the crucial difference between the penultimate and the last glaciations which explains this overshoot: much faster sea level rise during the penultimate glaciation or the fact that sea level from Grant et al. (2012) overshoots Holocene sea level by ca. 10 m already at the beginning of LIG? The authors mentioned that they performed similar simulations with the L&R04 sea level reconstruction. Since L&R04 stack suggests a slower rate of sea level rise and

does not overshoot present sea level during LIG, I wonder what is the WAIS dynamics in this experiment.

6. Although the mechanism for the WAIS disintegration found in the study by Goelzer et al. differs from that proposed by Holden et al. (2010), I do not believe that the modeling results presented in the manuscript under consideration can be used to rule out completely importance of submarine melt for stability of the WAIS. The reason is that simulated in the current study bipolar see-saw is very weak compare to other modeling results and paleoclimate data. The later reveal significant temperature overshoots at the beginning of LIG essentially everywhere in the SH, and the magnitude of temperature overshoots (above present) in different Antarctic locations was at least several degrees. At the same time, in the work by Goelzer et al. (2015) only a tiny (0.2C) temperature overshoot is seen in subsurface South Ocean temperature (Fig 7b) and essentially nothing in SH or Antarctic temperatures. This seems to be a typical feature of the LOVECLIM model (e.g. Menviel et al., 2015, EPSL). I believe, this potential caveat of the current study should be mentioned in the discussion.

Specific comments

L 82 It should be Pollard et al. (2015)

L 182 What is the meaning of "dynamically computed"?

L 183 Does "governing" means here "major"?

L 187 "... assumes ice volume to be independent of deep-sea temperatures" This incorrect formulation. In fact, the sea level reconstruction based on Red Sea d18O, unlike benthic d18O, does not require information about deep-sea temperature because it based on planktonic forams. It is also affected by temperature (sea surface temperatures) but to a lesser degree than benthic d18O.

L 223 Would be useful to clarify how the "stand-alone ice sheet forcing" was defined for penultimate glacial cycle.

[Figure]

L 255 Would be interesting to know why "the retreat of the WAIS" in the interactive experiment "occurs 2 kyr later compared to the one-way experiment"

L 310 I fully agree that if "NEEM temperature reconstruction is applied uniformly in space and over seasons, than in any model GrIS will melt completely. However, if Eemian warming had strong seasonality, as proposed by Merz et al. (2015, CP) with large warming in winter and small warming in summer, then in combination with some other factors, "NEEM paradox" can be resolved.

L 322 See my previous comment

L 355. As I already stated in general comment, not much happened in the Southern Hemisphere in response to freshwater forcing in the Northern Hemisphere. This is why it is not surprising that Antarctic temperature is so flat.

L. 370 Would be useful to show also ocean (subsurface) temperature in the respective figure.

L. 411 Which "environmental forcing" is meant here?

L. 412 It should be Pollard et al. (2015)

L. 428 "Ocean expansion is steep. . ." Rather I would say "the fastest sea level rise due to thermal expansion . . ."

L. 440 "0.42+-0.11" This is a typo. Chapter 5 of AR5 does not contain this number. Instead it referrs to the only available estimate of thermal expansion during the LIG of 0.4 +-0.3 m by McKay et al. (2011). In such case I would recommend to cite original publication rather than IPCC report.

L. 452 "0.42+-0.11" m is not the estimate of glacier contribution to sea level during the LIG but rather the maximum possible sea level rise due to melting of all existing at present glaciers and small ice caps. Obviously, there is no reason to believe that all glaciers melted completely during the LIG and therefore real contribution of glaciers

and ice caps during LIG was probably much smaller than 0.4 m.

L. 523 "...by preventing tundra warming affecting proximal ice sheet margins". This is not very clear.

L. 539 Please correct doi of Berger's paper

L 575. Correct reference is "Science, 349, doi: 10.1126/science.aaa4019, 2015"

Figure 1. Brovkin et al (1997) is not in the reference list

L 717 I suppose this is not original Grant et al. (2012) reconstruction but its smoothed version. Please, make it clear.

L 746 Does "forced" here means the same as "one-way"?

---

## Referee Comment (RC4) · A. Ganopolski (Referee) · 16 Feb 2016

The comment of the Referee#1 "Line 104: Wrong reference: Robinson et al. 2011 ice sheet model uses output of a transient EMIC simulation as input but does not give feedback to the climate model. Helsen et al. 2013 CP would be a more appropriate reference here" is incorrect. In Robinson et al. (2011) the ice sheet model was coupled bi-directionally to the regional climate model REMBO and elevation and albedo feedbacks were accounted for. The only difference between Robinson et al. (2011) and Helsen et al. (2013) is that the former used simplified regional climate model while the later used regional GCM. In both cases regional models were forced on their lateral boundaries by the output of global climate models.

---

## Editor Comment (EC2) · D.-D. Rousseau (Editor) · 21 Feb 2016

Dear author, As Climate of the Past format offers you the opportunity, please reply this review so that some discussion can be engaged between you and the reviewer during the discussion phase.

All the very best

denis-didier Rousseau

Co-Editor in chief Climate of the Past

---

## Editor Comment (EC3) · D.-D. Rousseau (Editor) · 21 Feb 2016

Dear author, As Climate of the Past format offers you the opportunity, please reply this review so that some discussion can be engaged between you and the reviewer during the discussion phase.

All the very best

denis-didier Rousseau

Co-Editor in chief Climate of the Past

---

## Referee Comment (RC5) · EW Wolff (Referee) · 23 Feb 2016

This paper does represent something of a technical achievement, succeeding in making a coupled run of climate and both Greenland and Antarctic ice sheets across the last interglacial (LIG). To demonstrate that ability, and highlight the steps that are needed to improve on it, I think the paper should eventually be published in CP. However it does need quite a lot of work to explain both details and its limitations correctly. I notice that the paper has already achieved several reviews, so I will not go into huge detail but just give some overall comments, with a little more emphasis on data aspects of the study.

The strength of the paper, as I have indicated, comes from the achievement of making such a study. However I think it is important that it is correctly labelled. It is really a

demonstration simulation, not a testable prediction. The Greenland ice sheet coupling is achieved only after applying a randomly chosen scaling to the temperature data (it's a tuning in the sense of aiming at a Greenland SL contribution the authors think is sensible, but random in the sense that there is no reason at all to think that a linear tuning is correct). The Antarctic ice sheet apparently responds despite the ice dynamics processes that many glaciologists consider paramount for West Antarctica not being present (or at least I don't think they are). Given these two issues, the actual values that are achieved seem almost meaningless. I don't suggest they should not be explored, and the relative timing of the contributions is of interest for example, but the paper should make much clearer that it does not in any way represent a success in explaining LIG sea level, rather it is a demonstration of how one might start to assess that in a consistent manner.

Another significant issue I would like to see addressed concerns data. This is in two senses; firstly some critical data seem a little misquoted, and others seem to be ignored. But also there is an opportunity here to test different aspects of the model results rather than just the SL response. In particular the climate response in both polar regions could be well-tested using the recent Capron et al (2014, QSR) compilation; but in fact this paper is not even cited. I suspect for example that this paper would allow the authors less room to suggest that the Greenland temperature response is overestimated in the model, and force them instead to consider that the ice sheet may be too sensitive, which is quite a critical issue.

A final major issue I think the authors need to address concerns the mechanism by which they achieve a significant loss of WAIS – this seems to be global SL and ice shelf viscosity. This seems really surprising to me: global sea level is higher than today really only because of the loss of WAIS in these expts, so it is hard to see why this should be a part of provoking such a loss. That leaves us having to accept that Antarctic temperature in Fig 7a apparently provokes a change in viscosity and loss of ice just a few tenths of a degree above present: this would be a very alarming result,

but seems quite at odds with the mechanisms that usually concern people about WAIS (they generally worry about dynamic loss through the major ice streams and glaciers on the Amundsen Sea side, which have little or no ice shelf restraint, rather than the ice flowing into the large ice shelves). Perhaps I have not understood your mechanism but this definitely needs exploring: either your model is way too sensitive to this process, or glaciologists are worrying about the wrong thing and should be very urgently concerned about ice shelf viscosity. I rather suspect the former as I can't see how there can be such a sharp breakpoint in ice shelf viscosity that a couple of degrees would drain the whole of WAIS and destroy the Ross and Ronne-Filchner Ice Shelves. In any case this certainly needs a discussion.

More detailed comments:

Line 47: Turney and Jones compiled data that were not contemporaneous, ie they combined the maximum temperature at each site over a long time slab. It is therefore impossible to deduce a global mean temperature anomaly from their paper. Probably better to acknowledge this.

Line 56. I think the most commonly cited numbers for LIG sea level are 5-10 m from IPCC AR5, and 6-9 m from the recent Dutton et al (2015, Science) review paper. There is not a great basis for emphasising 6 m in particular.

Page 4. Here is a first place one could mention the Capron et al compilation which could act as a check on your climate outputs or as a forcing in standalone experiments.

Line 186-188 is badly worded. The Grant et al paper uses an approach that doesn't use synchronisation to a mixed record of SL and deep sea temperatures but it doesn't assume anything about their independence or otherwise does it?

Line 192-203. While I understand your decision to scale I think it needs more discussion. From Fig 4a I read off that without forcing you would estimate a Greenland warming of about 3 degrees. This is not only below the NEEM estimate, it's below

other NEEM lower estimates (such as Masson-Delmotte et al 2015), and I am pretty sure it is already similar to other model estimates. Your preferred estimate allows only a one degree warming and this would be really hard to reconcile with NEEM data or with compiled SST data in Capron et al. So, for pragmatic reasons, Ok use the scaling, but I feel you should admit that this might be telling you that your Greenland model is too sensitive, and at least discussing your model in the context of others.

Line 277. While the elevation at NEEM is not perfectly constrained, I suspect its equally important that ice sheet elevation at NEEM is not a strong constraint on the size/area of GrIS. Perhaps re-word.

Line 284. I am not sure what point you want to make here about Cap Century. The same paper also suggests no ice older than 115 ka at Summit but this is clearly not taken to mean there was no Eemian ice there.

Line 314 and around. While we don't understand how an ice sheet at +8 degrees could survive, I still question whether your result illustrates a NEEM paradox or an oversensitive Greenland ice sheet model. You should at least discuss both options.

Line 353-359 and beyond is really confusing. Firstly you say that "Antarctic surface climate is isolated from millennial fluctuations". But then later you agree with previous authors in ascribing the warm Antarctic to the bipolar seesaw. Please make your text consistent. I assume in fact you do think it is the bipolar seesaw response to NH melting that is important in warming the Antarctic at a time when orbital forcing would cool it.

Fig 6b: I could not follow this figure, please explain it better.

Fig 10 is really not comprehensible. It needs a much better caption. In any case I am not sure it serves any purpose since the NHIS evolution dominates everything. This means that while the extent of the highstand above present is a prediction that can be aimed at, the shape of the deglacial rise is really dominated by your (prescribed) NHIS loss.

---

## Author Comment (AC1) · 11 Jun 2016

**We have revised our manuscript 'Last Interglacial climate and sea-level evolution from a coupled ice sheet-climate model'.**

**We would like to thank all four reviewers for their constructive comments that helped to improve the manuscript.**

**Please find below the reviewer's comments in regular italic and a point-by-point rebuttal in bold font.**

**Reviewer 1**

*General comments:*

*The submitted manuscript by Goelzer et al. investigates a new transient simulation of the last interglacial (LIG) period with a bi-directionally coupled climate-ice sheet model. More precisely, the authors use LOVECLIM1.3, an earth system model of intermediate complexity including interactive components for Greenland and Antarctica, i.e., the solely remaining ice sheets during the LIG. Consequently, the focus of the paper lies on climate and ice sheet changes in Greenland and Antarctica and the resulting sea level evolution throughout the LIG. The simulation is compared to previous experiments which exclude ice sheet changes or use a one-way coupling approach. Furthermore, they analyze different sensitivity experiments where specific climate processes are modified or omitted in the experimental setup. The main result of the paper is that the evolution of the Greenland ice sheet (GrIS) is dominated by changes in the surface mass balance whereas the Antarctic ice sheet (AIS) is mainly governed by melting of the shelf area driven by sea-level rise and reduced ice shelf viscosity in a warming climate. A valuable outcome of the model effort is also the temporal and spatial evolution of both the GrIS and the AIS generated within the same climate-ice sheet simulation and thus a consistent experimental setting.*

*The baseline of the paper is the remarkable technical effort to produce a fully-coupled climate-ice sheet model simulation for the LIG, i.e., a simulation which allows feedbacks between all components of the climate system and hence somewhat represents the "best possible estimate" of the LIG climate with a modeling approach. To my knowledge, the simulation is the first of its kind for the LIG and certainly a valuable contribution for the paleoclimate science community. However, in the present manuscript I am missing a comparison of the simulated climate with proxy records, at least for the two key regions Greenland and Antarctica, as this comparison would have the role of an evaluation of the novel model setup. Moreover I expect a more critical discussion of the chosen model approach regarding remaining improvements and challenges.*

*Concerning the formal aspects, I think the manuscript needs to be improved in several aspects. Whereas Sections 1-4 are mostly well-written, the results*

*(Section 5) are sometimes hard to follow and need a revision to become more complete and comprehensive. Some figures are only partly described and very poorly referenced in the text (Table 1 is not mentioned a single time in the text). As you will see, I have many minor comments where I feel the language could be more precise to make the manuscript more reader-friendly.*

**We thank the reviewer for the detailed comments that we have all considered for the revised version of the manuscript. Please find our response to the individual comments below.**

*Please find below the full list of major and minor issues.*

Major issues:

*1. Critical discussion of experimental setup*

*As stated above I expect a section which critically reflects on the quality of your model setup. I think as much as your reference simulation deserves credit for having a pioneering role as a fully-coupled simulation of the LIG it asks for a discussion of its strengths and weaknesses as well as of remaining challenges and possible improvements. This additional section could be in form of a "discussion" or an "outlook" section which both are non-existent at the moment. The discussion should also include a comparison to Helsen et al. 2013 CP, who previously assessed the GrIS retreat during the LIG with a bi-directionally coupled model approach.*

**We have included a new discussion section to discuss in more detail comparison with former work (including the mentioned reference), limitations of the model and possible improvements. Please find details in response to individual comments.**

*Regarding modeling the climate in Greenland I further wonder if your setup includes the relevant feedbacks on temperature and precipitation found in response to a retreating Greenland ice sheet (Merz et al. 2014 CP, Merz et al, 2014 JGR, Hakuba et al. 2012 JGR). I suspect that the limited spatial resolution of the EMIC in the atmosphere (T21) might be a problem here. Furthermore, the authors should address the use of the positive-degree-day method (PDD) for the ice sheets as this is a serious issue for the LIG as shown by van de Berg et. al. 2011 NatGeo.*

**Feedbacks arising from the coupling between ice sheets and climate are in principle included in the model, in particular the albedo-temperature feedback for a retreating ice sheet and for changing surface properties due to surface melting. However, resolution of the atmospheric model is indeed a limiting factor, a so far unavoidable side effect of running a fully coupled model for several thousands of years. The katabatic wind effect discussed by Merz et al. (2014) is therefore likely underrepresented. A quantification of how much the feedback strength depends on spatial resolution of the**

climate model would be an interesting study, but is not something we can add to with our model set-up.

Possible limitations of the model due to its spatial resolution and of the applied PDD scheme are now discussed in a new discussion section in the manuscript:

"A so far unavoidable side effect to running a fully coupled model for several thousands of years is the limited horizontal resolution of the atmospheric model. The katabatic wind effect discussed by Merz et al. (2014) and other small-scale circulation patterns are therefore likely underrepresented. A quantification of how much the strength of ice sheet-climate feedbacks depends on spatial resolution of the climate model would be an interesting study, but is not something we can add to with our model set-up.

The applied PDD scheme has been extensively validated for simulations of the recent past (e.g. Vernon et al., 2013), but several studies point to limitations of this type of melt model when applied for periods in the past with a different orbital configuration (e.g. Berg et. al. 2011; Robinson and Goelzer, 2014). Their results indicate that the stronger summer insolation during the LIG should result in additional surface melt on the Greenland ice sheet compared to simulations based on temperature changes alone. We note that this suggests an underestimation of LIG melt with the PDD model and increased melt if it was corrected for. Thus, including a melt contribution due to insolation would further increase the contrast of the NEEM paradox in our simulation."

*2. Scaling factor*

*You use the scaling factor (described on lines 192-203) as a necessary tuning factor to avoid a complete loss of the Greenland ice sheet during the LIG. I wonder if the scaling factor is necessary due to the simplified representation of the climate in LOVECLIM over Greenland as I guess that the climate-ice sheet feedbacks previously mentioned in major issue 1 (described in Merz et al. 2014 CP, Merz et al, 2014 JGR, Hakuba et al. 2012 JGR) are probably not included.*

Our understanding is that the predominant (temperature-related) feedbacks that are discussed in the mentioned publications and have an impact on the scaling are included in our model. Furthermore, any missing positive feedback, especially if acting in the summer, would further increase the need for scaling we have encountered. Hence, the scaling is needed in any case. See also response to the point before.

*How do you feel that this artificial control affects your result concerning Greenland ice sheet evolution and consequently its contribution to the LIG sea level?*

**It is clear that the scaling has a large effect on the sea-level contribution and that it strongly limits the prognostic capability of the model in this regard.**

*Do I understand it correctly that no scaling factor is applied for the Antarctic ice sheet?*

**Yes, correct. We have not identified a physical process that would justify a similar procedure for the Antarctic ice sheet. Since surface mass balance changes have generally a minor effect for the AIS, we would also not have constraints that could be used to evaluate a scaling on the AIS.**

**A clarifying sentence has been added to the text:**

**"The scaling is only applied for the GrIS, since we have not identified a physical process that would justify a similar procedure for the AIS."**

*Line 274: I think you should state clearly here that the choice of the scaling factor crucially affects the contribution of the GrIS to the sea-level high stand of the LIG.*

**OK, made that explicit:**

**"For the two sensitivity experiments (high, low) with modified scaling (R=0.5, 0.3), the contribution changes to 2.8 m and 0.6 m, respectively, crucially controlled by the scaling factor (Table 2)."**

*3. Additional part describing all (sensitivity) experiments*

*Currently, the manuscript presents results from various sensitivity experiments at different occasions, which makes it hard for the reader to keep the overview. Therefore, it would be much more reader-friendly to add a subsection to Section 4 describing all (sensitivity) experiments and their purpose. I think this subsection could be complemented with a respective list in a table.*

**OK. We have included a new section 4.2, which describes the reference and sensitivity experiments with reference to a new table that lists all discussed experiments.**

*I further advise to clearly state in the text that you define the two-way coupled simulation as "reference". Similar definitions might be worth for the stand-alone experiments etc. Make sure that you use these terms consistently in all text and figures.*

**OK. We have defined the reference simulation in a new section 4.2 as suggested and now consistently refer to "reference" throughout the text. Standalone experiments are now consistently referred to as "forced".**

*4. Extended analysis/description of results*

*I think the manuscript would greatly profit from an extended analysis and some additional figures in order to present a complete picture of your two-way coupled simulation rather than just showing selected aspects.*

*Specifically I request:*

*As I like Figure 3 showing the gained value of the two-way coupling, I think a similar figure for temperature in Greenland and Antarctica would be highly appreciated as these two regions are the main areas of interest in your paper.*

**We have now included additional figures (S1, S2) for Greenland and Antarctic temperature evolution in comparison with ice core records. These are discussed in the new version of the manuscript.**

*Moreover, I am missing a clear statement in the text regarding the results shown in Fig. 3: (i) the simulation with two-way coupling only marginally differentiates from the simulation with one-way coupling with respect to global mean temperature throughout the LIG. (ii) Excluding ice sheet changes and freshwater forcing as done in the noIS simulation leads to a decreased glacial-interglacial temperature contrast and an earlier warming going into the LIG. However, there is only a small difference to the one-way and reference simulation after ca. 128ka! I wonder whether the latter result also applies for temperatures in Greenland and Antarctica?*

**For a discussion of the temperature response over the ice sheets, see response to previous comment. We have extended the interpretation and discussion of Figure 3 following the reviewer's suggestion. The section now reads as follows:**

**"The fully-coupled experiment exhibits a global mean temperature evolution during the LIG that is very similar to the One-way experiment. A much larger temperature contrast at the onset of the LIG in the reference experiment compared to noIS arises mainly from changes in surface albedo and melt water fluxes of the Northern Hemisphere ice sheets, which freshen the North Atlantic and lead to a strong reduction of the Atlantic meridional overturning circulation (Loutre et al., 2014). All three simulations show only small differences in the global mean temperature evolution after 127 kyr BP."**

*In order to evaluate your two-way simulation against data I strongly suggest a comparison of the simulated Greenland and Antarctic temperature evolution with respective ice cores (e.g., NEEM, EPICA). As the NEEM delta18O-based temperature reconstruction likely assumes an overestimated delta18O-temperature relationship you could also include the NEEM temperature curve based on the recent delta18O-temperature relationship presented in Masson-Delmotte et. al. 2015 Cryosphere.*

**Comparison with ice core data is included in the additional figures (S1, S2) showing the temperature response over the ice sheets. See previous comments.**

*It might also be worth to show the evolution of the freshwater fluxes throughout the LIG to complement your findings presented at Line 353pp and Line 429pp.*

**We have instead added a reference to Goelzer et al. (2015), where the climate response to freshwater forcing is discussed in more detail (line 353pp in the manuscript). We estimate that the discussion on thermal expansion (line 429pp) does not warrant a new figure and we have kept the (not shown) there.**

*5. Mass balance of Greenland and Antarctic ice sheet*

*I think a proper definition of the (surface) mass balances for the Greenland and the Antarctic ice sheet is required. Please clearly state what you refer to as accumulation, ablation, runoff, (surface) melting, calving flux and how they combine to the mass balance. Please use the same terms in the text as in the axis labels of Figs. 4 & 7.*

**We have revised the manuscript to be consistent in our terminology and have e.g. replaced all occurrences of "ablation" by "runoff".**

**We have also added a reference to Huybrechts et al. (2011), where the mass balance components of the ice sheet models are described in detail.**

*I think it would be a valuable addition to show the net mass balance as a further panel in Figs. 4 & 7 so the reader can reconstruct the evolution of the ice volume shown in Figs. 4e and 7e. Whenever possible use the same scales for the different terms of the mass balance in Figs. 4 and 7.*

**We have included additional panels in Figures 4 and 7 that show the net mass balance. Display of the different variables on the same scale would render the panels difficult to read, because of the different magnitudes (no change).**

*6. References to figures in text*

*Throughout the paper I miss many references to the corresponding figures, which would substantially help the reader to understand the descriptions in text-form. Please be more precise when discussing panel plots, e.g. put reference to Fig. 4a rather than just to Fig. 4.*

**We have revised the entire manuscript to include sufficient and precise referencing to figures and individual panels.**

*Some examples of missing/imprecise figure references:*

*Line 365: Fig. 7b*

**OK.**

*Line 375: Fig. 7a,d*

**OK.**

*Line 382: Fig. 7d*

**OK (Fig. 7e).**

*Line 393: Fig. 9b after "experiments"*

**OK, included in next sentence.**

*Line 402: Fig. 7a and 2c*

**OK.**

*Line 427: Fig. 3 after "evolution"*

**OK.**

*Line 445: Fig. 10c*

**OK.**

*Line 446: Fig. 10b*

**OK.**

*Minor issues:*

*Lines 23: Please be more specific than "surface mass balance changes"*

**OK. Specified surface meltwater runoff as the governing component.**

*Lines 24-25: "Our results indicate" could be replaced with "The comparison of fully-coupled with stand-alone Greenland ice sheet simulations emphasizes"*

**Not changed.**

*Line 68: change "lower bound of 5.5m" to "lower bound of Eemian sea level rise of 5.5 m above present-day levels"*

**OK, changed.**

*Lines 75-78: This sentence is misleading as it implies that any Southern Ocean warming is induced by the interhemispheric seesaw effect.*

**OK, added "possibly" to allow for other interpretations.**

*Line 104: Wrong reference: Robinson et al. 2011's ice sheet model uses output of a transient EMIC simulation as input but does not give feedback to the climate model. Helsen et al. 2013 CP would be a more appropriate reference here.*

**The reference is correct as confirmed by the comment from reviewer 3 (cp-2015-175-RC4). We have included a reference to Helsen et al. (2013) as another example of a transient LIG simulation of the GrIS.**

*Line 111: "climate and oceanic conditions over the ice sheets and in their proximity" seems not to be a correct/precise statement.*

**OK, removed "over the ice sheets and in their proximity".**

*Lines 109-118: Whereas I like the rest of the introduction, this last paragraph should be improved to better stress the focus and strategy of the paper. I think you should highlight here that you generate the first transient simulation of the LIG with a bi-directional coupling of climate and GrIS/AIS model components. Furthermore, please clarify that you study key mechanisms and feedback processes with the aid of sensitivity experiments and with the comparison to one-way coupled and stand-alone ice sheet simulations. I would also state here that you focus on climate and ice sheet changes in Greenland and Antarctica and the resulting sea-level evolution throughout the LIG.*

**OK. We have extended the last paragraph of the introduction following the reviewer's suggestions.**

**"Here, we present modelling results from the first fully coupled climate-ice sheet simulation of the LIG period (135 kyr BP to 115 kyr BP) using ice sheet models of the GrIS and AIS and a climate model of intermediate complexity. In this set-up LIG sea-level evolution and climate-ice sheet interactions can be modelled in a consistent framework. With focus on climate and ice sheet changes in Greenland and Antarctica and corresponding sea-level changes, we compare results from the fully coupled model to former climate simulations with prescribed ice sheet changes and uncoupled ice sheet experiments."**

*Lines 170-171: specify "ice loading changes" e.g., with "ice loading changes coming out of the penultimate glacial period".*

**OK. Modified as suggested.**

*Line 184-191: You use the sea-level reconstruction by Grant et al. 2012 as boundary condition for your simulations. Wouldn't it be more consistent to use the "internal" sea level corresponding to the simulated global ice sheet changes?*

*What would be the consequences for the melting of the AIS which apparently most strongly responds to sea level changes? What are the reasons for driving the model with a respective sea-level reconstruction instead?*

**Ultimately, it would indeed be desirable to apply a consistent 'internal' sea-level forcing. However, there are a number of complications that led us to use a prescribed forcing. 1) The predominant sea-level forcing is the NH contribution, which we currently do not model prognostically. 2) The GrIS and AIS models need forcing well before the modeled period for the spin-up, which would require some sort of anomaly method. 3) For the AIS, where sea-level change is a dominant forcing and the AIS contribution itself would have to be accounted for, regional sea-level changes would also need to be estimated.**

*Line 205: introduce the abbreviation "(SA)" here.*

**We have revised the terminology and now consistently refer to the additional experiments as "forced" experiments. The term "stand-alone" is only used for former experiments and ice sheet model runs in the spin-up.**

*Line 215: The title of this subsection could be more specific, e.g., "Initialisation of the reference simulation"*

**OK.**

*Line 241: might be more precise to replace "importance of ice sheet changes" with "importance of two-way coupling between the climate model and the ice sheet models for the GrIS and the AIS"*

**Not changed. Comparison here includes a case without NH ice sheet forcing, thus not limited to GrIS and AIS.**

*Line 247-251: This finding is somewhat hard to understand. May be it would help if you show the freshwater fluxes in a figure (as also requested in major issue 4) and put a respective reference.*

**These results are largely based on mechanisms well documented in the studies of Loutre et al. (2014) and Goelzer et al. (2015). We have made that clear in the text and added the references again.**

*Line 264: Rather put the reference to Fig. 4e here.*

**OK. Modified to include references to both Fig. 5 (showing the retreat) and Fig. 4e (showing the volume and area change).**

*Lines 334-336: I think you could add here that the ice-albedo feedback is a positive feedback.*

**OK.**

*Line 351pp: "The warming necessary...." This sentence is not easy to comprehend. Please revise.*

**OK, passage revised:**

**"The warming before the peak is around a factor two faster than the cooling afterwards, with both transitions being near linear on the millennial time scale. "**

*Line 365: show the freshwater fluxes in a figure or add (not shown) after "hemispheres".*

**OK, added "not shown".**

*Line 367: You speak of "ablation" but in Figure 7 you name it runoff – is this the same? Please be consistent with all terms describing the mass balance of the GrIS and the AIS (see also major comment 5)*

**OK. We have replaced "ablation" by "runoff" everywhere in the manuscript.**

*Line 376: add (not shown) after "130 kyr BP".*

**OK, added "(not shown)".**

*Line 380: add (not shown) at end of sentence or put a reference to Goelzer et. al. 2015.*

**OK, included reference to Goelzer et al. (2015).**

*Line 418-419: Is "their model" equal to the simulation you termed "one-way" at other occasions in the script?*

**Yes, modified the text accordingly:**

**"The main retreat in their one-way coupled climate model run happened ~129.5 kyr BP, a timing predating the time of retreat in the fully coupled model by ~2 kyr due to the difference in atmospheric and oceanic forcing. "**

*Line 470-472: I think this sentence should be rephrased to state more clearly that you artificially limit the melting of the GrIS to conform to existing ice core constraints.*

**OK. Added a sentence to describe this limitation:**

**"However, this result is strongly controlled by the need to scale the climate forcing to match existing ice core constraints on minimal ice sheet extent. "**

*Line 477-478: Please be more specific. I think "ice-climate feedback" is a too general term for a take-home message in the conclusions.*

**OK. reformulated:**

**"The treatment of albedo changes at the atmosphere-ice sheet interface plays an important role for the GrIS and constitutes a critical element when accounting for ice sheet-climate feedbacks in our fully-coupled approach. "**

*Line 482: I think it should also be stated here that an unconstrained fully-coupled climate-ice sheet simulation does not fully agree with data, e.g., the GrIS would melt away completely during the LIG. This implies deficiencies in the model physics or unknown/excluded processes. It also emphasizes the NEEM paradox of strong warming coinciding with limited GrIS melting that can hardly be understood in a model perspective.*

**We have included statements in the conclusion following the suggestion of the reviewer:**

**"However, this result is strongly controlled by the need to scale the climate forcing to match existing ice core constraints on minimal ice sheet extent. This shortcoming in our modelling reflects the NEEM paradox, that strong warming over the ice sheet coincides with limited mass loss from the GrIS, indicative of a fundamental missing link in our understanding of the LIG ice sheet and climate evolution."**

*Table 1: Needs to be discussed in the text or should be removed.*

**OK, now referring to Table 1 in two places in the results section, where the results in Table 1 were already discussed.**

*Figure 1: The references (Opsteegh et al. 1998, Brovkin et al. 1997 and Goosse and Fichefet, 1999) mentioned in Fig. 1 should also be added to the reference list.*

**OK, references included.**

*Fig. 4b,c,d: Does the horizontal stippled line represents the pre-industrial level? Please clarify in figure caption.*

**Yes, have included a clarification:**

**"Horizontal dashed lines give the pre-industrial reference values."**

*Fig. 6b: This schematic is somewhat difficult to comprehend and it is only mentioned once in the text. Should be revised or removed.*

**Most of the last paragraph of 5.1 is relying on this schematic, which aims to illustrate the main controls on albedo changes in the model. We prefer to keep it in.**

*Figure 7d and text: Is there a difference between shelf melting and sub-shelf melting? Please be consistent in text and figures*

**OK, we now consistently refer to sub-shelf melting throughout the manuscript.**

*Figure 9b: Does the blue curve represent the experiment with excluded surface AND sub- shelf melting or just the latter? In line 392 you mention both. Please revise to be consistent in text and figures.*

**OK. The blue curve denotes an experiment with no sub-shelf melting. Added clarifications in the figure caption and in the text.**

*Figure 10: please number the panels with a,b,c. Furthermore, the figure caption should include additional information, e.g., the meaning of the stippled lines.*

**OK, added panel indicators (a,b,c) and description of the median and percentiles.**

*Technical corrections:*

*Line 89: remains*

**OK.**

*Line 96: van de Berg*

**OK.**

*Line 256: "is retreating" rather than "has retreated"*

**No change. Surface melt water runoff is the dominant mass loss for a predominantly land-based ice sheet because the calving flux is close to zero.**

*Line 379: a weakening*

**OK.**

*Caption of Fig. 8: move listing of (a), (b), (c) in front of description as done in all other figure captions.*

**We have added alphabetic panel indicators in all multi-panel figures and now consistently refer to panels in the captions with in-line indicators.**

**Reviewer 2**

*This study assesses the Last Interglacial climate and ice sheet evolution in a two-way coupled approach. The novelty is in the fully coupled method. Especially promising is the simulated evolution of both the Greenland and the Antarctic ice sheet in one overarching climate-ice sheet framework, which allows for assessing their relative contributions to the global mean sea-level highstand during the Last Interglacial. As such the study is interesting as should be published. However, some parts are unclear and lack information and/or discussion.*

*Please discuss the comments below before publication in CP.*

**Many thanks for the detailed comments that have helped to improve the manuscript. Please find our answers to the comments below.**

*GENERAL COMMENTS*

*1) Sea-level forcing from a Red Sea record is prescribed. Are the simulated sea-level changes from the Greenland and Antarctic ice sheet somehow added to this during the simulation?*

**No. Interpreted as a global sea-level record, the Red Sea record already includes the contributions of the ice sheets. See also discussion of point by reviewer 1 (Line 184-191).**

*How certain is the Red Sea record? And how much does it affect the sea-level contributions of the two ice sheets and the total sea-level changes simulated? The discussion on this (lines 420-424) is too short.*

**The sea-level contribution of the GrIS is largely independent from the sea-level forcing. For the AIS, however, a comparison with a sea-level forcing based on a benthic δ18O record shows a large influence on the timing of the WAIS retreat. We have not attempted to formally quantify the uncertainty associated with the sea-level forcing but note that there are large uncertainties in the timing. This was already described in the manuscript, but we have included clarifications to improve on that point:**

**"It is noteworthy in this context that the prescribed sea-level forcing imposes an important control for the timing of the Antarctic retreat and is a source of large uncertainty. We have only used the central estimate of the Grant et al. (2012) sea-level reconstruction, but propagated dating**

**uncertainties could accommodate a shift of the forcing by up to 1 kyr either way. "**

*2) Related to this: Would it be possible to fit your model results better to the Kopp et al. (2009) reconstructions if uncertainties in the Red Sea sea-level record are included, or if you use the benthic d18O-stack? In other words can you suggest improvements to the NH ice sheet retreat records, based on the comparison between your simulations and the Kopp reconstructions?*

**As suggested in response to the previous comment, uncertainty in the age model of the Grant et al. sea-level reconstruction could in principle be used to force the AIS to an earlier retreat, better in line with the Kopp reconstructions. We have not attempted that, since other uncertainties, in particular in the climate forcing are large and do not warrant to attempt a precise chronology. Conversely, using the benthic d18O-stack would lead to a later retreat of the AIS and thus increase the mismatch to the Kopp reconstruction.**
**We have included a discussion item of similar content in the text.**

**Earlier work (Loutre et al., 2014; Goelzer et al., 2015) has shown that the NH ice sheet reconstruction based on Lisiecki and Raymo (2005) is preferable to other reconstructions. We refer to these publications, with detailed discussion on this aspect.**

**In both cases (AIS and NHIS) the climate response (to ice sheet retreat and resulting FWF) was our main guideline in evaluating model performance, which renders comparison to the Kopp et al. (2013) an additional, independent validation, rather than a tuning goal in itself.**

*3) Why is the temperature forcing over Greenland so high that it melts away the Greenland ice sheet entirely? What are the summer and annual mean temperature anomalies for the Last Interglacial? Please compare and discuss this with respect to proxy data, and previous climate model simulations (see e.g. Bakker et al. (2013) and Lunt et al. (2013) for global intercomparisons). The method of uniform scaling is a bit eccentric, and needs better argumentation.*

**Please compare response to comment 4. of reviewer 1.**

*4) Related to this: the experimental set-up misses a section that describes how the simulated temperatures (and accumulation) are converted to (surface) mass balance. Which scheme do you use? With which parameter settings? The latest studies simulating the Last Interglacial Greenland ice sheet evolution show that differences in parameter settings have a huge effect on how much the ice sheet melts (e.g. Robinson et al., 2011; Stone et al., 2013; Langebroek and Nisancioglu, 2016).*

**We have added a description of the surface mass balance treatment in the model description. The model parameters remain unmodified from earlier**

**studies with the same model (e.g. Huybrechts et al., 2011) and have been extensible validated against other SMB models (e.g. Vernon et al., 2013). See also next point:**

**"The surface mass balance model is based on the positive degree-day (PDD) method (Janssens and Huybrechts, 2000) and distinguishes between snow accumulation, rainfall and meltwater runoff, all parameterized as a function of temperature. Surface melt is estimated based on two distinct PDD factors for ice and snow and may be retained and refreeze in the snow pack. Melt model parameters are unmodified compared to earlier studies (Goosse et al., 2010; Huybrechts et al., 2011) and have been extensively validated for the present day (e.g. Vernon et al., 2013)."**

*5) These studies validate their ice sheet model results to the present-day observed ice sheets. I think this is what you need to do as well. Compare your present-day or pre-industrial climate and ice sheet configuration to observations and discuss the differences. This will validate the model set- up, and increase confidence in your model results.*

**The same has been done for our model in earlier studies (e.g. Huybrechts and de Wolde, 1999). For the GrIS the model has been validated recently for present day simulations (Fürst et al., 2015) with parameters very close to the ones in our study.**
**We have included figures of the simulated present day configurations of both ice sheets at the end of this rebuttal for information. Since our focus in this study is the LIG and large-scale changes in the ice sheets, we estimate that a close match to present-day observations is less of an issue and we would not include these figures in the manuscript.**

*How do you deal with the differences between the atmospheric and ice sheet model grids?*

**The ice sheet models are forced in anomaly mode. We have included additional information in the model description:**

**"Climate anomalies are interpolated to the ice sheet grids using Lagrange polynomials and the SMB-elevation feedback is accounted for directly in the PDD model on the ice sheet grid."**

*6) Also for Antarctica some discussion is lacking:*

*a. Lines 375-380: Can you show model "evidence" for the see-saw effect taking place in your model results? E.g. assess Atlantic meridional ocean circulation or heat transport. Do they really decrease?*

**This result pertaining mainly to the climate response to the NH freshwater forcing is discussed in Goelzer et al. (2015) and not repeated here. A reference has been added in the text.**

*b. Lines 381-390: What do you mean with "overshoot behaviour"? Is the Antarctic ice loss not related to the positive temperature anomaly? Which part is overshoot?*

**The main mass loss from the AIS in that period is due to grounding-line retreat, not due to surface melting. The overshoot behaviour discussed in the manuscript concerns this mechanism. Please see also response to comment 5, reviewer 3.**

*c. Also, how does the present-day/pre-industrial simulated Antarctic ice sheet look like? Is this not too sensitive to the temperature forcing, as is the case for Greenland? So in other words, no correction is needed for the temperature forcing over Antarctica?*

**No correction needed. See response to comment reviewer 1.**

*d. Lines 391-402: these sensitivity experiments need more explanation, and a reference to Figure 9b.*

**We have included an additional sub-section 4.2 in the Experimantal setup to extend the description of the sensitivity experiments.**

**OK, reference to Figure 9b included.**

*7) The Section about freshwater input and thermal expansion of the ocean is very interesting, but also lacking information. How large is the freshwater input (Sv) and how long do the episodes take? Another figure or table would be useful.*

**See response to similar comment by reviewer 1.**

*8) Concerning the "double" peak in the Kopp reconstruction: Do you have suggestions why your model results do not reproduce this? Is it because of too constant the climate forcing, too slow regrowth of the ice sheets, or other missing feedbacks? Please discuss.*

**Our model results do not provide evidence for a double peak, mainly because the forcing does not show such variations. However, while the median projections in Kopp et al., (2009) visually suggest a double-peak structure, the uncertainty range is wide enough to accommodate a global sea-level trajectory without intermediate low stand. Our discussion in the manuscript has been extended in that regard to clarify that we are not convinced reproducing a double peak structure is a necessity.**

*SPECIFIC AND TECHNICAL COMMENTS*

*1) Greenland and Antarctic ice sheets are abbreviated in line 59, please use these abbreviations in the remainder of the text*

**OK, used abbreviations consistently throughout the text.**

*2) A bit more information on the coupling procedure is necessary (Section 4.2). How often do they interact or are the components updated, every day/year/1000 years?*

**No change. This information is already present in section 3.**

*3) Lines 275-293: You can also use the reconstructed limits for the Last Interglacial surface elevation change at the ice core locations compared to PI (e.g. NGRIP-members, 2004, Johnsen and Vinther, 2007, NEEM community members, 2013) to evaluate your model results.*

**In our estimate reconstructed elevation changes are highly uncertain. This was already mentioned in the text.**

**"Elevation changes from that ice core are however not very well constrained and even if they were, would leave room for a wide range of possible retreat patterns of the northern GrIS (e.g. Born and Nisancioglu, 2012)"**

*4) Lines 294-305: I don't understand the need of such a speculative section. What is the surface mass balance evolution over the Greenland ice sheet? The resulting ice volume changes are shown in Fig. 4.*

**No change. The timing of the GrIS contribution to sea-level is a key question of this paper. It is important in how far the evolution can be constrained by existing data and model evidence. However, we have moved this part to the new discussion section.**

*5) Lines 325-346: This section is difficult to read. It would be better to better explain the sensitivity experiments. Is "forced" the same a "stand-alone" as you call it earlier in the text? Better also to discuss the simulated maximum sea-level contribution in two steps: 1) effect of temperature scaling factor on resulting ice volume changes, 2) effect of coupling ("forced/stand-one" vs "coupled") on ice volume change.*

**We have revised the use of stand-alone and forced throughout the document, the latter referring now exclusively to the forced repeat-experiments using climate data from the fully coupled run.**

*6) Lines 360-365: comparing the Last Interglacial accumulation to pre- industrial is a bit difficult if you base the calculation on differently sized areas. Maybe the accumulation actually didn't increase in many locations? What happens over NEEM? Maps for certain time slices would be much more helpful.*

**We agree with the reviewer that time resolved maps would be better suited to reveal details of the accumulation change. However, as a minor**

**contribution to the overall ice sheet mass balance we prefer to keep accumulation change treated in condensed form as is the case now. NEEM is not on the Antarctic ice sheet discussed here.**

*Line 24: "reference experiment", either describe the reference experiment, or omit the mentioning of this and change the values to express the full range of your results (0.62-2.77m)*

**Replaced "the reference experiment" by "our reference experiment".**

*Lines 32-33: would be nice to add which part of the ~5m is due melting of the Greenland and which due to the Antarctic ice sheet*

**Numbers for the GrIS and AIS are given for the individual peaks just before. Although the timing of the two ice sheets is not identical, we believe this is sufficient information for an abstract.**

*Line 63: skip "e.g."*

**OK.**

*Line 71: "mean" instead of "central"*

**Not changed. Estimates are given in different form, not always as a mean with standard deviation.**

*Line 77: add "possibly" caused by*

**OK.**

*Lines 84-86: make new section, and add "evidence" for possible reduction of the LIG AIS*

**No change. We are not aware of direct evidence of an AIS reduction as discussed. The whole paragraph is dedicated to the uncertain AIS contribution.**

*Line 87: better constrained than ...? (I assume AIS evolution)*

**OK.**

*Line 102: also mention latest work (Langebroek and Nisancioglu, 2016)*

**We have updated our reference list to include recent publications (e.g. DeConto and Pollard, 2016). Reference to Langebroek and Nisancioglu (TCD, 2016), Rasmus et al., (CPD, 2016), Merz et al., (CPD, 2016) and Landais et al., (CPD, 2016) are foreseen as they get published.**

*Line 99: correct reference is Born and Nisancioglu, 2012; please also update in rest of text*

**OK. Corrected throughout the manuscript.**

*Line 104: incorrect reference, maybe you meant regional climate model, or a different reference*

**No change. See Reviewer comment 4 in CP discussion.**

*Line 106: reformulate "results" – what results?*

**We meant results from "Ice sheet modelling studies on the Antarctic ice sheet during the LIG" as mentioned in the sentence before. Added some clarification:**

**"However, some results on the AIS during the LIG have been presented in studies with main focus on other time periods …"**

*Line 108: check correct reference in reference list for Pollard and DeConto, 2009 or 2015?*

**No change. Correct reference for an Antarctic ice sheet simulation spanning the Last Interglacial, but without specific focus on it.**

*Lines 113-114: skip "high-resolution", grid boxes of 10 or 20 km is normal, not high for ice sheet models*

**OK.**

*Line 121: EMIC description with capital letters or not – make consistent with abstract*

**OK. Abbreviation is not used anymore:**

**"Earth system model of intermediate complexity"**

*Lines 123-124: "The model has been utilised ..." – but without dynamic ice sheets, and two-way coupling, right? Rewrite to make clear.*

**No change. All listed references used the fully coupled model.**

*Lines 133-134: what is the resolution of T21 in degrees or km, approximately? "high-resolution ice sheet models", see earlier comment*

**OK. Replaced "high-resolution" by "higher resolution" to focus on the relative difference.**

*Lines 137-138: are the freshwater fluxes etc the same as in the earlier version of the model, or is the set-up the same? Please rewrite.*

**OK. Sentence split and rewritten:**

**"The ice sheet models in turn provide the climate model with changing topography, ice sheet extent (albedo) and spatially and temporally variable freshwater fluxes. The coupling procedure for these variables is unmodified to earlier versions of the model (Goosse et al., 2010), while recent model improvements for the ice-climate coupling interface are described in Appendix A."**

*Section 3.1: Would make more sense to make Section 3.1 a part of 3.2*

**No change. Section 3.1 is about forcing, while 3.2 is about the model response.**

*Line 157: change to "sea-level equivalents (SLE)"*

**OK. Changed to "sea-level equivalent", but SLE only used in Table 1 and defined there.**

*Lines 158-160: sentence very unclear, please rewrite*

**OK. Sentence split and reformulated:**

**"The Antarctic contribution to global sea-level change is calculated taking into account corrections for ice replacing seawater, seawater replacing ice and isostatic bedrock movements replacing seawater. These effects are mainly of importance for the marine sectors of the WAIS."**

*Lines 181-183: Is insolation calculated for each latitude and for each month? Not entirely clear, especially because figure only shows 2 months and 2 latitudes.*

**Insolation is spatially and temporally resolved. Added clarification in caption to Figure 2 that the two curves are for illustration:**

**"Average monthly insolation anomaly (a) at 65° North in June (black) and 65° South in December (blue) to illustrate the spatially and temporally resolved forcing (Berger, 1978) …"**

*Line 186: change "the latter" to "this data"*

**OK.**

*Would be nice to explain what this reconstruction is based on.*

**This sentence has been revised according to comment by reviewer 4:**

**"The chronology of this data is thought to be superior compared to sea-level proxies based on scaled benthic δ18O records (Grant et al., 2012; Shakun et al., 2015). "**

*Line 193: Skip "As a measure"*

**OK.**

*Line 208: skip "comparison between"*

**OK, reformulated.**

*Line 209: skip "recorded"*

**No change. Important to mention that the climate forcing is recorded.**

*Lines 208-210: The ice sheet response to what?*

**OK, replaced "response" by "evolution".**

*Line 211: Are these "Additional experiments" stand-alone experiments or coupled?*

**Yes, stand-alone experiments. We are still describing the same "forced" experiments. Added clarifications in the text.**

*Lines 217-219: What is the climate forcing for this initialisation? And how large are the 'initial' Greenland and Antarctic ice sheets, so at 135ka?*

**This was done following established procedures, recently updated in Goelzer et al. (2015). References have been included in the text to clarify that (Huybrechts and de Wolde, 1999; Huybrechts, 2002; Goelzer et al. 2015).**

**We have included additional panels for the initial Greenland and Antarctic ice sheets at 135 kyr BP in figures 5 and 8.**

*Line 231: The first section of the Result should be named "5.1 Climate evolution" or something similar*

**OK, added section header "5.1 Climate evolution"**

*Lines 231-235: and what are the differences to Loutre et al., 2014?*

**Comparison to Loutre et al. (2014) and Goelzer et al. (2015) are given in the first section.**

*Line 249: Southern Ocean (SO)*

**OK.**

*Lines 250-251: I don't see this cooling event in the one-way experiment, please rewrite.*

**Added reference to Goelzer et al. (2015), where the one-way experiment is described.**

*Line 254: change to "mass balance dominated by ablation"*

**OK. Also refer to runoff instead of ablation now following comments of the other reviewers.**

*Section 5.1: What do you call "ablation"? runoff + calving or only runoff? Need for some definitions here.*

**OK. Have revised the terminology. "Ablation" is replaced by "runoff" or "surface meltwater runoff".**

*Lines 254-255: "Marginal" could mean "just a bit" or "on the rim", please clarify.*

**OK. Replaced "Marginal … runoff" by "runoff from the margins";**

*Section 5.1: "Temperatures", are these summer mean or annual mean? Surface or air temperatures? Please be more precise.*

**OK, further specified "air" temperatures. We are describing a physical process here. Physically, accumulation increase is due to increased temperature not due to increased mean temperature, or for that matter, annual temperature.**

*Figure 4: are the dashed lines the pre-industrial values? Would be great to have these numbers also for the ice area and volume.*

**Yes, see also comment of reviewer one.**
**Reference values for volume and area have been included in Figure 4.**

*Line 268: change "furthest" to "maximum"*

**No change. We mean the furthest retreat as "over the largest distance". "Maximum" retreat could mean the maximum attainable retreat.**

*Line 269: change "Conversely" to "At the same time"*

**OK.**

*Line 317: Not sure if Merz et al., 2014 is the correct reference here, as they focus on the effect of topography on precipitation during the Last Interglacial.*

**No change. There are two papers of Merz et al., in 2014. The one we refer to is the one about temperature.**

*Line 334: "Figure 6, left" should be "Figure 6a", check also rest of section.*

**OK. Also replaced twice "(Figure 6, right)" by "(Figure 6b)"**

*Line 340: skip "therefore"*

**OK.**

*Line 365, "Figure 7b"*

**OK.**

*Line 367: so ablation is runoff?*

**Yes, replaced "ablation" by "runoff" throughout.**

*Figure 7: what is the present-day ice area and volume in your model set-up?*

**OK. Reference values for volume and area have been included in Figure 7.**

*Line 375: include reference to Figure 7d*

**OK.**

*Line 414: "included" instead of "attempted"*

**OK.**

*Line 428: "Ocean expansion is rapid during ..."*

**OK.**

*Line 439: skip "well"*

**OK.**

*Lines 439-440: the estimated LIG ocean thermal expansion is 0.4+0.3m according to the IPCC report, they use McKay et al., 2011 as a reference. Please rewrite.*

**Thank you for spotting this mistake. Corrected.**

*Line 443: "AIS and thermal expansion"*

**OK.**

*Lines 443-445: add reference to Figure 10*

**OK.**

*Figure 10: add information on confidence levels to figure caption*

**OK.**

*Line 453: change "hiatus" to "regrowth" or similar*

**OK.**

*REFERENCES*

*Bakker, P., Stone, E. J., Charbit, S., Gröger, M., Krebs-Kanzow, U., Ritz, S. P., Varma, V., Khon, V., Lunt, D. J., Mikolajewicz, U., Prange, M., Renssen, H., Schneider, B., and Schulz, M.: Last interglacial temperature evolution – a model inter-comparison, Clim. Past, 9, 605–619, doi:10.5194/cp-9-605-2013, 2013.*

*Born, A., and Nisancioglu, K. H.: Melting of Northern Greenland during the last interglaciation, Cryosphere, 6, 1239-1250, doi:10.5194/tc-6-1239-2012, 2012.*

*Johnsen, S. J. and Vinther, B. M.: Greenland stable isotopes, in: Encyclopedia of Quaternary Science, edited by Elias, S., vol. 2, pp. 1250–1258, Elsevier, 2007.*

*Kopp, R. E., Simons, F. J., Mitrovica, J. X., Maloof, A. C., and Oppenheimer, M.: Probabilistic assessment of sea level during the last interglacial stage, Nature, 462, 863-867, 628 doi:10.1038/nature08686, 2009.*

*Langebroek, P. M. and Nisancioglu, K. H.: Moderate Greenland ice sheet melt during the last interglacial constrained by present-day observations and paleo ice core reconstructions, The Cryosphere Discuss., doi:10.5194/tc-2016-15, in review, 2016.*

*Loutre, M. F., Fichefet, T., Goosse, H., Huybrechts, P., Goelzer, H., and Capron, E.: Factors controlling the last interglacial climate as simulated by LOVECLIM1.3, Clim. Past., 10, 1541-1565, doi:10.5194/cp-10-1541-2014, 2014.*

*Lunt, D. J., Abe-Ouchi, A., Bakker, P., Berger, A., Braconnot, P., Charbit, S., Fischer, N., Herold, N., Jungclaus, J. H., Khon, V. C., Krebs-Kanzow, U., Langebroek, P. M., Lohmann, G., Nisancioglu, K. H., Otto-Bliesner, B. L., Park, W., Pfeiffer, M., Phipps, S. J., Prange, M., Rachmayani, R., Renssen, H., Rosen-bloom, N., Schneider, B., Stone, E. J., Takahashi, K., Wei, W., Yin, Q., and Zhang, Z. S.: A multi-model assessment of last in- terglacial temperatures, Clim. Past, 9, 699–717, doi:10.5194/cp- 9-699-2013, 2013.*

*McKay, N. P., J. T. Overpeck, and B. L. Otto-Bliesner, 2011: The role of ocean thermal expansion in Last Interglacial sea level rise. Geophys. Res. Lett., 38, L14605.*

*NEEM community members: Eemian interglacial reconstructed from a Greenland folded ice core, Nature, 493, 489–494, 2013.*

*NGRIP-members: High-resolution record of Northern Hemisphere climate extending into the last interglacial period, Nature, 431, 147–151, 2004.*

*Robinson, A., Calov, R., and Ganopolski, A.: Greenland ice sheet model parameters constrained using simulations of the Eemian Interglacial, Clim. Past., 7, 381-396, doi:10.5194/cp-7-381-2011, 2011.*

*Stone, E. J., Lunt, D. J., Annan, J. D., and Hargreaves, J. C.: Quantification of the Greenland ice sheet contribution to Last Interglacial sea level rise, Clim. Past., 9, 621-639, doi:10.5194/cp-9-621-2013, 2013.*

**Reviewer 3 – Andrey Ganopolski**

*The manuscript by Goelzer et al. presents results of the first fully interactive simulation of climate and ice sheet evolution during the penultimate glacial termination and the last interglacial (LIG) using an Earth system model of intermediate complexity. The authors show that reconstructed temporal dynamics of sea level during the LIG can be successfully reproduced by their model. The authors for the first time demonstrated that disintegration of the last fraction of the West Antarctic ice sheet (WAIS) at the beginning of LIG can be solely explained by the dynamical response of the ice sheet to sea level rise. The manuscript presents in depth analysis of the processes and feedbacks operating in the system supported by a set of sensitivity experiments. The manuscript is well-written and properly illustrated. I believe this is an important scientific contribution and I would recommend it for publication in CP after minor revision.*

**Thank you very much for the comments and suggestions that we have responded to in detail below.**

*General comments*

*1. Although the manuscript by Goelzer et al. is not the first paper produced in the framework of the same project and many technical details have been already described in Loutre et al (2014) and Goelzer et al (2015), for the readers' convenience a more detailed description of experimental design would be*

*helpful. In particular I would suggest (i) provide information of how surface mass balance of ice sheets was simulated and give in the table the values of semi-empirical parameters; (ii) explain how temperature and precipitation anomalies from low-resolution climate component were applied to high resolution ice sheet models and how changes in ice sheet elevation and extent were accounted for; (iii) how simulated ocean temperature anomalies were used to compute submarine melt of ice shelves; (iv) how one-way coupling experiments have been performed; (v) how "present" GrIS and AIS have been simulated.*

**We have included additional information in the model description as follows. i) A PDD model is used to calculate the SMB with unchanged parameters compared to other studies (included references). ii) Climate anomalies are interpolated to the ice sheet grids using Lagrange polynomials. The SMB-elevation feedback is accounted for on the high-resolution ice sheet model grid.**

**iii) The submarine melt parameterisation is described in Appendix A.**

**iv) Forced experiments (as we now refer to consistently) are identical to the fully coupled experiments except that climate forcing is read from file (from an earlier simulation) rather than dynamically calculated. We have included an additional sub-section 4.2 describing these experiments in more detail.**

**v) It is not feasible to run the fully coupled model from 135 kyr BP all the way to the present day. Our present-day ice sheet simulations are therefore the result of standalone ice sheet experiments continuing from the standalone spin-up simulations to the present day following established procedures (references to Huybrechts and de Wolde, 1999; Huybrechts, 2002 and Goelzer et al., 2015 have been included).**

*2. I have a question concerning scaling technique to reconstruct Northern Hemisphere (NH) continental ice sheets during penultimate termination. According to the manuscript, evolution of NH ice sheets were prescribed using Lisiecki and Raymo (2005) benthic stack L&R04 and the Fig. 4 from Goelzer et al. (2015) shows that according to L&R04 the termination was only half-way at 130 ka with the global sea level still ca. 50 m below present. This would imply existence of large continental ice sheets in the NH which is consistent with the Fig. 2 from Goelzer et al. (2015). How- ever, according to the Figure 10 (top) from the new manuscript, the volume of NH ice sheets at 130 ka was only 10 meters in sea level equivalent which is only 10% of their LGM value. If I misunderstood your approach, please clarify.*

**In the revised version of Goelzer et al. (2015), we have included an extended description of the reconstruction methods used for the NH ice sheets, which explains our approach. "Our method does not guarantee that the sea-level contribution of the reconstructed NH ice sheets closely**

follows the global ice volume curve. This is generally due to the mismatch between global ice volume and NH ice sheet reconstruction during the post-LGM period, and in part related to the unconstrained contribution of other components (AIS, thermal expansion)."

*3. To prevent GrIS from complete melt, the authors scaled down simulated temperature anomalies used for calculation GrIS surface mass balance. This is somewhat surprising in a view that simulated glacial-interglacial global temperature change in the model is only about 2C which is much less than results of PMIP2 and 3 models which simulated global LGM cooling of 4-5C. Moreover, uncorrected simulated GrIS temperature anomalies during LIG are only about 3C which is still well below "NEEM temperature reconstructions". It would be useful to show simulated summer temperature anomalies over the GrIS because summer temperatures are the most important for ice sheet mass balance.*

**The global mean temperature anomaly is not a good measure for the LOVECLIM model, which exhibits a relatively strong polar amplification. Furthermore, summer temperature anomalies are larger than annual mean anomalies because of the quasi- instantaneous albedo-temperature feedback, which is predominant at the margins of the ice sheet.**
**We have now included an extra figure (S4) showing annual mean and summer temperatures in comparison.**

*4. While I have no problem with the pragmatic decision to scale GrIS temperature anomalies down, I am missing an explanation why the authors decided to use the factor 0.4 as the reference value and considered 0.3 and 0.5 as the upper and lower limits. I wonder whether simulation for scaling factor 0.4 is better than for other two, can the value 0.5 can be accepted or rejected by empirical constraints and whether any larger scaling factors can (or cannot) be ruled out? I believe that at present the only thing we can say with some confidence about GrIS during LIG is that melting of more than half of modern GrIS would be difficult to reconcile with the existing empirical constraints. Any number below 3 meters is equally probable and therefore implied accuracy of reported "1.4 m" significantly underestimates uncertainties of this estimate. I also found it noteworthy that three numbers for the range of GrIS contribution during LIG ( 0.6, 1.4, 2.8 m) given by the authors are almost identical to the values given in the recent paper by Calov et al. (2015, CP): 0.6, 1.4, 2.5 m.*

**We have included explanations in the experiment description as follows. "The range of parameter R is chosen to retain an acceptable agreement of the minimum GrIS extent during the LIG with reconstructions. In practice, the high scaling factor is chosen to produce the smallest minimum ice sheet extent, which still has ice at the NEEM site. The low scaling factor was adopted to produce the smallest minimum ice sheet extent still covering Camp Century."**

**The match of our results with the numbers in Calov et al. (2015) is purely coincidental.**
**We have added a note on that fact in the discussion section:**

**"Our range of modelled GrIS sea-level contribution is in close agreement with results from a large ensemble study of the LIG sea-level contribution constrained against present-day simulations and elevation changes at the NEEM ice core site (Calov et al., 2015). Despite a possible degree of coincidence, this similarity between results reached by two completely different methods of constraining the simulations gives some confidence in the resulting range."**

*5. While the estimates of GrIS contribution fall well within the range reported in a number of previous studies, dynamical collapse of the WAIS during LIG is new and very important finding presented in the manuscript. Thereby it would be interesting to learn more about the mechanisms. The authors show that Antarctic ice volume overshoot is not related to enhanced surface or subsurface melting, as was proposed in some previous studies, but mostly of dynamical WAIS response to prescribed global sea level rise. In this relation I have a question. What is the crucial difference between the penultimate and the last glaciations which explains this overshoot: much faster sea level rise during the penultimate glaciation or the fact that sea level from Grant et al. (2012) overshoots Holocene sea level by ca. 10 m already at the beginning of LIG? The authors mentioned that they performed similar simulations with the L&R04 sea level reconstruction. Since L&R04 stack suggests a slower rate of sea level rise and does not overshoot present sea level during LIG, I wonder what is the WAIS dynamics in this experiment.*

**The main difference between Termination II and Termination I is indeed the speed of sea-level rise (faster for the penultimate deglaciation than for the most recent deglaciation) and to a lesser extent the fact that the sea-level forcing by itself overshoots the Holocene sea-level stand. A similar experiment with L&R04 sea-level forcing brought to light that the Antarctic ice volume overshoot is reduced by 50% as the rate of sea-level rise is smaller in L&R04 than in the Grant record. The sensitivity experiments discussed in Huybrechts (2002) showed the importance of the speed of bedrock rebound with respect to the speed of sea-level rise to generate overshoot behaviour. With slow isostatic rebound during the last deglaciation (characteristic time scale of 10000 years as compared to 3000 years in the reference experiment having no overshoot), the Antarctic ice volume overshoot was ~4 m SLE, while with very fast isostatic rebound (characteristic time scale of 1000 years), WAIS grounding line retreat got stuck halfway the present-day Ross and Ronne-Filchner ice shelves (or an 'undershoot' of ~ 4 m SLE). This behaviour is easily understood as both sea-level change and bedrock elevation change have a similar effect on**

**grounding-line migration being controlled by hydrostatic equilibrium. If the bedrock rebound after ice unloading is faster than the sea-level rise, this will dampen grounding-line retreat. If on the contrary, the sea-level rise is faster than the bedrock uplift, grounding line retreat will be enhanced, as was the case during the penultimate deglaciation.**

*6. Although the mechanism for the WAIS disintegration found in the study by Goelzer et al. differs from that proposed by Holden et al. (2010), I do not believe that the modeling results presented in the manuscript under consideration can be used to rule out completely importance of submarine melt for stability of the WAIS. The reason is that simulated in the current study bipolar see-saw is very weak compare to other modeling results and paleoclimate data. The later reveal significant temperature overshoots at the beginning of LIG essentially everywhere in the SH, and the magnitude of temperature overshoots (above present) in different Antarctic locations was at least several degrees. At the same time, in the work by Goelzer et al. (2015) only a tiny (0.2C) temperature overshoot is seen in subsurface South Ocean temperature (Fig 7b) and essentially nothing in SH or Antarctic temperatures. This seems to be a typical feature of the LOVECLIM model (e.g. Menviel et al., 2015, EPSL). I believe, this potential caveat of the current study should be mentioned in the discussion.*

**We have included a discussion on this point in the revised manuscript:**

**"The sea-saw effect evoked by NH freshwater forcing leads to millennial time scale temperature variations in the SO, but surface climate over the AIS is hardly affected in our simulations. Despite some improvement when ice sheet changes are included, the limited Antarctic temperature response appears to be a general feature of the LOVECLIM model (e.g. Menviel et al., 2015), which fails to reproduce a several degree warming during the LIG inferred from deep ice core locations. "**

*Specific comments*

*L 82 It should be Pollard et al. (2015)*

**OK.**

*L 182 What is the meaning of "dynamically computed"?*

**The meaning is that insolation is calculated at run time. Removed 'dynamically' to avoid confusion.**

*L 183 Does "governing" means here "major"?*

**Yes. Greenhouse gas forcing is of minor importance and ice sheets have retreated at that time.**

*L 187 "... assumes ice volume to be independent of deep-sea temperatures" This incorrect formulation. In fact, the sea level reconstruction based on Red Sea d18O, unlike benthic d18O, does not require information about deep-sea temperature because it based on planktonic forams. It is also affected by temperature (sea surface temperatures) but to a lesser degree than benthic d18O.*

**OK, reformulated.**

*L 223 Would be useful to clarify how the "stand-alone ice sheet forcing" was defined for penultimate glacial cycle.*

**This was done following established procedures. References have been included in the text to clarify that (Huybrechts and de Wolde, 1999; Huybrechts, 2002).**

*L 255 Would be interesting to know why "the retreat of the WAIS" in the interactive experiment "occurs 2 kyr later compared to the one-way experiment"*

**The reason is differences in atmospheric and oceanic forcing as described in section 5.3.**

*L 310 I fully agree that if "NEEM temperature reconstruction is applied uniformly in space and over seasons, than in any model GrIS will melt completely. However, if Eemian warming had strong seasonality, as proposed by Merz et al. (2015, CP) with large warming in winter and small warming in summer, then in combination with some other factors, "NEEM paradox" can be resolved.*

*L 322 See my previous comment*

**Yes, this is what our discussion in this paragraph is about.**

*L 355. As I already stated in general comment, not much happened in the Southern Hemisphere in response to freshwater forcing in the Northern Hemisphere. This is why it is not surprising that Antarctic temperature is so flat.*

**The amplitude of climate changes in the SH is indeed lower than in the NH. However, the point we are making here is that the Antarctic ice sheet surface climate appears to be largely isolated from those (millennial time scale) changes in the surrounding oceans.**

*L. 370 Would be useful to show also ocean (subsurface) temperature in the respective figure.*

**Instead, we refer now to Goelzer et al 2015, where the ocean response is discussed in detail.**

*L. 411 Which "environmental forcing" is meant here?*

**OK, replaced "environmental" by "climatic".**

*L. 412 It should be Pollard et al. (2015)*

**OK.**

*L. 428 "Ocean expansion is steep. . ." Rather I would say "the fastest sea level rise due to thermal expansion . . ."*

**OK. Replaced "steep" by "rapid" as suggested by other reviewer.**

*L. 440 "0.42+-0.11" This is a typo. Chapter 5 of AR5 does not contain this number. Instead it referrs to the only available estimate of thermal expansion during the LIG of 0.4 +-0.3 m by McKay et al. (2011). In such case I would recommend to cite original publication rather than IPCC report.*

**Thank you for spotting this mistake. Corrected.**

*L. 452 "0.42+-0.11" m is not the estimate of glacier contribution to sea level during the LIG but rather the maximum possible sea level rise due to melting of all existing at present glaciers and small ice caps. Obviously, there is no reason to believe that all glaciers melted completely during the LIG and therefore real contribution of glaciers and ice caps during LIG was probably much smaller than 0.4 m.*

**OK. Reformulated to "maximum possible contribution".**

*L. 523 ". . .by preventing tundra warming affecting proximal ice sheet margins". This is not very clear.*

**OK, reformulated:**

**"This is accomplished by calculating surface temperatures independently for different surface types (ocean, ice sheet, tundra), which most importantly prevents tundra warming to affect proximal ice sheet margins."**

*L. 539 Please correct doi of Berger's paper*

**We have verified the record, this appears to be the correct doi.**

*L 575. Correct reference is "Science, 349, doi: 10.1126/science.aaa4019, 2015"*

**OK.**

*Figure 1. Brovkin et al (1997) is not in the reference list*

**OK. Added reference.**

*L 717 I suppose this is not original Grant et al. (2012) reconstruction but its smoothed version. Please, make it clear.*

**No smoothing has been applied. The maximum probability curve given by Grant et al. (2012) is already as smooth.**

*L 746 Does "forced" here means the same as "one-way"?*

**Yes, modified throughout the manuscript.**

**Reviewer 4 – Eric Wolff**

General comments:

*This paper does represent something of a technical achievement, succeeding in making a coupled run of climate and both Greenland and Antarctic ice sheets across the last interglacial (LIG). To demonstrate that ability, and highlight the steps that are needed to improve on it, I think the paper should eventually be published in CP. How- ever it does need quite a lot of work to explain both details and its limitations correctly. I notice that the paper has already achieved several reviews, so I will not go into huge detail but just give some overall comments, with a little more emphasis on data aspects of the study.*

*The strength of the paper, as I have indicated, comes from the achievement of making such a study. However I think it is important that it is correctly labelled. It is really a demonstration simulation, not a testable prediction. The Greenland ice sheet coupling is achieved only after applying a randomly chosen scaling to the temperature data (it's a tuning in the sense of aiming at a Greenland SL contribution the authors think is sensible, but random in the sense that there is no reason at all to think that a linear tuning is correct). The Antarctic ice sheet apparently responds despite the ice dynamics processes that many glaciologists consider paramount for West Antarctica not being present (or at least I don't think they are). Given these two issues, the actual values that are achieved seem almost meaningless. I don't suggest they should not be explored, and the relative timing of the contributions is of interest for example, but the paper should make much clearer that it does not in any way represent a success in explaining LIG sea level, rather it is a demonstration of how one might start to assess that in a consistent manner.*

*Another significant issue I would like to see addressed concerns data. This is in two senses; firstly some critical data seem a little misquoted, and others seem to be ignored. But also there is an opportunity here to test different aspects of the model results rather than just the SL response. In particular the climate response in both polar regions could be well-tested using the recent Capron et al (2014, QSR) compilation; but in fact this paper is not even cited. I suspect for example*

*that this paper would allow the authors less room to suggest that the Greenland temperature response is overestimated in the model, and force them instead to consider that the ice sheet may be too sensitive, which is quite a critical issue.*

*A final major issue I think the authors need to address concerns the mechanism by which they achieve a significant loss of WAIS – this seems to be global SL and ice shelf viscosity. This seems really surprising to me: global sea level is higher than today really only because of the loss of WAIS in these expts, so it is hard to see why this should be a part of provoking such a loss. That leaves us having to accept that Antarctic temperature in Fig 7a apparently provokes a change in viscosity and loss of ice just a few tenths of a degree above present: this would be a very alarming result, but seems quite at odds with the mechanisms that usually concern people about WAIS (they generally worry about dynamic loss through the major ice streams and glaciers on the Amundsen Sea side, which have little or no ice shelf restraint, rather than the ice flowing into the large ice shelves). Perhaps I have not understood your mechanism but this definitely needs exploring: either your model is way too sensitive to this process, or glaciologists are worrying about the wrong thing and should be very urgently concerned about ice shelf viscosity. I rather suspect the former as I can't see how there can be such a sharp breakpoint in ice shelf viscosity that a couple of degrees would drain the whole of WAIS and destroy the Ross and Ronne-Filchner Ice Shelves. In any case this certainly needs a discussion.*

**Thank you very much for the comments. The referee raises important issues, not all of which can reasonably be answered within the scope of the present paper.**

**We first of all note that a rather detailed comparison of the climate response of LOVECLIM during the LIG with data (without considering Antarctic or Greenland ice-sheet changes) was presented in Loutre et al. (2014), which paper had Emily Capron as co-author, and made extensive reference to Capron et al. (QSR, accepted at that time). In Goelzer et al. (2015) the emphasis was on the effects of prescribed Antarctic and Greenland ice sheet changes on the oceans and atmospheres, and in that paper more comparisons with data were made, also explicitly referring to Capron (2014). The present paper concentrated more on the ice sheets and sea level, and emphasized less the comparison with climate data.**

**We also agree that the possibility of a too sensitive Greenland ice sheet model should not be discarded a priori, but we found little additional elements to support that. As noted further below in reply to the detailed comments, our results are very much in line with other Greenland model**

studies on the LIG, regardless of the mass balance model (e.g. Huybrechts, 2002; Robinson et al., 2011; van de Berg et al., 2013; Calov et al., 2015). Moreover, our PDD surface mass balance model was compared with the Polar MM5, RACMO, and MAR models over Greenland for the period 1960-2008 and found to be even slightly less sensitive than the other models (Vernon et al., 2013), which does not seem indicative of a suspiciously sensitive modelling approach in the present study.

As already mentioned in our reply to question 5 of reviewer 3, we found the main mechanism for WAIS retreat during Termination II to be sea-level rise. The ice volume 'overshoot' of ca. 4 m is primarily a consequence of the delayed bedrock response with respect to the rising sea level, and secondly, of the overshoot in the sea-level forcing itself. Ice shelf viscosity changes also play a role during the deglacial retreat and the sea-level overshoot, but are not dominant. The comparison with future climate warming conditions is however hard to make because of different forcing and different response times. The response time of viscosity changes in the ice shelves is governed by vertical heat conduction, having a characteristic time scale of order 500 years with respect to surface temperature (Huybrechts and de Wolde, 1999). In future warming scenarios, the effect of shelf viscosity changes is therefore usually too slow compared to the anticipated direct effect of increased surface and basal melting rates. For instance, in future warming scenarios with LOVECLIM under 4xCO2 conditions (Huybrechts et al, 2011), we found the ice shelves to be largely gone from melting before they had a chance to warm substantially, and found shelf melt rates to increase 5-fold, compared to the +20% increase for the LIG found here.

More detailed comments:

*Line 47: Turney and Jones compiled data that were not contemporaneous, ie they combined the maximum temperature at each site over a long time slab. It is therefore impossible to deduce a global mean temperature anomaly from their paper. Probably better to acknowledge this.*

OK, we have modified the text to take this comment into account and have used the opportunity to refer to Capron et al., 2014.

**"During the LIG, global mean annual surface temperature is thought to have been 1°C to 2°C higher and peak global annual sea surface temperatures 0.7°C ± 0.6°C higher than pre-industrial (e.g. Turney and Jones, 2010; McKay et al., 2011), with the caveat that warmest phases were assumed globally synchronous in these data syntheses (Masson-Delmotte et al., 2013). These numbers are largely confirmed by a recent compilation, which resolves the temporal temperature evolution (Capron et al., 2014)."**

*Line 56. I think the most commonly cited numbers for LIG sea level are 5-10 m from IPCC AR5, and 6-9 m from the recent Dutton et al (2015, Science) review paper. There is not a great basis for emphasising 6 m in particular.*

**Not changed. The IPCC AR5 literally states "The best estimate is 6 m higher than present" in Section 13.2.1.3, page 1146.**

*Page 4. Here is a first place one could mention the Capron et al compilation which could act as a check on your climate outputs or as a forcing in standalone experiments.*

**We have added a reference to Capron et al. (2014) in the section before (see comment 1) and in the following:**

**"Despite recent advances (e.g. Capron et al., 2014), the fundamental shortcoming at present for improving modelled constraints on the LIG ice sheet contribution to sea level with physical models is the sparse information on LIG polar climate and oceanic conditions"**

*Line 186-188 is badly worded. The Grant et al paper uses an approach that doesn't use synchronisation to a mixed record of SL and deep sea temperatures but it doesn't assume anything about their independence or otherwise does it?*

**OK, reformulated:**

**"The chronology of this data is thought to be superior compared to sea-level proxies based on scaled benthic δ18O records (Grant et al., 2012; Shakun et al., 2015)"**

*Line 192-203. While I understand your decision to scale I think it needs more discussion. From Fig 4a I read off that without forcing you would estimate a Greenland warming of about 3 degrees. This is not only below the NEEM estimate, it's below other NEEM lower estimates (such as Masson-Delmotte et al 2015), and I am pretty sure it is already similar to other model estimates. Your preferred estimate allows only a one degree warming and this would be really hard to reconcile with NEEM data or with compiled SST data in Capron et al. So, for pragmatic reasons, Ok use the scaling, but I feel you should admit that this might be telling you that your Greenland model is too sensitive, and at least discussing your model in the context of others.*

**The crucial temperature for ice-sheet changes is summer temperature at the margin where the melting takes place, and these are higher than 3°C, which we are showing in a new figure (S2) now. We don't think our model is too sensitive, or at least not more sensitive than other models. For one thing, the melt model has been compared with other surface mass balance models and found to be even slightly less sensitive to recent late-20[th] century climate changes (Vernon et al. 2013). See also reply to comment 3 of reviewer 3 and below in response to comment line 314.**

*Line 277. While the elevation at NEEM is not perfectly constrained, I suspect its equally important that ice sheet elevation at NEEM is not a strong constraint on the size/area of GrIS. Perhaps re-word.*

**OK, sentence reworded:**

**"Elevation changes from that ice core are however not well constrained and even if they were, would leave room for a wide range of possible retreat patterns of the northern GrIS (e.g. Born and Nisancioglu, 2012)"**

*Line 284. I am not sure what point you want to make here about Cap Century. The same paper also suggests no ice older than 115 ka at Summit but this is clearly not taken to mean there was no Eemian ice there.*

**Yes, agreed. Sentence removed.**

*Line 314 and around. While we don't understand how an ice sheet at +8 degrees could survive, I still question whether your result illustrates a NEEM paradox or an oversensitive Greenland ice sheet model. You should at least discuss both options.*

**We agree that without further information the results could initially be interpreted as illustrating a too sensitive ice sheet model. However, other elements leave little room for that interpretation. Other surface mass balance models of similar and of higher complexity show a similar or larger sensitivity for the LIG period (e.g. van de Berg et al., 2011). In a comparison and validation for the recent past, the applied melt model is within the range and even slightly less sensitive than the other models (Vernon et al. 2013).**

**We have now included discussion of these aspects in the manuscript. See also response to comment 1 of reviewer 1.**

*Line 353-359 and beyond is really confusing. Firstly you say that "Antarctic surface climate is isolated from millennial fluctuations". But then later you agree with previous authors in ascribing the warm Antarctic to the bipolar seesaw. Please make your text consistent. I assume in fact you do think it is the bipolar seesaw response to NH melting that is important in warming the Antarctic at a time when orbital forcing would cool it.*

The temperature evolution over the Antarctic ice sheet is not showing millennial time-scale variations, which is the case for the surrounding ocean subject to the bipolar see-saw. We have modified the text to clarify that and added a reference to Goelzer et al., 2015, where the SH temperature evolution in response to freshwater fluxes is discussed in detail:

"The surface climate over the AIS appears to be largely isolated from millennial time scale perturbations occurring in the Southern Ocean in response to changing freshwater fluxes in both hemispheres (Goelzer et al., 2015). "

*Fig 6b: I could not follow this figure, please explain it better.*

We have include additional information in the figure caption and in the main text to improve the explanation:

"The underlying surface type with different characteristic albedo values for tundra and ice sheet is determined by the relative amount of ice cover, which is modified when the area of the ice sheet is changing. On much shorter time scales, the albedo can change due to changes in snow depth and also due to changes of the snow cover fraction, which indicates how much surface area of a grid cell is covered with snow (Figure 6b)."

*Fig 10 is really not comprehensible. It needs a much better caption. In any case I am not sure it serves any purpose since the NHIS evolution dominates everything. This means that while the extent of the highstand above present is a prediction that can be aimed at, the shape of the deglacial rise is really dominated by your (prescribed) NHIS loss.*

Figure 10 is given to show that with our modelling approach we can roughly match the reconstructed range of LIG sea-level evolution. The NHIS reconstruction is part of this approach. It was not prescribed to fit with Kopp, but chosen between two alternative reconstructions to give the best climate response (Loutre et al., 2014).

The caption has been updated to explain the percentile curves in the Kopp et al. (2009) reconstruction:

"Modelled sea-level contributions from this study (colour lines) compared to probabilistic sea-level reconstructions (black lines) from Kopp et al. (2009) for the NH (a) the SH (b) and global (c). For the reconstructions, solid lines correspond to the median projection, dashed lines to the 16th and 84th percentiles, and dotted lines to the 2.5th and 97.5th percentiles."

**Modified figures**

[Figure]

Figure 4 Greenland ice sheet forcing characteristics for the reference run (black) and with higher (red) and lower (green) temperature scaling. Climatic temperature anomaly relative to pre-industrial (a). Accumulation rate (b) and runoff rate (c) given as ice sheet wide spatial averages over grounded ice. Calving flux (d), net mass balance (e) and other mass balance terms (b, c) given in water equivalent. Ice area (blue) and ice volume (black) for the reference run (f). All lines are smoothed with a 400 years running mean except for the grey lines giving the full annual time resolution for the reference run. Horizontal dashed lines give the pre-industrial reference values, except for panel e, where it is the zero line.

[Figure]

Figure 5: Greenland ice sheet geometry at 135 kyr BP (a), 130 kyr BP (b), for the minimum ice sheet volume at 123 kyr BP with a SL contribution of 1.4 m (c) and at the end of the reference experiment at 115 kyr BP (d). The red dots indicate the deep ice core locations (from south to northwest: Dye-3, GRIP, NGRIP, NEEM, Camp Century).

[Figure]

Figure 7 Antarctic ice sheet forcing and characteristics. Temperature anomaly relative to pre-industrial (a), average ice sheet wide accumulation rate (b), average ice sheet wide runoff rate (c), average sub-shelf melt rate diagnosed for the area of the present-day observed ice shelves (d) and net mass balance of the grounded ice sheet (e). Mass balance terms (b-e) are given in water equivalent. (f) Grounded ice sheet area (blue) and volume (black). Grey lines give full annual time resolution, while black lines (and blue in f) are smoothed with a 400 years running mean. Horizontal dashed lines give the pre-industrial reference values, except for panel e, where it is the zero line.

[Figure]

Figure 8: Antarctic grounded ice sheet geometry at 135 kyr BP (a), 130 kyr BP (b), for the minimum ice sheet volume at 125 kyr BP with a SL contribution of 4.4 m (c) and at the end of the reference experiment at 115 kyr BP (d).

**Additional figures**

[Figure]

Figure S1 Comparison of modelled East Antarctic temperature evolution with reconstructed temperature changes at deep ice core sites. Modelled temperature anomalies are averaged over a region 72° - 90° S and 0° - 150° E. Ice core temperature reconstructions for the sites EPICA Dronning Maud Land (EDML, 75°00′ S, 00°04′ E), Dome Fuji (DF, 77°19′ S, 39°40′ E), Vostok (VK, 78°28′ S, 106°48′ E) and EPICA Dome C (EDC, 75°06′ S, 123°21′ E) are from Masson-Delmotte et al. (2011).

[Figure]

Figure S2 Comparison of modelled North-East Greenland annual mean (solid) and summer (June-July-August, dashed) surface temperature evolution (72° - 83° N and 306°33′ - 317° 48′ E) with reconstructed temperature changes (grey) at deep ice core site NEEM (77°27′ N, 308°56′ E). The solid grey line is the central estimate and grey dashed lines give the estimated error range for NEEM.

[Figure]

Figure S3: Present-day Antarctic ice sheet configuration from the model (left) compared to observations (right).

[Figure]

[Figure]

Figure S4: Present-day Greenland ice sheet configuration from the model (left) compared to observations (right).

---

## Referee Report (RR1)

**General comments:**

The authors have made a good effort to improve their manuscript and responded to all my points in my initial review. Particularly I appreciate the inclusion of an extensive discussion that clearly helps the reader to put the findings in the right context. I still have a number of remaining issues, which should be addressed before this manuscript goes in print.

**Main points:**

**1. Mass balance for both GrIS and AIS (follow-up of my initial point 5)**
I am not fully satisfied how the authors present their findings (in figures and text) on mass balance changes for both the GrIS and AIS. Although the authors have added some valuable information to the respective figures (Fig. 4 & 8) I still have the feeling that these figures should be created in a way that the reader can see very quickly which terms are the dominant ones for the mass balance changes of the two ice sheets. In particular, in Fig. 8 I have difficulties to figure out what are the dominant processes contributing to the mass balance of the grounded ice sheet and I think this should be revised accordingly.

Specifically I request:

1.1 Please state for both ice sheets what exactly goes into your calculation of the mass balance, so the reader can understand the whole budget. I think this goes best with showing all terms of the budget as a formula. For Antarctica, make clear how the mass balance of the grounded ice sheet connects to the mass balance of the ice shelf.

1.2 As usually done for a budget, all terms of the mass balance should be expressed in the same quantity (in Fig. 4 & 8) – currently you are mixing m/yr (water equivalent) and m$^3$/yr which is confusing.

1.3 In Fig. 4 you show calving flux as a "positive" quantity although I have the understanding that increased calving leads to a decrease in the mass balance. Please clarify this e.g., expressing the budget as a formula (as suggested above).

**2. Section 5.3 (Antarctic ice sheet) still could be improved**
I still feel that the writing of this section could be improved as it is quite hard to read. For example you list a lot processes that you find not to be of crucial importance for the LIG decrease of the AIS before you actually describe the main processes that are driving your mass balance changes. Connected to point 1 I would prefer to have a more systematic description of the mass balance changes.

**3. Sea level rise as main driver of WAIS retreat**
Connected to a comment by Reviewer #4 which you have not really addressed in the revised manuscript: how should one understand that the majority of the LIG sea level high stand (coming from the WAIS) is triggered by a prescribed sea level increase? Is this kind of a positive feedback mechanism that any sea level rise (from whatever process) leads to an additional sea level rise from the WAIS? Please clarify in the manuscript.

Also in I think you could improve your message (for example in the abstract and conclusions) explaining how the sea level rise does lead to a WAIS retreat as currently you just say that the ice shelf viscosity is reduced but not how this explicitly relates to a melting of the ice sheet.

**4. Extend the possible improvements**

I generally like the section 6.6 about possible improvements. I think it should be extended by a discussion of the steps that are needed to come up with a fully-coupled simulation with more "predictive" skill than your current approach (e.g., using an "internally" sea level also for the ice sheet models etc.). Also neither in section 6.5 nor 6.6 you give some insights whether there are remaining issues to be improved in terms of ice sheet processes (i.e. the representation of ice sheet dynamics in models)

**Minor points:**

1. Consistently use "present-day" or "present day"

2. Same as above but for "fully-coupled" and "fully coupled"

3. Line 246: This sentence is somehow confusing: the low scaling factor does not lead to the smallest minimum ice sheet.

4. Table 1: description of the simulations "Forced high" and "Forced low" is confusing. I assume that e.g., "Forced high" is forced with climate output from "High" and equivalently "Forced low" uses output from "Low".

5. In Fig. 4 and the description of it in section 5.2 you describe that Greenland accumulation (Fig. 4b) increases with warmer temperatures (Fig. 4a). But why does the accumulation remain at a high level towards the end of the LIG (120-115ka) when temperatures decrease again?

6. Similar issue as in the point above but for Antarctic accumulation and temperatures (Fig. 8a,b)

7. Fig. 6: Please give a reference for the source of the ice core temperature curves (incl. the uncertainty estimates)

8. Introduce "SO" as abbreviation for "Southern Ocean" at the first instance in the text.

9. Consistently use the "NH" and "SH" abbreviations

10. Since your experimental description is quite lengthy (for good reasons) it would be good to remind the reader of the goals of the study at the beginning of the results (section 5). Make again clear that the focus lies on the comparison of different experiments to show the importance of the fully-coupled approach rather than expecting the "reference" simulation to compare perfectly with observed/reconstructed data.

11. Lines 394-396: Please rephrase this sentence.

12. Line 547: Rasmus et al., 2016 should read Pedersen et al., 2016

---

## Author Response (AR2)

We have revised our manuscript 'Last Interglacial climate and sea-level evolution from a coupled ice sheet-climate model'.

We would like to thank the reviewers for their constructive comments that helped to improve the manuscript further.

Please find below the reviewer's comments in regular italic and a point-by-point rebuttal in bold font.

**Reviewer 1**

*General comments:*

*The authors have made a good effort to improve their manuscript and responded to all my points in my initial review. Particularly I appreciate the inclusion of an extensive discussion that clearly helps the reader to put the findings in the right context. I still have a number of remaining issues, which should be addressed before this manuscript goes in print.*

*Main points:*

*1. Mass balance for both GrIS and AIS (follow-up of my initial point 5)*

*I am not fully satisfied how the authors present their findings (in figures and text) on mass balance changes for both the GrIS and AIS. Although the authors have added some valuable information to the respective figures (Fig. 4 & 8) I still have the feeling that these figures should be created in a way that the reader can see very quickly which terms are the dominant ones for the mass balance changes of the two ice sheets. In particular, in Fig. 8 I have difficulties to figure out what are the dominant processes contributing to the mass balance of the grounded ice sheet and I think this should be revised accordingly.*

*Specifically I request:*

*1.1 Please state for both ice sheets what exactly goes into your calculation of the mass balance, so the reader can understand the whole budget. I think this goes best with showing all terms of the budget as a formula. For Antarctica, make clear how the mass balance of the grounded ice sheet connects to the mass balance of the ice shelf.*

**We have included a description of the net mass balance of both ice sheets in section 3.3 to clarify the contribution to sea-level change:**

**"Changes in the sea-level contribution of the GrIS can be directly related to its integrated net mass balance (*MB*), composed of snow accumulation (*ACC*), surface meltwater runoff (*RUN*), basal melting (*BAS*) and iceberg**

**calving flux (*CAL*):**

$$MB = ACC - RUN - BAS - CAL$$

**Since the GrIS model ignores the small bodies of floating ice in the north, these values are taken over the grounded ice sheet only.**

**For the AIS, *CAL* is replaced by the flux across the grounding line (*GRF*) in the definition of the net mass balance of the grounded ice sheet *MB$_{gr}$*, which needs further corrections to estimate changes in sea level (see below):**

$$MB_{gr} = ACC - RUN - BAS - GRF$$

**The net mass balance of Antarctic floating ice shelves *MB$_{fl}$* given here for completeness includes *GRF* as an additional source term, but does not contribute to sea-level changes in our model:**

$$MB_{fl} = GRF + ACC - RUN - BAS - CAL \text{ ''}$$

*1.2 As usually done for a budget, all terms of the mass balance should be expressed in the same quantity (in Fig. 4 & 8) – currently you are mixing m/yr (water equivalent) and m$^3$/yr which is confusing.*

**We have followed the reviewer's suggestion as far as possible without compromising the usefulness of Figure 4 and 8. We now display the components of the mass budget in m$^3$/yr. For Figure 8 we have decided to maintain shelf melt rate, which is not part of the sea level-relevant mass budget of the grounded ice sheet in units of m/yr. We believe this unit is easier to interpret for readers e.g. familiar with present day melt rates. We have moved this panel down to make that separation clearer.**

*1.3 In Fig. 4 you show calving flux as a "positive" quantity although I have the understanding that increased calving leads to a decrease in the mass balance. Please clarify this e.g., expressing the budget as a formula (as suggested above).*

**The calving flux (amount of produced ice bergs) is in our understanding indeed a positive quantity and calving removes ice from the ice sheet. Therefore, it appears in the mass budget (see response to your comment 1.1) with a minus sign in front. This is comparable to surface meltwater runoff, which is a measurable (positive) quantity that removes ice from the ice sheet.**

*2. Section 5.3 (Antarctic ice sheet) still could be improved*

*I still feel that the writing of this section could be improved as it is quite hard to*

*read. For example you list a lot processes that you find not to be of crucial importance for the LIG decrease of the AIS before you actually describe the main processes that are driving your mass balance changes. Connected to point 1 I would prefer to have a more systematic description of the mass balance changes.*

**We have reformulated some difficult passages, but have decided to keep the overall structure. We believe the section is well structured and follows a clear logic, which we have listed below by paragraph.**

**- Surface forcing and ice sheet response**

**- Role of sub-shelf melting (oceanic forcing)**

**- Area, volume and sea-level contribution**

**- Sensitivity experiments, with specific forcing processes suppressed**

**- Explanation for limited effect of surface and sub-shelf melting (climate forcing too small)**

**- Timing of ice sheet retreat and possible constraints**

**- Impact of sea-level forcing on timing of ice sheet retreat**

*3. Sea level rise as main driver of WAIS retreat*

*Connected to a comment by Reviewer #4 which you have not really addressed in the revised manuscript: how should one understand that the majority of the LIG sea level high stand (coming from the WAIS) is triggered by a prescribed sea level increase? Is this kind of a positive feedback mechanism that any sea level rise (from whatever process) leads to an additional sea level rise from the WAIS? Please clarify in the manuscript.*

**The discussion has been revised in section 6.3 to further clarify the points raised by the reviewer. We now more explicitly link the WAIS retreat during the LIG to a combination of differences in speed of sea-level rise during Termination II and Termination I, the (albeit limited) effect of the additional sea-level rise during the LIG, and the effect of surface warming over the ice shelves. A higher ice-shelf temperature softens and thins the ice, and this promotes additional grounding-line retreat as there is less buttressing and increased thinning at the grounding line.**

**"The main forcing for WAIS retreat during Termination II and the LIG was found to be global sea-level rise from melting of the NH ice sheets, and to a**

**lesser extent surface warming causing a gradual thinning of the ice shelves as the ice softened, contributing to an additional grounding-line retreat as there is less buttressing and increased thinning at the grounding line. These processes also played during Termination I and into the Holocene in simulations with the same ice sheet model (Huybrechts, 2002), but did not produce an overshoot in the sense that the WAIS retreated further inland from its present-day extent. The difference in behaviour between the LIG and the Holocene is mainly the speed of sea-level rise, which was slower during Termination I, and the fact that the global sea-level stand itself did not overshoot the present-day level during the Holocene, giving a less strong forcing. Of particular importance to generate overshoot behaviour is the speed of sea-level rise relative to the speed of bedrock rebound as both control the water depth at the grounding line and hence, grounding-line migration because of the criterion for floatation (hydrostatic equilibrium). If the sea-level rise is fast compared to the bedrock uplift, grounding line retreat will be enhanced, as was the case during Termination II in our model experiments. In that case, the grounding line is able to retreat to a more inland position until the lagged bedrock rebound halts and reverses the process. If on the contrary, the bedrock rebound after ice unloading is fast compared to the sea-level rise, this will tend to dampen grounding-line retreat, as shown in the sensitivity experiments discussed in Huybrechts (2002). "**

*Also in I think you could improve your message (for example in the abstract and conclusions) explaining how the sea level rise does lead to a WAIS retreat as currently you just say that the ice shelf viscosity is reduced but not how this explicitly relates to a melting of the ice sheet.*

**Thanks for the suggestion. However, clarifying issues on processes requires more text than is warranted for the abstract and conclusions and are therefore explained in the main text. Note that WAIS retreat is not caused by melting of the ice sheet (as is the case for GrIS) but results from a dynamic interplay between ice shelf and ice sheet and ensuing grounding-line changes.**

*4. Extend the possible improvements*

*I generally like the section 6.6 about possible improvements. I think it should be extended by a discussion of the steps that are needed to come up with a fully-coupled simulation with more "predictive" skill than your current approach (e.g., using an "internally" sea level also for the ice sheet models etc.). Also neither in section 6.5 nor 6.6 you give some insights whether there are remaining issues to be improved in terms of ice sheet processes (i.e. the representation of ice sheet dynamics in models)*

We have extended the discussion of possible improvements in part with reference to targeting the limitations in the section before. We consider improving representation of ice sheet dynamics of secondary importance as discussed in the added material:

"Ultimately, it would be desirable to apply a consistent sea-level forcing, based on physical models (e.g. de Boer et al., 2014). However, this would require a prognostic model of NH ice sheet evolution (e.g. Zweck and Huybrechts, 2005) and a general solution of the sea-level equation, which would considerably increase complexity and required resources.

Targeting model limitations described in the previous sub-section hinges to a large extent on improving the atmospheric component of the climate model, which equally goes hand in hand with an increase in needed computational resources. Given the large remaining uncertainties in the climate forcing during the LIG and a limited impact of an improved physical approximation for ice flow applied to future projections (Fürst et al., 2013), we consider improving the representation of ice sheet dynamics as of secondary importance. However, fully physical treatment of the surface mass balance solution in a coupled climate-ice sheet model framework, as currently targeted by several groups (e.g. Nowicki et al., 2016) appears like a promising development that may eventually be applied for paleo applications such as the transient LIG simulations of interest in the present paper. "

*Minor points:*

*1. Consistently use "present-day" or "present day"*

OK. We have revised the manuscript in this regard. However, note the difference in usage between e.g. "the present day" as noun and "present-day sea level" as adjective.

*2. Same as above but for "fully-coupled" and "fully coupled"*

OK. We now use "fully coupled" consistently throughout the manuscript in line with the usage in the companion paper.

*3. Line 246: This sentence is somehow confusing: the low scaling factor does not lead to the smallest minimum ice sheet.*

We agree with the reviewer that "the low scaling factor does not lead to the smallest minimum ice sheet". However, the argument here is to find a scaling factor for which the minimum ice sheet extent is covering Camp Century *and* of those we want the one with the smallest extent. This is the exact same situation as in the sentence just before. We have reformulated

**the sentence to clarify the intended meaning.**

**"In practice, the high scaling factor (R=0.5) is chosen to produce the smallest minimum ice sheet extent, which still has ice at the NEEM site. The low scaling factor (R=0.3) was adopted to produce the smallest minimum ice sheet extent, which is still covering Camp Century."**

*4. Table 1: description of the simulations "Forced high" and "Forced low" is confusing. I assume that e.g., "Forced high" is forced with climate output from "High" and equivalently "Forced low" uses output from "Low".*

**The description in the table is correct and necessary to avoid this incorrect assumption. The naming of "high" and "low" refer to the scaling factor in use: "Forced high" has the same scaling factor as "High".**

*5. In Fig. 4 and the description of it in section 5.2 you describe that Greenland accumulation (Fig. 4b) increases with warmer temperatures (Fig. 4a). But why does the accumulation remain at a high level towards the end of the LIG (120-115ka) when temperatures decrease again?*

**Please note that we have modified the figures to display net accumulation and runoff instead of rates as suggested.**

**The later increase for average accumulation rate was because the ice sheet grows into regions with higher accumulation. Net accumulation shows a pronounced increase at the end of the experiment due to the area increase. We have confirmed that the climate model output shows consistent increase in precipitation with warming and decrease with cooling when averaged over continental Greenland as expected.**

*6. Similar issue as in the point above but for Antarctic accumulation and temperatures (Fig. 8a,b)*

**The average accumulation rate over Antarctic grounded ice did show a slight decrease in the second half of the experiment. Net accumulation as displayed now is consistently increasing after 128 kyr BP, basically following changes in grounded ice sheet area.**

*7. Fig. 6: Please give a reference for the source of the ice core temperature curves (incl. the uncertainty estimates)*

**OK. Included a reference to NEEM community members (2013) as suggested.**

*8. Introduce "SO" as abbreviation for "Southern Ocean" at the first instance in the text.*

**OK. This was already the case, but we now also use the abbreviation**

**consistently in the text.**

*9. Consistently use the "NH" and "SH" abbreviations*

**OK. We now use the abbreviations consistently in the text, except for the abstract and Fig. 1 where NH is not yet defined.**

*10. Since your experimental description is quite lengthy (for good reasons) it would be good to remind the reader of the goals of the study at the beginning of the results (section 5). Make again clear that the focus lies on the comparison of different experiments to show the importance of the fully-coupled approach rather than expecting the "reference" simulation to compare perfectly with observed/reconstructed data.*

**Following this suggestion, we have extended the introductory paragraph in section 5, which now reads as follows:**

**"The modelled LIG climate evolution and comparison with proxy reconstructions were presented in detail in two earlier publications (Loutre et al., 2014; Goelzer et al., 2016) for the same climate model setup. In the following, we focus on differences to those two works that arise from a different ice sheet evolution and from the incorporation of feedbacks between climate and ice sheets that are taken into account in our present, fully coupled approach. In addition, we present results pertaining to the ice sheet evolution and simulated sea-level changes."**

*11. Lines 394-396: Please rephrase this sentence.*

**This sentence has been rephrased and now reads:**

**"Pre-industrial surface temperature levels are first reached at 128 kyr BP and then again at 118 kyr BP after cooling throughout the second half of the experiment"**

*12. Line 547: Rasmus et al., 2016 should read Pedersen et al., 2016*

**OK, corrected. We've also corrected the occurrence at line 101.**

**Reviewer 2**

*The revised manuscript improved substantially, and is almost ready to be published. Please find some small suggestions for corrections below:*

We thank the reviewer for the additional comments that we have addressed as specified below.

*- Figure 11 seems to be discussed before Figure 10.*

Not changed. Indeed, Figure 11 is the first time referred to on page 11, well before Figure 10, but also before figure 6 – 9. As summary figure, it contains information relevant at different places in the manuscript. Therefore, we prefer to keep it where it is now, towards the end of the manuscript.

*- Page 16, line 464: change to "control on the timing"*

OK. Modified as suggested.

*- Page 17-18, lines 521-532: Reason for this section is unclear. Please omit or rewrite.*

The discussion in this paragraph leads to the important result that the timing of the GrIS sea-level contribution had to occur late during the LIG. We have made that clearer in the text and now refer to another study, which arrives at the same conclusion based on different ice core data and modelling.

*- Page 18, lines 533-552: Maybe refer also to Landais et al. (under review for CP), as they discuss this further.*

OK. Included a reference as suggested:

"We refer to this mismatch between reconstructed temperatures and assumed minimum ice sheet extent as the "NEEM paradox" (see also Landais et al., 2016)."

*- Page 19, lines 558-561: Please rewrite.*

OK. The relevant text has been rephrased in 6.3 as:

[revised manuscript text omitted]

---

## Author Response (AR3)

**We have requested technical corrections to our manuscript 'Last Interglacial climate and sea-level evolution from a coupled ice sheet-climate model'.**

**Please find below the reviewer's comments in regular italic and our response in bold font.**

**Reviewer 1**

*Dear authors,*

*I am pleased with your revisions and the responses to my remaining issues. I herewith can recommend to publish the manuscript in its current form is. As a technical correction, I would only like to see that you update the references since a number of "Discussion papers" meanwhile have been finalized, e.g., Goelzer et al., 2016, Landais et al., 2016, Merz et al., 2016.*

**Reviewer 2**

*The revised manuscript is ready to be published after updating the following references:*

*Goelzer, H., Huybrechts, P., Loutre, M.-F., and Fichefet, T.: Impact of ice sheet meltwater fluxes on the climate evolution at the onset of the Last Interglacial, Clim. Past, 12, 1721-1737, doi:10.5194/cp-12-1721-2016, 2016.*

*Landais, A., Masson-Delmotte, V., Capron, E., Langebroek, P. M., Bakker, P., Stone, E. J., Merz, N., Raible, C. C., Fischer, H., Orsi, A., Prié, F., Vinther, B., and Dahl-Jensen, D.: How warm was Greenland during the last interglacial period?, Clim. Past, 12, 1933-1948, doi:10.5194/cp-12-1933-2016, 2016.*

*Pedersen, R. A., Langen, P. L., and Vinther, B. M.: Greenland during the last interglacial: the relative importance of insolation and oceanic changes, Clim. Past, 12, 1907-1918, doi:10.5194/cp-12-1907-2016, 2016.*

**Thank you for the reminder. We have updated the references in the manuscript:**

Goelzer, H., Huybrechts, P., Loutre, M. F., and Fichefet, T.: Impact of ice sheet meltwater fluxes on the climate evolution at the onset of the Last Interglacial, Clim. Past, 12, 1721-1737, doi:10.5194/cp-12-1721-2016, 2016.

Landais, A., Masson-Delmotte, V., Capron, E., Langebroek, P. M., Bakker, P., Stone, E. J., Merz, N., Raible, C. C., Fischer, H., Orsi, A., Prié, F., Vinther, B.,

and Dahl-Jensen, D.: How warm was Greenland during the last interglacial period?, Clim. Past, 12, 1933-1948, doi:10.5194/cp-12-1933-2016, 2016.

Merz, N., Born, A., Raible, C. C., and Stocker, T. F.: Warm Greenland during the last interglacial: the role of regional changes in sea ice cover, Clim. Past, 12, 2011-2031, doi:10.5194/cp-12-2011-2016, 2016.

Pedersen, R. A., Langen, P. L., and Vinther, B. M.: Greenland during the last interglacial: the relative importance of insolation and oceanic changes, Clim. Past, 12, 1907-1918, doi:10.5194/cp-12-1907-2016, 2016.

**We have also removed the following reference in text and reference list. "A revision of this discussion paper for further review has not been submitted":**

Langebroek, P. M., and Nisancioglu, K. H.: Moderate Greenland ice sheet melt during the last interglacial constrained by present-day observations and paleo ice core reconstructions, The Cryosphere Discuss., 2016, 1-35, doi:10.5194/tc-2016-15, 2016.